# Non-Stationary Bandits with Auto-Regressive Temporal Dependency

**Qinyi Chen**
Operations Research Center
Massachusetts Institute of Technology
qinyic@mit.edu

**Negin Golrezaei**
Sloan School of Management
Massachusetts Institute of Technology
golrezae@mit.edu

**Djallel Bouneffouf**
IBM Research
djallel.bouneffouf@ibm.com

## Abstract

Traditional multi-armed bandit (MAB) frameworks, predominantly examined under stochastic or adversarial settings, often overlook the temporal dynamics inherent in many real-world applications such as recommendation systems and online advertising. This paper introduces a novel non-stationary MAB framework that captures the temporal structure of these real-world dynamics through an auto-regressive (AR) reward structure. We propose an algorithm that integrates two key mechanisms: (i) an alternation mechanism adept at leveraging temporal dependencies to dynamically balance exploration and exploitation, and (ii) a restarting mechanism designed to discard out-of-date information. Our algorithm achieves a regret upper bound that nearly matches the lower bound, with regret measured against a robust dynamic benchmark. Finally, via a real-world case study on tourism demand prediction, we demonstrate both the efficacy of our algorithm and the broader applicability of our techniques to more complex, rapidly evolving time series.

## 1   Introduction

The multi-armed bandit (MAB) framework [51, 46] is commonly used to study online decision-making under uncertainty. It is primarily examined under either the stochastic setting [51, 6], where arms have fixed unknown reward distributions, or the adversarial setting [7, 5, 31, 16], where an adversary determines the reward distributions that can change arbitrarily over time. However, neither setting accurately represents real-world decision-making problems (see examples in dynamic pricing [30], online advertising [42], and online auctions [33]), where reward distributions change over time following intrinsic temporal structures. These structures often exhibit frequent variations and temporal dependencies, making them challenging to approximate using stationary distributions.

This motivates us to consider a non-stationary bandits setting involving certain temporal structure, which captures the real-world characteristics. Specifically, we are interested in temporal structures with linear amount of changes, which are distinct from the infrequent, limited changes typically handled by change-point detection [25] or seasonality analysis [15]. Such frequent changes are commonly observed in applications such as financial data [14] and click-through rates in online advertising [2], where the data experience lots of volatility within a short time. Traditional bandits algorithms, such as Thompson sampling or UCB, may explore too aggressively in response to these drastic changes, leading to suboptimal performance [39]. That being said, the observed time series in these scenarios typically exhibit certain temporal dependencies between past and present, which can be leveraged to improve decision-making.

To encapsulate the main characteristics of real-world time series, we relax the previously restrictive assumptions on the reward distributions and study a non-stationary MAB problem, where the reward

of each arm evolves over time following an auto-regressive (AR) temporal structure. The AR model is a popular time series model for predicting outcomes in applications such as sales [47], advertising [1] and marketing [18]. It simultaneously captures frequent changes (i.e., linear amount of changes over the time horizon) and temporal dependencies, both of which impact the quality of online decision-making. Although real-world data cannot be perfectly represented by an AR model, it has been shown that the AR parameters accurately capture temporal dependencies and can serve as a useful proxy [47]. The non-stationary bandits framework with an AR reward structure can effectively model a wide range of real-world applications, including:

(1) **recommendation systems**, where MAB can determine the best product/contents to display to users, while the AR model captures the evolution of user preferences and demand over time [21] (see Section 7 for a related case study on tourism demand prediction);

(2) **online advertising**, where MAB adjusts budget allocation for ad campaigns, and the AR model can capture the evolution of click-through rates of ads. The amount of changes are usually linear in time, making the AR structure a potential modeling choice [2];

(3) **financial portfolio management**, where MAB can determine the investment allocation, and the AR model can capture the evolution of returns for each investment/asset [4, 19].

When making online decisions in the presence of frequent changes, traditional methods that are designed for stationary or adversarial settings can be ineffective, due to two main challenges.

1. Balancing exploration and exploitation becomes more challenging, as complete exploration is often unattainable. Continuous exploration is crucial, but there is also a risk of over-exploration. Therefore, finding the right balance becomes essential in navigating this tradeoff effectively.

2. Frequent changes in the environment can significantly diminish the value of past learnings. Failing to promptly leverage the knowledge we have gained can lead to a swift deterioration in our confidence levels regarding the accuracy of our reward estimates.

The goal of this work is to show, via our non-stationary bandits framework with an AR reward structure, that *effectively leveraging knowledge of temporal structure can address these challenges*. In our approach, we would use our knowledge or estimates of the temporal dependency, measured by the AR parameter. Since non-stationary bandits with temporal structures exhibiting such real-world characteristics have been scarcely explored in the literature, we consider the AR structure as a suitable starting point for showcasing key ideas and techniques. We anticipate that this work will inspire future research on bandits with alternative temporal structures with similar features.

## 1.1 Main Contributions

We present an algorithm for the non-stationary AR bandits, called AR2, which stands for "Alternating and Restarting" algorithm for non-stationary "AR" bandits. AR2 features two mechanisms: (i) an *alternation* mechanism that handles the first challenge above by enforcing exploitation at least every other round, and switch to exploration only when we discover an arm with potential, instead of simply adopting the principle of "optimism in the fact of uncertainty" which can lead to over-exploration. (ii) a *restarting* mechanism that handles the second challenge by discarding unnecessary information when our confidence level deteriorates significantly, which balances a second tradeoff known as "remembering"-and-"forgetting"; this tradeoff is crucial even for time series with limited changes [10]. Overall, our work addresses an important gap in the literature by studying non-stationary bandits with temporal structure that exhibit frequent changes and temporal dependency. While the AR structure does not capture all temporal structures in the real world, we believe that our high-level messages and techniques will guide future research on bandits with other temporal structures.

In evaluating the performance of our algorithm, we employ the concept of *per-round steady state dynamic regret* (Definition 2.1). This metric serves as a robust dynamic benchmark that competes with the best arm in each round. It surpasses the static benchmark used in stochastic and adversarial MAB, making it a considerably stronger measure of performance. We provide an upper bound on the regret of AR2 (Theorem 5.2), for which the analysis is rather intricate (Section 6), and show that it almost matches the regret lower bound (Theorem 3.1). Our lower bound result also characterizes the challenges embedded in AR temporal processes, as it shows the per-round dynamic regret does not go to zero as $T$ increases, implying that any algorithm needs to keep exploring over time.

Finally, we conduct a real-world case study on tourism demand prediction [36] in Section 7, confirming the superiority of AR2 compared to benchmark algorithms. There, we also show that the techniques and high-level ideas of our algorithm can be readily extended to handle more complicated, rapidly changing temporal structure (e.g., general AR-$p$ processes) while still achieving good performance. Our case study is complemented by synthetic experiments in Appendix A which show the strength of AR2 against a number of benchmarks designed for stationary and non-stationary settings.

## 1.2 Related Work

Our work, which focuses on MAB with an AR temporal structure, contributes to the non-stationary MAB literature that draws attention in recent years. Most related works on non-stationary bandits focused on either the *rested bandits* [26] or *restless bandits* [53]. In rested bandits, only the expected reward of the arm that is being pulled will change, while in restless bandits, the expected reward of each arm changes at each round according to a known yet arbitrary stochastic process, regardless of whether the arm is pulled. Recently, there has also been a third stream of literature that studies non-stationary bandits where the changes in reward distributions depend on both which arm is pulled and the time period that passes by since its last pull (see, e.g., [32, 13]). The non-stationary bandit problem that we study belongs to the restless setting, since the reward distributions change at each round based on AR processes, independent of our actions. Restless bandits, however, is known to be difficult and intractable [43], and many work thus focus on studying its approximation [28] or relaxation [9].

One line of closely related work on restless bandits studies non-stationary MAB problems with limited amount of changes. These works either assume piecewise stationary rewards (see [25, 37, 12]), or impose a variation budget on the total amount of changes (see [10]). They differ from our setting in that they rely on the amount of changes in the environment to be sublinear in $T$, while in AR processes, changes are more rapid and linear in $T$. Another line of relevant research on restless bandits assumes different types of temporal structures for the reward distributions. See, e.g., [38, 41, 50, 40] for finite-state Markovian processes, [27] for stationary $\phi$-mixing processes and [48] for Brownian processes, which is a special case of an AR model but does not experience the exponential decay in the correlation between the past and the future. Recently, [39] proposed a predictive sampling algorithm that can also be applied to AR-1 bandits; their main focus, however, is to show the algorithm's superiority over the traditional Thompson sampling algorithm in non-stationary environments.

Similar to the second line of works above, our work also aims to exploit the temporal structure of the reward distributions to devise well-performing learning algorithms. However, the AR process is much more unpredictable than the temporal structures studied previously, due to its infinite state space and the fast decay in the values of the past observations. We are thus prompted to take a different approach that is designed specifically for adapting to the rapid changes attributed to the AR process.

## 2 Preliminaries

**Expected rewards.** Consider a non-stationary MAB problem with $k$ arms over $T$ rounds. The state (expected reward) of arm $i \in [k]$ at any round $t \in [T]$ is denoted by $r_i(t)$, where $r_i(t) \in [-\mathcal{R}, +\mathcal{R}]$. Conditioned on the state $r_i(t)$, the realized reward of arm $i$ at round $t$, denoted by $R_i(t)$, is given by $R_i(t) = r_i(t) + \epsilon_i(t)$, where $\epsilon_i(t) \sim N(0, \sigma)$ is independent across arms and rounds and we call $\sigma \in (0, 1)$ the stochastic rate of change. More generally, if each arm $i \in [k]$ has a different, unknown stochastic rate of change $\sigma_i$, as long as we have an upper bound $\sigma \geq \max_i \sigma_i$, all of our theoretical results naturally carries over with the same dependency on $\sigma$.

**Evolution of expected rewards.** The expected reward of each arm $i \in [k]$, $r_i(t)$, evolves over time. In our model, we assume that $r_i(t)$ undergoes an independent AR-1 process:

$$r_i(t + 1) = \alpha(r_i(t) + \epsilon_i(t)) = \alpha R_i(t),$$

or equivalently, $R_i(t + 1) = \alpha R_i(t) + \epsilon_i(t + 1)$, where $\alpha \in (0, 1)$ is the parameter of the AR-1 model and can be used as a proxy to measure temporal correlation over time. As $\alpha$ increases, the temporal correlation increases. As commonly seen in the bandit literature (e.g., [35]), we make the rewards bounded by truncating $r_i(t + 1)$ when its absolute value exceeds some finite boundary value $\mathcal{R} > 0$. That is, $r_i(t + 1) = \mathcal{B}\big(\alpha(r_i(t) + \epsilon_i(t))\big)$, where $\mathcal{B}(y) = \min\{\max\{y, -\mathcal{R}\}, \mathcal{R}\}$.

Here, we assume that the AR parameter $\alpha$ is known to the decision maker, a common assumption in literature [10, 27, 48], and often justifiable in real-world scenarios where sufficient historical data enables accurate fitting of AR models (see Section 7). This assumption is made to simplify the analysis. However, in Section 8, we would show that this assumption can be relaxed, and the AR parameter can be effectively estimated via maximum likelihood estimation.

We also assume here that all arms share the same AR parameter $\alpha$. As our numerical studies in Appendix A and Section 8 suggest, this assumption can be relaxed too. There, we generalize our model by considering heterogeneous AR parameters $\alpha_i$ for each arm $i \in [k]$, and our algorithm maintains its performance. Nonetheless, it is worth noting that the homogeneity of AR parameter can be justified in some applications. One example can be seen in our case study in Section 7, where a travel agency offers vacation packages to travelers originating from a particular origin and heading

to a specific destination. In this scenario, the demand is modeled as a general AR-4 process, with the AR parameters being consistent across all arms. This is because the temporal correlation here is primarily determined by exogenous factors related to the travel route, such as popular attractions and the availability of flights between the two locations, hence impacting all arms in the same way.

**Feedback structure and goal.** Our goal is to design an algorithm that obtains high reward, against a strong dynamic benchmark that in every round $t$, pulls an arm that has the highest expected reward. This benchmark is much more demanding than the time-invariant benchmark used in traditional MAB literature. Obtaining good performance against the dynamic benchmark is particularly challenging as in any round $t \in [T]$, only the reward of the pulled arms can be observed. Not being able to observe the reward of unpulled arms in dynamic and time-varying environments introduces *missing values* in the AR process associated with arms. This, in turn, deteriorates the prediction quality of the future expected rewards and the performance of any algorithm that relies on such predictions (see [3] for a work that studies the impact of missing values on the prediction quality in AR processes).

We now formally define how we measure the performance of any non-anticipating algorithm.

**Definition 2.1** (Dynamic steady state regret). *Let $\mathcal{A}$ be a non-anticipating algorithm that, at each round $t \in [T]$, pulls arm $I_t$ based on the history $\{I_1, R_1, \ldots, I_{t-1}, R_{t-1}\}$, where $R_{t'} = R_{I_{t'}}(t')$ is the observed reward at round $t' < t$. The dynamic regret of $\mathcal{A}$ at round $t$ is defined as*

$$\text{REG}_{\mathcal{A}}(t) = r^\star(t) - r_{I_t}(t),$$

*where $r^\star(t) = \max_{i \in [k]}\{r_i(t)\}$ is the maximum expected reward at round $t$. Furthermore, the per-round steady state regret of algorithm $\mathcal{A}$ is defined as*

$$\overline{\text{REG}}_{\mathcal{A}} = \lim\sup_{T \to \infty} \mathbb{E}\left[\frac{1}{T}\sum_{t=1}^{T} \text{REG}_{\mathcal{A}}(t)\right].$$

We remark that in Definition 2.1, our per-round regret is defined asymptotically mainly because we would like to focus on evaluating the steady state performance of our algorithm (i.e., under the steady state distribution).[1] All of our theoretical analyses for regret lower bound (Section 3) and upper bounds (Section 5) are also performed under the steady state distribution.

In this work, we would like to design an algorithm with a small per-round steady state regret, where the per-round regret is a function of the stochastic rate of change ($\sigma$) and the temporal correlation ($\alpha$).

## 3 Regret Lower Bound

We now provide a lower bound on the best achievable dynamic per-round steady state regret.

**Theorem 3.1** (Regret Lower Bound). *Consider a non-stationary MAB problem with $k$ arms, where the expected reward of each arm evolves as an independent AR-1 process with parameter $\alpha$, stochastic rate of change $\sigma$ and truncating boundaries $[-\mathcal{R}, \mathcal{R}]$. Then the per-round steady state regret of any algorithm $\mathcal{A}$ is at least $\Omega(g(k, \alpha, \sigma)\alpha\sigma)$, where $g(k, \alpha, \sigma)$ is the probability that two best arms are within $\alpha\sigma$ distance of each other at any given round, under the steady state distribution. See the expression of this probability in Equation (16) in Appendix F.*

The proof of Theorem 3.1, provided in Appendix F, builds on the following idea: even if algorithm $\mathcal{A}$ has access to all past information at round $t$, due to stochastic noise $\epsilon_i(t)$, $\mathcal{A}$ will pull a sub-optimal arm with constant probability at round $t + 1$. We show that the probability of the two best arms' expected rewards being within $\alpha\sigma$ of each other can be expressed as $g(k, \alpha, \sigma)$ (see (16)).[2] If the two best arms are $\alpha\sigma$ close, $\mathcal{A}$ will incur $\Omega(\alpha\sigma)$ regret with constant probability at round $t + 1$.

In the extreme case when $\alpha$ is close to one, the steady state distribution of $r_i(t)$ can be approximated with a uniform distribution within the boundaries and two probability masses at the boundaries (see Appendix D). Using the uniform approximation, one can show that $g(k, \alpha, \sigma) = \Omega(k\alpha\sigma)$ (see discussion in Appendix F). This then yields a regret lower bound of order $\Omega(k\alpha^2\sigma^2)$.

Theorem 3.1 implies that under our setup, the best achievable per-round regret with respect to a strong dynamic benchmark does not converge to zero, which differs from stationary or adversarial

---

[1]If we assume that the initial state $r_i(0)$ is drawn from the steady state distribution, defined in Appendix D, we can simply define our per-round regret as $\overline{\text{REG}}_{\mathcal{A}} = \mathbb{E}\left[\frac{1}{T}\sum_{t=1}^{T}\text{REG}_{\mathcal{A}}(t)\right]$, and all of our results remain valid. This also matches the definition of dynamic regret in works such as [10, 11, 39].

[2]See, also, Appendix E.1 for an illustration of how $g(k, \alpha, \sigma)$ evolves with respect to $\sigma$ under different $\alpha$.

---

**Algorithm 1** Alternating and Restarting algorithm for non-stationary AR bandits (AR2)

---

**Input:** AR Parameter $\alpha$, stochastic rate of change $\sigma$, epoch size $\Delta_{\text{ep}}$, parameter $c_0$.

1. Set the epoch index $s = 1$, parameter $c_1 = 24c_0$.

2. Repeat while $s \leq \lceil T/\Delta_{\text{ep}} \rceil$:

   (a) Initialization: Set $t_0 = (s-1)\Delta_{\text{ep}}$. Set the initial triggered set $\mathcal{T} = \emptyset$. Pull each arm $i \in [k]$ at round $t_0 + i$ and set $\tau_i = t_0 + i$. Set estimates $\widehat{r}_i(t_0 + k + 1) = \alpha^{k-i} \cdot \mathcal{B}(\alpha R_i(t_0 + i))$.

   (b) Repeat for $t = t_0 + (k+1), \ldots, \min\{t_0 + \Delta_{\text{ep}}, T\}$

   - Update the identity of the superior arm and its estimated reward

   $$i_{\text{sup}}(t) = \begin{cases} I_{t-1} & \text{if } \widehat{r}_{I_{t-1}}(t) \geq \widehat{r}_{I_{t-2}}(t) \\ I_{t-2} & \text{if } \widehat{r}_{I_{t-1}}(t) < \widehat{r}_{I_{t-2}}(t) \end{cases} \quad \text{and} \quad \widehat{r}_{\text{sup}}(t) = \widehat{r}_{i_{\text{sup}}(t)}(t). \tag{1}$$

   - Trigger arms with potential: For $i \notin \mathcal{T} \cup \{i_{\text{sup}}(t)\}$, trigger arm $i$ if

   $$\widehat{r}_{\text{sup}}(t) - \widehat{r}_i(t) \leq c_1 \sigma \sqrt{(\alpha^2 - \alpha^{2(t-\tau_i+1)})/(1-\alpha^2)}. \tag{2}$$

   If triggered, add $i$ to $\mathcal{T}$ and set $\tau_i^{\text{trig}} = t$.
   - Alternate between exploration and exploitation:
     - If $t$ is odd and $\mathcal{T} \neq \emptyset$, pull a triggered arm with the earliest triggering time: $I_t = \arg\min_{j \in \mathcal{T}} \tau_j^{\text{trig}}$.
     - Otherwise, pull the superior arm $I_t = i_{\text{sup}}(t)$.
   - Receive a reward $R_{I_t}(t)$, and set $\tau_{I_t} = t$.
   - Maintain Estimates: Set $\widehat{r}_{I_t}(t+1) = \mathcal{B}(\alpha R_{I_t}(t))$ and set $\widehat{r}_i(t+1) = \alpha \widehat{r}_i(t)$ for $i \neq I_t$.

   (c) Set $s = s + 1$.

---

bandits with a time-invariant benchmark where algorithms like UCB for stationary bandits and Exp3 for adversarial bandits can achieve zero per-round regret as $T$ approaches infinity. In our setting, each arm undergoes linear amount of changes, making it challenging for any algorithm to adapt to the changing environment in time. Our result aligns with the lower bound from [10], which states that when the total variation of expected rewards is $O(T)$, the regret also grows linearly.

## 4  Algorithm AR2

In this section, we present our algorithm, called AR2 (Alternating and Restarting algorithm for non-stationary "AR" bandits). Algorithm 1 outlines the workings of AR2, which operates in epochs of a fixed length. Within each epoch, AR2 maintains and updates estimates of the expected rewards for each arm $i \in [k]$. It alternates between exploitation and exploration steps based on these estimates. During exploitation, AR2 plays a "superior arm" expected to yield high rewards, while in exploration, it selects a "triggered arm" that hasn't been pulled recently but has high potential. At the end of an epoch, the algorithm restarts. As alluded earlier, AR2 effectively balances two inherent tradeoffs in non-stationary MAB problems with AR reward structure. Firstly, it addresses the exploration-exploitation tradeoff by alternating between exploiting the superior arm and exploring the triggered arm within each epoch. Secondly, it handles the tradeoff between "remembering"-and-"forgetting", a commonly considered tradeoff in non-stationary environments [10, 11], via restarting.

**Maintaining estimates of expected reward of arms.** For any round $t$ and arm $i \in [k]$, let $\tau_i(t)$ be the last round before $t$ (including $t$) at which arm $i$ is pulled, and let $\tau_i^{\text{next}}(t)$ be the next round after $t$ (excluding $t$) at which arm $i$ is pulled. Let $\Delta \tau_i(t) = \tau_i^{\text{next}}(t) - \tau_i(t)$ be the gap between two consecutive pulls of arm $i$ around $t$. Define $\widehat{r}_i(t)$ as the estimate of the reward of arm $i$ at $t$ based on the most recent observed reward of arm $i$ (i.e., $R_i(\tau_i(t))$). Via recursive updates (see Step (b)), AR2 maintains the following estimate of expected reward for each arm:

$$\widehat{r}_i(t) = \alpha^{t-\tau_i(t)-1}\widehat{r}_i(\tau_i(t)+1) = \alpha^{t-\tau_i(t)-1}\mathcal{B}(\alpha R_i(\tau_i(t))).$$

**Superior arms.** The superior arm at round $t$, denoted by $i_{\text{sup}}(t)$, is one of the two most recently pulled arms, i.e., $I_{t-1}$ or $I_{t-2}$, that has the higher estimated reward (see (1)). We further define $\widehat{r}_{\text{sup}}(t) = \widehat{r}_{i_{\text{sup}}(t)}(t)$ as the estimated reward of the superior arm at round $t$. We remark that here, for simplicity of analysis, our definition of the superior arm only considers the two most recently pulled arms. We can in fact set the superior arm to be the one with the highest estimated reward among the $m$ most recently pulled arms for any constant $m \geq 2$. A similar theoretical analysis will then yield the same theoretical guarantee, as shown in Theorem 5.2. We further verify the robustness of AR2 to the choice of $m$ in our numerical studies, where we consider all arms as potential candidates for the superior arm, and AR2 maintains its competitive performance (see Appendix A).

**Triggered arms.** In order to adapt to changes, AR2 identifies and keeps track of a set of arms with potential that are not pulled recently. We refer to these arms as *triggered arms* and denote their associated set by $\mathcal{T}$. In a round $t$, an arm $i \neq i_{\sup}(t)$ gets *triggered* if the triggering criteria in (2) is satisfied. We call the earliest such time $t > \tau_i(t)$ as the *triggering time* of arm $i$ and denote it as $\tau_i^{\text{trig}}(t)$. If arm $i$ is triggered, it is added to the triggered set $\mathcal{T}$. When arm $i$ gets pulled or is chosen as the superior arm, it is removed from $\mathcal{T}$. The right-hand side of the triggering criteria is a confidence bound constructed based on Hoeffding's Inequality (see details in Section 6 and Appendix H).

Note that at the exploration step, AR2 would only pull a triggered arm if the triggered set $\mathcal{T}$ is non-empty. This inherently adjusts the rate of exploration in our alternation mechanism. In a slowly-changing environment (e.g., with small $\sigma$), the triggered set may not always include arms needing exploration, thus allowing focused exploitation of the superior arm; in a fast-changing environment, the rate of exploration can be as high as the rate of exploitation.

## 5 Regret of Algorithm AR2

Our algorithm AR2 works for AR-1 model with any choice of parameter $\alpha \in (0, 1)$. However, it turns out that the problem is less challenging when the future is weakly correlated with the past (i.e., when $\alpha$ is small). Theorem 5.1 states that *any* algorithm—including both a naive approach that continuously pulls the same arm throughout the horizon and our algorithm AR2—can achieve near-optimal performance when $\alpha$ is too small. This is because with small $\alpha$, (i) for any arm $i \in [k]$, the steady state distribution of its expected reward $r_i(t)$ concentrates around zero with high probability, and (ii) our observations of past rewards quickly deteriorate in their value of providing insights on the evolution of expected rewards in the future.

**Theorem 5.1.** *Any non-anticipating algorithm $\mathcal{A}$ incurs per-round steady state regret of at most*
$O\left( \min(\sqrt{\log(1/\alpha\sigma) + \log k} \cdot \frac{\alpha\sigma}{\sqrt{1-\alpha^2}}, 2\Re) \right).$

In Figure 1, we compare the order of the regret upper bound in Theorem 5.1 with the order of the regret lower bound in Theorem 3.1.[3] Observe that $\bar{\alpha} \triangleq 0.5$ serves as a rough threshold value such that when $\alpha \in (0, \bar{\alpha})$, the orders of lower and upper bounds almost match each other, which suggests limited room for improvement. (That being said, our numerical studies in Appendix A in fact show that AR2 still outperforms other benchmarks even when $\alpha$ is small.) On the other hand, as $\alpha \rightarrow 1$, the gap between lower and upper bounds start to expand, implying that the problem becomes more difficult and simplistic approaches such as the naive algorithm would no longer produce satisfying performance.

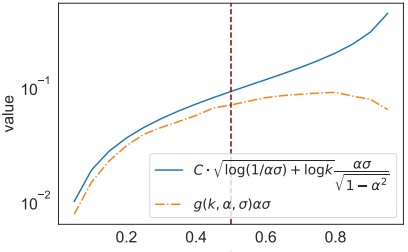

Figure 1: Comparison of the orders of the regret upper bound in Theorem 5.1 and the lower bound in Theorem 3.1, obtained with $k = 5, \sigma = 0.2, C = 0.4$. (Similar plots can be obtained for different values of $k$ and $\sigma$.)

In the rest of the discussion, we thereby focus on the more challenging regime (i.e., $\alpha \in [\bar{\alpha}, 1)$), and establish a regret upper bound for AR2 when $\alpha \in [\bar{\alpha}, 1)$.

**Theorem 5.2.** *Let $\mathcal{K}(\alpha) \triangleq \lfloor \frac{1}{2}(\log(\frac{1}{8})/\log \alpha + 1) \rfloor$, and let $\mathcal{A}$ be the AR2 algorithm that gets restarted in each epoch of length $\Delta_{ep} = \lceil k\alpha^{-3}\sigma^{-3} \rceil$. If $\alpha \in [\bar{\alpha}, 1)$ and $k \leq \mathcal{K}(\alpha)$, the per-round steady state regret of $\mathcal{A}$ satisfies $\overline{\text{REG}}_{\mathcal{A}} \leq O(c_0^2\alpha^2\sigma^2 k^3 \log(c_0\alpha\sigma\sqrt{k}))$, where $c_0 = \sqrt{4\log(1/\alpha\sigma) + 4\log \Delta_{ep} + 2\log(4k)}$.*

Recall that if $\alpha$ is close to one, the regret lower bound can be characterized as $\Omega(k\alpha^2\sigma^2)$. This shows that our algorithm is optimal in terms of AR parameter $\alpha$ and stochastic rate of change $\sigma$ (up to logarithmic factors). In Appendix E.3, we further highlight the significance of Theorem 5.2 by illustrating the evolution of upper and lower bounds, as well as the per-round regret attained by AR2, at different values of $\alpha$ and $\sigma$. We show that Theorem 5.2 well characterizes the performance of AR2 under different settings.

We remark that in terms of the dependency on $k$, Theorem 5.2 requires $k \leq \mathcal{K}(\alpha)$ mainly for the rigor of theoretical analysis, and this bound loosens as $\alpha$ approaches one (for example, when $\alpha = 0.95$,

---

[3]See, also, Figure 5 in Appendix E.2 for a comparison of upper/lower bounds with respect to $\sigma$.

$\mathcal{K}(\alpha) = 20.$).[4] Our numerical studies in Appendix A also reveal that (i) AR2 maintains its competitive performance even when the number of arms $k$ exceeds $\mathcal{K}(\alpha)$; and (ii) its per-round regret grows modestly with $k$. These suggest that the assumption/dependency on $k$ for our upper bound are artifacts of our analysis, rather than an intrinsic property of our algorithm.

## 6 Proof of Theorem 5.2

The proof of Theorem 5.2 proceeds in four steps. Step 1 introduces *distributed regret*, a novel notion that distributes instantaneous regret over subsequent rounds, enabling us to rephrase per-round steady state regret. Step 2 defines a high-probability *good event* crucial for analyzing distributed regret. In Step 3, we leverage the good event to establish an upper bound for each round's distributed regret. Finally, Step 4 aggregates these regrets across rounds and epochs, establishing the per-round steady state regret of AR2 against the dynamic benchmark.

**Step 1: Distributed regrets.** Let $\Delta r_i(t) = r^\star(t) - r_i(t)$ be the instantaneous regret from pulling arm $i$ at round $t$, where $r^\star(t) = \max_{i \in [k]} r_i(t)$ is the maximum expected reward at round $t$. Consider the expected per-round regret: $\overline{\mathrm{REG}}_{\mathcal{A}} = \mathbb{E}\left[\frac{1}{T} \sum_{t=1}^{T} \mathrm{REG}_{\mathcal{A}}(t)\right] = \frac{1}{T} \sum_{t=1}^{T} \mathbb{E}\left[\Delta r_{I_t}(t)\right]$, where $\Delta r_{I_t}(t) = r^\star(t) - r_{I_t}(t)$ is the instantaneous regret incurred at round $t$. In the following, we show that we can distribute this instantaneous regret over subsequent rounds.

Note that for any $i \in [k]$, let $\ell_i^{(j)}$ denote the round at which arm $i$ gets pulled for the $j$th time within an epoch. Then during the period between two consecutive pulls $\ell_i^{(j)} \leq t < \ell_i^{(j+1)}$, the only regret incurred from pulling arm $i$ is the instantaneous regret $\Delta r_i(\ell_i^{(j)})$. By previous definition, we also have $\tau_i(t) = \ell_i^{(j)}$, $\tau_i^{\mathrm{next}}(t) = \ell_i^{(j+1)}$ and $\Delta \tau_i(t) = \ell_i^{(j+1)} - \ell_i^{(j)}$ for all $\ell_i^{(j)} \leq t < \ell_i^{(j+1)}$. Observe that we can decompose $\Delta r_i(\ell_i^{(j)})$ as follows, which then leads to the notion of *distributed regret*:

$$\Delta r_i(\ell_i^{(j)}) = \sum_{t=\ell_i^{(j)}}^{\ell_i^{(j+1)}-1} \left( \frac{\Delta r_i(\ell_i^{(j)}) \alpha^{2(t-\ell_i^{(j)})}}{1 + \alpha^2 + \cdots + \alpha^{2(\ell_i^{(j+1)}-1-\ell_i^{(j)})}} \right) = \sum_{t=\ell_i^{(j)}}^{\ell_i^{(j+1)}-1} \left( \frac{\Delta r_i(\tau_i(t)) \alpha^{2(t-\tau_i(t))}}{1 + \alpha^2 + \cdots + \alpha^{2(\Delta \tau_i(t)-1)}} \right).$$

**Definition 6.1** (Distributed regret). *The distributed regret of arm $i$ at round $t$ is defined as*

$$D_i(t) = \left( \frac{\Delta r_i(\tau_i(t))}{1 + \alpha^2 + \cdots + \alpha^{2(\Delta \tau_i(t)-1)}} \right) \alpha^{2(t-\tau_i(t))} \tag{3}$$

Note that $D_i(t) = 0$ if arm $i$ is the best arm at round $\tau_i(t)$. Now, let $T_i$ be the total number of rounds we pull arm $i$ in the horizon, we can rewrite the expected per-round regret as

$$\overline{\mathrm{REG}}_{\mathcal{A}} = \frac{1}{T} \sum_{i=1}^{k} \sum_{j=1}^{T_i} \mathbb{E}\left[\Delta r_i(\ell_i^{(j)})\right] = \frac{1}{T} \sum_{i=1}^{k} \sum_{i=1}^{T} \mathbb{E}\left[D_i(t)\right] = \frac{1}{T} \sum_{i=1}^{k} \sum_{s=1}^{S} \sum_{t=(s-1)\Delta_{\mathrm{ep}}+1}^{s\Delta_{\mathrm{ep}}} \mathbb{E}[D_i(t)],$$
$$\tag{4}$$

where the second equality follows from the Definition 6.1 and the third equality follows from that AR2 proceeds in epochs of length $\Delta_{\mathrm{ep}}$, and $S = T/\Delta_{\mathrm{ep}}$ is the total number of epochs.

**Step 2: Good event and its implications.** Before proceeding, we first define a good event $\mathcal{G}(t)$ at round $t$, which would help simplify our analysis. In principle, we say that a good event $\mathcal{G}(t)$ happens at round $t$ if the noises within the epoch including $t$ are not too large in magnitude. Recall from Section 2 that $r_i(t)$ follows an AR-1 process with truncating boundaries: $r_i(t+1) = \mathcal{B}\left(\alpha(r_i(t) + \epsilon_i(t))\right)$, where $\mathcal{B}(y) = \min\{\max\{y, -\mathcal{R}\}, \mathcal{R}\}$. Hence, we need to first define a new noise term that shows the influence of the truncating boundary.

$$\tilde{\epsilon}_i(t) = \begin{cases} \epsilon_i(t) & \text{if } \alpha(r_i(t) + \epsilon_i(t)) \in [-\mathcal{R}, \mathcal{R}] \\ \frac{1}{\alpha}[\mathcal{B}\left(\alpha(r_i(t) + \epsilon_i(t))\right) - \alpha r_i(t)] & \text{otherwise} \end{cases} \tag{5}$$

---

[4]Here, the bound $\mathcal{K}(\alpha)$ results from our loose upper bound for the number of triggered arms in the proof of Lemma H.3, where we used the fact that there are at most $k - 1 < \mathcal{K}(\alpha)$ triggered arms at any round to limit how quickly the values of the past observations diminish. If, at any given round, the number of triggered arm is $O(1)$, the upper bound $\mathcal{K}(\alpha)$ would no longer be required, and the regret upper bound can be further reduced to $O(c_0^2 \alpha^2 \sigma^2 k \log(c_0 \alpha \sigma))$, which matches the lower bound up to logarithmic factors.

In particular, the new noise $\tilde{\epsilon}_i(t)$ satisfies the following recursive relationship: $r_i(t+1) = \alpha r_i(t) + \alpha\tilde{\epsilon}_i(t)$. We now formally define the good event $\mathcal{G}(t)$, which states that for any sub-interval $[t_0, t_1]$ within the epoch including $t$, the weighted sum of $\tilde{\epsilon}_i(t)$ satisfies a concentration inequality.

**Definition 6.2** (Good event at round $t$). *We say that the good event $\mathcal{G}(t)$ occurs at round $t$ if for every $i \in [k]$, and for every sub-interval $[t_0, t_1]$, where $t - \Delta_{ep} \le t_0 < t_1 \le t + \Delta_{ep}$, the weighted sum of the noises $\tilde{\epsilon}_i(t)$ satisfies $\left| \sum_{t=t_0}^{t_1-1} \alpha^{t_1-t}\tilde{\epsilon}_i(t) \right| < c_0\sigma\sqrt{(\alpha^2 - \alpha^{2(t_1-t_0+1)})/(1-\alpha^2)}$, where $c_0 = \sqrt{4\log(1/\alpha\sigma) + 4\log\Delta_{ep} + 2\log(4k)}$.*

By building a connection between $\tilde{\epsilon}_i(t)$ and $\epsilon_i(t)$ and Hoeffding's inequality, we show Lemma 6.3, which confirms that the good event $\mathcal{G}(t)$ happens *with high probability*.

**Lemma 6.3.** *For any $t \in [T]$, we have $\mathbb{P}[\mathcal{G}^c(t)] \le (\alpha\sigma)^2$.*

**Step 3: Distributed regret analysis.** For a given round $t$ in the $s$-th epoch, we can bound its expected distributed regret by first decomposing it into two terms: $\mathbb{E}[D_i(t)] = \mathbb{E}[D_i(t)\mathbb{1}_{\mathcal{G}^c(t)}] + \mathbb{E}[D_i(t)\mathbb{1}_{\mathcal{G}(t)}]$, where $\mathbb{1}_{\mathcal{E}}$ is the indicator function of event $\mathcal{E}$. We can bound the first term by applying Lemma 6.3

$$\mathbb{E}[D_i(t)\mathbb{1}_{\mathcal{G}^c(t)}] \le 2\mathcal{R} \cdot \mathbb{P}[\mathcal{G}^c(t)] \le 2\mathcal{R} \cdot (\alpha\sigma)^2 = O(\sigma^2\alpha^2). \tag{6}$$

To bound the second term, we rely on the implications of the good event $\mathcal{G}(t)$ (presented in Appendix H.1) to show the following.

**Lemma 6.4.** *Suppose that we apply $\mathtt{AR2}$ to the non-stationary MAB problem with $\alpha \in [\bar{\alpha}, 1)$ and and $k \le \mathcal{K}(\alpha)$, for every arm $i \in [k]$ and some round $\tau_i(t) \le t < \tau_i^{\text{next}}(t)$, where $\tau_i(t)$ and $\tau_i^{\text{next}}(t)$ are two consecutive rounds at which $i$ gets pulled, we have $\mathbb{E}[D_i(t)\mathbb{1}_{\mathcal{G}(t)}] \le O(c_0^2\alpha^2\sigma^2k^2\log(c_0\sigma\alpha\sqrt{k}))$.*

The proof of Lemma 6.4 is deferred to Appendix H.2. There, we provide a bound for the nominator and denominator of $D_i(t)$ respectively, conditioning on the good event $\mathcal{G}(t)$ and the value of $\Delta r_i(t)$. In the proof, we critically use the implications of the good event (see Appendix H.1). Now, combining (6) and Lemma 6.4, for any round $t$ within the $s$th epoch such that $\tau_i(t) \le t < \tau_i^{\text{next}}(t)$, we have

$$\mathbb{E}[D_i(t)] = \mathbb{E}[D_i(t)\mathbb{1}_{\mathcal{G}^c(t)}] + \mathbb{E}[D_i(t)\mathbb{1}_{\mathcal{G}(t)}] \le O(c_0^2\sigma^2\alpha^2k^2\log(c_0\sigma\alpha\sqrt{k})). \tag{7}$$

**Step 4: Summing the distributed regrets.** Given Equations (4), (7) and $\Delta_{\text{ep}} = \lceil k\alpha^{-3}\sigma^{-3} \rceil$, summing the expected distributed regrets first within an epoch, and then along the horizon yields

$$\overline{\text{REG}}_{\mathcal{A}} = \frac{1}{T}\sum_{i=1}^{k}\sum_{s=1}^{S}\sum_{t=(s-1)\Delta_{\text{ep}}+1}^{s\Delta_{\text{ep}}} \mathbb{E}[D_i(t)] \le \frac{1}{T}\sum_{i=1}^{k} S \cdot [O(\tilde{C}^2\log(\tilde{C}))\Delta_{\text{ep}} + 2\mathcal{R}] = O(k\tilde{C}^2\log(\tilde{C})),$$

where the additional $2\mathcal{R}$ term comes from the initialization round and the last round we pull arm $i$ within an epoch. This concludes the proof if we plug in $\tilde{C} = c_0\alpha\sigma\sqrt{k}$. ∎

# 7 A Real-World Case Study on Tourism Demand Prediction

In this section, we numerically demonstrate the efficacy of $\mathtt{AR2}$ via a real-world case study based on a international Tourism Demand dataset for Australia [36]. In this case study, we act as a travel agency that needs to determine which vacation package to offer, where the demand for each vacation package is highly dependent on the tourism demand during each quarter. Our case study is further complemented by a number of synthetic experiments in Appendix A, where we compare $\mathtt{AR2}$ against various benchmarks designed for both stationary and non-stationary settings and again show the superior performance of $\mathtt{AR2}$ in adapting to the rapidly changing environment.

**Dataset and setup.** The international tourism demand dataset [36], obtained from the Australian Bureau of Statistics, records the number of individual tourist arrivals to Australia from Hong Kong during each quarter between the years 1975-1989. The authors of [36] have fitted an AR model to the logarithms of quarterly tourist arrivals, which results in an AR-4 model with a trend component[5]: $r_i(t) = -0.01 + 0.32R_i(t-2) + 0.6R_i(t-4)$. Note that here, the number of tourist arrivals is strongly correlated with the number of arrivals four quarters before, which is likely due to seasonal patterns. The time series analysis in [36] also supports our assumption in Section 2 that the AR

---

[5]We keep the significant lags in forecasting tourist arrivals, which are the second and fourth lags.

parameters are known. In this dataset, as well as many others from the real world, decision-makers are likely to have access to historical data, which enables them to fit AR processes with good precision.

Given the AR model, we consider a recommendation setting where the travel agencies have $k = 5$ vacation packages (arms) to offer to the tourists from Hong Kong, and wish to dynamically feature the package with the highest demand (reward) at each round. For each arm, we simulate its reward sequence by randomly drawing the initial reward from a uniform distribution in $[0, 1]$ and simulate $r_i(t)$ for a total of $T = 200$ rounds, based on the AR-4 model above. We take the stochastic rate of change to be $\sigma = 0.1$ for all arms.

**Extension of `AR2` to AR-$p$ processes.** We extend our algorithm `AR2` to handle MAB problems with rewards modeled using general AR-$p$ processes with trends; see Algorithm 2 in Appendix B.2. At a high level, the extension of `AR2` uses similar techniques as described in Section 4. The key difference is that, for each arm $i \in [k]$, it maintains not only an estimate $\widehat{r}_i(t)$ of the arm's reward at round $t$, but also an estimate $\widehat{E}_i(t)$ that captures the associated estimation error. These estimates are dynamically updated based on the structure of the AR-$p$ process with trends. The extended algorithm, akin to Algorithm 1, (i) selects the arm with the highest estimated reward $\widehat{r}_i(t)$ and (ii) triggers arms with potential, where the confidence bound in the triggering criteria now depends on the estimate of the error $\widehat{E}_i(t)$. For a comprehensive discussion of our extension, please refer to Appendix B.2.

**Results.** We compare the performance of `AR2` against the two most competitive benchmarks ($\epsilon$-greedy and a modified UCB algorithm) that we identfied in synthetic experiments; see Appendix A for comparisons with the other benchmarks. The comparison is shown in Table 1. It can be seen that `AR2` stands out in terms of both the regret[6] and the number of times it pulls the optimal arm. In this case study, while the $\epsilon$-greedy algorithm frequently selects the optimal arm, it still accumulates high regret due to its purely random exploration, which can lead to the selection of arms with low rewards. Similarly, the modified UCB algorithm, which leverages the knowledge of the temporal structure in its confidence bound (see description in Appendix C), doesn't perform well. Due to the rapidly changing nature of the AR-$p$ process, modified UCB over-explores. `AR2`, on the other hand, prove to be effective even amidst the rapid changes introduced by more complex time series.

Table 1: Performance comparison.

| Algorithm | normalized regret | # optimal arms |
|---|---|---|
| `AR2` | 0.26 (0.14) | 142.04 (15.71) |
| UCB-mod | 0.60 (0.27) | 106.62 (7.12) |
| $\epsilon$-greedy | 0.38 (0.16) | 133.83 (12.47) |

## 8 Extension on Learning the AR Parameter

In the previous sections, we have assumed full knowledge of the AR parameter, which as discussed in Sections 2 and 7, when we have access to past data. This section proposes an algorithm extension for when the AR parameter needs estimation. We introduce a maximum likelihood method and numerically evaluate the performance of `AR2`, which show that `AR2` still remains competitive despite noise in the estimated AR parameter.

**A Maximum Likelihood Estimator.** If the decision maker does not have prior knowledge of the AR parameter $\alpha$, one can learn the AR parameter via Maximum Likelihood Estimation (MLE). In the first $T_{\text{est}}$ rounds of the time horizon, where $T_{\text{est}} = O(T^\beta)$ for some $\beta \in (0, 1)$, we pull one arm $i \in [k]$ consecutively for a fixed arm $i$ and observe rewards $R_i(1), \ldots, R_i(T_{\text{est}})$. We then define the maximum likelihood estimator $\widehat{\alpha} = \arg\min_{\alpha \in (0,1)} \mathcal{L}(\alpha)$, where $\mathcal{L}(\alpha)$ is the negative of the log-likelihood function defined in (60) of Appendix I. Observe that here we cannot directly apply linear regression for estimating $\alpha$ because the existence of truncating boundaries would lead to a biased estimator. MLE, on the other hand, remains robust even when the expected reward of our arm hits the boundary. Since the number of rounds used for estimation $T_{\text{est}}$ scales sublinearly with $T$, the regret incurred within the first $T_{\text{est}}$ rounds would not impact our per-round regret upper bound. If we have heterogeneous AR parameters $\alpha_i$ for arms $i \in [k]$ (as in Appendix A), one can simply pull each arm $i \in [k]$ consecutively for $T_{\text{est}}$ rounds and perform MLE for each $\alpha_i$.

The following proposition quantifies the amount of noise in the estimated AR parameter.

**Proposition 8.1.** *Let $\widehat{\alpha} = \arg\min_{\alpha \in (0,1)} \mathcal{L}(\alpha)$ be the estimated AR parameter. Then, for $\gamma > 0$, with probability at least $1 - 2/T_{est}^\gamma - 2\exp(-T_{est}V^2/2\mathfrak{R}^2)$, there exists constants $L_1, L_2$ such that*

---

[6]Here we evaluate each algorithm using the *normalized regret*: $\sum_{t=1}^{T} \text{REG}_{\mathcal{A}}(t) / \sum_{t=1}^{T} r^\star(t)$, which normalizes the total regret with the optimal reward in hindsight.

$|\widehat{\alpha} - \alpha| \le (4\sigma\mathfrak{R}L_1)/(VL_2) \cdot \sqrt{2\gamma \log T_{est}/T_{est}}$, *where V is the variance of the observed reward in the steady state distribution, which is independent of our algorithm or the time horizon T.*

We plot the normalized regret incurred by AR2, $\epsilon$-greedy and mod-UCB using the estimated AR parameters, obtained through MLE with $T_{\text{est}} = \{25, 50, 100\}$. We observe that whenever the AR parameters $\alpha_i$ are small or large, the performance of AR2 outcompetes the other two benchmarks. In particular, AR2 appears to be robust to the noises in the estimated AR parameter, with performance close to what we would otherwise obtain with the accurate AR parameters (see Table 2 in Appendix A).

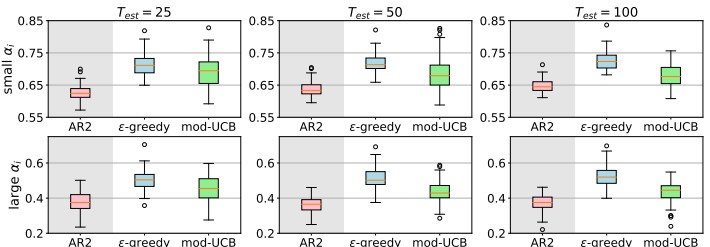

Figure 2: Box plots of normalized regret incurred by each algorithm at $T = 10,000$ (AR2 highlighted in gray). The input AR parameter $\widehat{\alpha}_i$ is the maximum-likelihood estimator of the true AR parameter. We vary the magnitudes of AR parameters $\mathbb{E}[\alpha_i] \in \{0.4, 0.9\}$ and the number of rounds used for estimation $T_{\text{est}} \in \{25, 50, 100\}$. The parameters of our instances are the same as in Appendix A, and we take $k = 6$.

## 9 Conclusion and Future Directions

In this paper, we studied a non-stationary MAB problem with an AR structure, which captures the rapid changes commonly observed in real-world dynamics. Our proposed algorithm, AR2, leverages our knowledge or estimate of the temporal dependency to effectively handle the challenges associated with the fast-changing environment, and our techniques can be potentially adapted to more complex temporal series. As the realm of non-stationary bandits with rewards governed by temporal structures remains largely unexplored, there are several exciting avenues for future research. One intriguing direction is to incorporate seasonality into our framework, exploring models such as seasonal ARIMA with both long-term and short-term changes. Additionally, building on the promising numerical results in Section 8, it would be interesting to theoretically characterize the performance of our algorithm when estimation of the AR parameter and online-decision making are performed simultaneously.

## Acknowledgments

We are grateful to the MIT-IBM Watson AI Lab for their generous support.

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

# Appendices for
# Non-Stationary Bandits with Auto-Regressive Temporal Dependency

---

## A   Numerical Studies on Synthetic Data

**Set-up.** We consider a non-stationary bandit problem with $k$ arms and $T = 10,000$ rounds. To make the setting more general, we let the expected reward $r_i(t)$ of each arm $i \in [k]$ follow an independent AR-1 process with heterogeneous AR parameter $\alpha_i$, stochastic rate of change $\sigma_i$, and truncating boundary $[-1, 1]$. We consider two different regimes of $\alpha_i$'s: (i) *Small $\alpha_i$'s*: the AR-1 parameters $\{\alpha_i\}_{i=1}^k$ are drawn from a dirichlet distribution with rescaling such that for all $i \in [k]$, $\mathbb{E}[\alpha_i] = 0.4$, which is less than the threshold value $\bar{\alpha} = 0.5$ that we defined in Section 5. (ii) *Large $\alpha_i$'s*: the AR-1 parameters $\{\alpha_i\}_{i=1}^k$ are chosen from a dirichlet distribution with rescaling such that $\mathbb{E}[\alpha_i] = 0.9$ for all $i \in [k]$. The concentration parameters of the dirichlet distributions are all set to be 5. For each regime of $\alpha_i$'s (small v.s. large), we consider 100 instances, where each instance corresponds to AR parameters $(\alpha_1, \ldots, \alpha_k)$ generated as described above; and stochastic rates of change $(\sigma_1, \ldots, \sigma_k)$ generated from i.i.d. uniform distributions in $(0, 0.5)$. The initial reward $r_i(0)$ is drawn independently from the steady state distribution discussed in Appendix D.

**Performance comparison.** We implement AR2 and compare it against a number of benchmarks designed for stationary and non-stationary bandits respectively. (See Appendix B.1 for the implementation details, where we generalize AR2 to the setting with heterogeneous $\alpha_i$'s and $\sigma_i$'s.) All parameters used in the following algorithms are tuned.

The benchmark algorithms designed for stationary settings are: (i) "explore-then-commit" (ETC) policy [24] that first selects each arm for $m$ rounds, and commits to the arm with the highest average pay-off in the remaining $T - mk$ rounds; (ii) UCB algorithm [6], which achieves sublinear regret under static settings; (iii) $\epsilon$-greedy algorithm [49] with probability of exploration $\epsilon = 0.1$, which acts greedily with probability $(1 - \epsilon)$ and explores a random arm otherwise. (iv) Exp3 algorithm [7], which is designed for the adversarial setting and achieves sublinear regret when evaluated against the weak static benchmark.

Further, we also consider several benchmark algorithms designed for non-stationary settings: (i) Rexp3 algorithm [10], which modifies the Exp3 algorithm for the non-stationary setting by introducing a restarting mechanism, and assumes that the total variation of expected rewards is bounded by a variation budget $V_T$; (ii) a modified UCB algorithm (mod-UCB), which modifies the upper confidence bound based on the structure of the AR-1 process; see Appendix C for a description and discussion of the algorithm; (iii) the sliding-window UCB (SW-UCB) algorithm [25] and the sliding-window Thompson Sampling (SW-TS) algorithm [52], both of which adopt sliding-window approaches; (iv) the predictive sampling (PS) algorithm for AR-1 bandits recently introduced by [39], which is shown to outperform the Thompson sampling algorithm in non-stationary environments.

**Results.** In Table 2, we evaluate and compare the performance of AR2 against the benchmark algorithms in instances with small $\alpha_i$'s ($\mathbb{E}[\alpha_i] = 0.4$) as well as large $\alpha_i$'s ($\mathbb{E}[\alpha_i] = 0.9$). We also vary the number of arms $k \in \{2, 10, 20\}$. To keep our results consistent across instances, we evaluate each algorithm $\mathcal{A}$ using the *normalized regret*[7]: $\sum_{t=1}^T \text{REG}_{\mathcal{A}}(t) / \sum_{t=1}^T r^\star(t)$, which normalizes the total regret using the reward of optimal in hindsight dynamic policy.

From Table 2, we see that AR2 clearly outperforms the benchmarks in almost all choices of $k$ and regimes of $\alpha_i$'s. The algorithms designed for stationary settings (ETC, UCB, Exp3) perform poorly, as the AR setting includes non-stationarity that differs substantially from the stochastic or adversarial settings they were intended for. Similarly, the algorithms designed for non-stationary setting with limited amount of changes (RExp3, SW-UCB, SW-TS) do not perform well since they are designed to tackle environments either with abrupt changes, or with sublinear amount of changes, and are not suited for the fast changes in the AR environment. Finally, when compared against benchmarks that take into account the AR structure of rewards ($\epsilon$-greedy, modified UCB, PS), AR2 also displays superior performance. Here, the PS algorithm indeed dominates TS-based approaches

---

[7]Note that since the expected rewards can be negative, the normalized regret can exceed one. Further, when $\alpha_i$'s are small, the normalized regret of each algorithm is in general larger. This is because the optimal reward used in normalization tends to be smaller, and hence does not contradict the fact that our problem is more challenging when the AR parameters are small.

Table 2: The mean and standard deviation (in parenthesis) of normalized regret incurred by each algorithm at $T = 10,000$. All experiments are run with 100 instances. The algorithm with the smallest normalized regret is highlighted in gray.

(a) Comparison with Stationary Benchmarks

| $\mathbb{E}[\alpha_i]$ | $k$ | AR2 | ETC | UCB | $\epsilon$-greedy | Exp3 |
|---|---|---|---|---|---|---|
| 0.4 | 2 | 0.38 (0.10) | 1.00 (0.03) | 0.68 (0.09) | 0.43 (0.09) | 0.99 (0.02) |
| | 10 | 0.67 (0.01) | 1.00 (0.01) | 0.80 (0.01) | 0.76 (0.02) | 1.00 (0.01) |
| | 20 | 0.72 (0.01) | 1.00 (0.01) | 0.84 (0.01) | 0.81 (0.01) | 1.00 (0.01) |
| 0.9 | 2 | 0.18 (0.06) | 1.00 (0.11) | 0.65 (0.13) | 0.36 (0.14) | 0.95 (0.12) |
| | 10 | 0.40 (0.04) | 1.00 (0.05) | 0.58 (0.04) | 0.60 (0.06) | 0.99 (0.02) |
| | 20 | 0.49 (0.02) | 1.00 (0.02) | 0.62 (0.02) | 0.64 (0.03) | 0.99 (0.01) |

(b) Comparison with Non-Stationary Benchmarks

| $\mathbb{E}[\alpha_i]$ | $k$ | AR2 | Rexp3 | mod-UCB | SW-UCB | SW-TS | PS |
|---|---|---|---|---|---|---|---|
| 0.4 | 2 | 0.38 (0.10) | 0.96 (0.03) | 0.45 (0.05) | 0.69 (0.05) | 1.00 (0.03) | 0.51 (0.04) |
| | 10 | 0.67 (0.01) | 0.99 (0.01) | 0.67 (0.03) | 0.90 (0.02) | 0.98 (0.01) | 0.78 (0.03) |
| | 20 | 0.72 (0.01) | 1.00 (0.01) | 0.72 (0.02) | 0.95 (0.01) | 0.99 (0.01) | 0.84 (0.02) |
| 0.9 | 2 | 0.18 (0.06) | 0.87 (0.08) | 0.20 (0.07) | 0.55 (0.13) | 0.99 (0.15) | 0.52 (0.33) |
| | 10 | 0.40 (0.04) | 0.97 (0.01) | 0.43 (0.05) | 0.70 (0.03) | 0.86 (0.04) | 0.47 (0.03) |
| | 20 | 0.49 (0.02) | 0.99 (0.01) | 0.53 (0.03) | 0.84 (0.02) | 0.91 (0.02) | 0.52 (0.02) |

in non-stationary environments, as suggested by [39], but AR2 remains competent in all kinds of AR-based setups; further, [39] also suggests that deriving the conditional probability distribution in the predictive sampling procedure and implementing it can be complicated in general, and hence it is unclear how one can adapt the PS algorithm for more complicated AR-$p$ processes as we did in our case study (see Section 7 and Appendix B.2). Among all stationary benchmarks, $\epsilon$-greedy is the most competitive; while mod-UCB stands out as the most competitive within the non-stationary benchmarks. However, there are instances where AR2 can contribute to a decrease in regret as high as $50\%$ when compared to $\epsilon$-greedy, and a $15.5\%$ decrease in regret when compared to modified UCB. All of the aforementioned results attest to the efficacy of AR2 in rapidly changing environments governed by the AR process.

# B   Implementation Details and Extensions of Algorithm AR2

## B.1   Implementation of AR2

To apply AR2 to the more general setting described in Appendix A, where the expected reward of each arm $i$ undergoes an AR-1 process with heterogeneous AR parameter $\alpha_i$ and stochastic rate of change $\sigma_i$, we make a few straightforward updates to AR2: (i) when maintaining estimated reward $\widehat{r}_i(t)$ of arm $i$, we use its AR parameter $\alpha_i$; (ii) we assume that an arm $i \neq i_{\text{sup}}(t)$ gets triggered if at round $t$,

$$\widehat{r}_{\text{sup}}(t) - \widehat{r}_i(t) \leq c_1 \sigma_i \sqrt{(\alpha_i^2 - \alpha_i^{2(t - \tau_i + 1)})/(1 - \alpha_i^2)}.$$

Empirically, we make two more minor changes to AR2 in its numerical implementation. (i) The definition of the superior arm: instead of considering one of the two most recently pulled arms, we consider all arms and choose the one with the highest estimated reward at round $t$ to be the superior arm at round $t$, i.e., $i_{\text{sup}}(t) = \arg\max_{i \in [k]} \widehat{r}_i(t)$. (ii) The triggered arm to be played: in the exploration round $t$ (when $t$ is odd and $\mathcal{T} \neq \emptyset$), we consider the upper confidence bound of the expected reward of each arm $i$, defined as:

$$\text{UCB}_i(t) = \widehat{r}_i(t) + c_1 \sigma_i \sqrt{(\alpha_i^2 - \alpha_i^{2(t - \tau_i(t))})/(1 - \alpha_i^2)}$$

and pull the triggered arm with the highest upper confidence bound, i.e. $I_t = \arg\max_{j \in \mathcal{T}} \text{UCB}_j(t)$. Note that the arms in the triggered set $\mathcal{T}$ differ in terms of both triggering time and estimated rewards. Previously, our criteria for selecting the triggered arm to play only takes into account one aspect of such heterogeneity, i.e., the triggering time. The new criteria allows us to also consider the estimated reward of each arm, which is another indicator of the arm's potential.

## B.2 Extension of AR2 to AR-$p$ Process with Trend

In this section, we provide an extension of AR2 (Algorithm 1), which would apply to non-stationary bandits where the rewards follow general AR-$p$ processes with trends for any $p \geq 1$. This algorithm is applied when we perform our case study on tourism demand prediction in Section 7, and is shown to adapt well to the more complex, rapidly evolving environment.

Let us assume that for each arm $i \in [k]$, the reward distribution following an AR-$p$ process, potentially with a trend component. That is, conditioning on past history during rounds $t - 1, \ldots, 1$, the expected reward of arm $i$ at round $t$ is

$$r_i(t) = \alpha_{i,0} + \sum_{j=1}^{p} \alpha_{i,j} R_i(t - j) \,,$$

where $\alpha_{i,0}$ is the trend component and $(\alpha_{i,1}, \ldots, \alpha_{i,p})$ is the set of AR parameters associated with arm $i$.

The details of the extension of AR2 is presented in Algorithm 2, which we called AR2-p. At a high level, the main techniques used in AR2-p are very similar to those used in AR2. As AR2-p is specifically designed to address practical scenarios where real-world dynamics are represented by more complex time series, we have incorporated the implementation changes discussed in Appendix B.1 to enhance its practical applicability.

At each round $t$ of Algorithm 2, we first determine the superior arm $i_{\mathrm{sup}}(t)$, which is the arm with the highest expected reward. We then decide if there are any arms that we wish to trigger, depending on whether some untriggered arm has an upper confidence bound that exceeds the expected reward of the superior arm $\widehat{r}_{\mathrm{sup}}(t)$. We then alternate between exploration and exploitation: (i) during exploitation rounds, we pull the superior arm; (ii) during exploration rounds, we pull the arm with the highest upper confidence bound out of the set of *triggered arms*, denoted as $\mathcal{T}$. Finally, after we pull arm $I_t$ at each round, we update the estimate for the state of each arm $i \in [k]$.

The main difference between AR2-p and AR2 for AR-1 process lies in how we maintain estimates for each arm. To account for the more complicated structure of an AR-$p$ process with trends, we need to maintain two estimates for each arm—an estimate for the expected reward $\widehat{r}_i(t)$, and an estimate for the error bound $\widehat{E}_i(t)$, defined as follows:

$$\widehat{r}_i(t) = \alpha_{i,0} + \sum_{j=1}^{p} \alpha_{i,j} \widehat{r}_i(t - j) \quad \text{and} \quad \widehat{E}_i(t) = \sum_{j=1}^{p} \alpha_{i,j} \widehat{E}_i(t - j) + \alpha_{i,j} \,.$$

Here, the estimate for error bound $\widehat{E}_i(t)$ essentially captures how much error is contained in our estimate for arm $i$ at round $t$, i.e., $\widehat{r}_i(t)$. Since we estimate $\widehat{r}_i(t)$ based on $p$ past estimates $\widehat{r}_i(t - 1), \ldots, \widehat{r}_i(t - p)$, the error associated with each of these terms, i.e., $\widehat{E}_i(t - 1), \ldots, \widehat{E}_i(t - p)$, would contribute the total error for arm $i$ at round $t$, i.e., $\widehat{E}_i(t)$.

We would then use the error estimate $\widehat{E}_i(t)$ in determining our triggering condition. We trigger an arm $i \notin \mathcal{T} \cup \{i_{\mathrm{sup}}(t)\}$ at round $t$ if

$$\widehat{r}_{\mathrm{sup}}(t) - \widehat{r}_i(t) \leq c \sigma_i \sqrt{\widehat{E}_i(t)} \,.$$

The error bound estimates would also be useful when we determine which arm to pull out of all the triggering arms. To do that, we form the upper confidence bound for each arm

$$\mathtt{UCB}_i(t) = \widehat{r}_i(t) + c \sigma_i \sqrt{\widehat{E}_i(t)}$$

and pick the one with the highest upper confidence bound: $I_t = \arg\max_{i \in \mathcal{T}} \mathtt{UCB}_i(t) \,.$

## C Description of Modified UCB Algorithm

In our numerical studies (Appendix A), we have considered the modified UCB algorithm as one of the non-stationary benchmarks. We now describe this algorithm, which borrows idea from the UCB algorithm [6] for stochastic MAB, and modifies the expression of the upper confidence bound based on the AR temporal structure.

**Algorithm 2** Alternating and Restarting algorithm for non-stationary bandits with AR-$p$ reward structure (AR2-p)

---

**Input:** AR parameters $\{\alpha_{i,j}\}_{\substack{i\in[k]\\j\in[p]}}$, trend components $\{\alpha_{i,0}\}_{i\in[k]}$, stochastic rate of change $\{\sigma_i\}_{i\in[k]}$, epoch size $\Delta_{\text{ep}}$, parameter $c$.

1. Set the epoch index $s = 1$.
2. Repeat while $s \leq \lceil T/\Delta_{\text{ep}} \rceil$:
   (a) Initialization:
      - Set $t_0 = (s-1)\Delta_{\text{ep}}$. Set the initial triggered set $\mathcal{T} = \emptyset$.
      - For each arm $i \in [k]$, set estimates for expected rewards $\widehat{r}_i(t) = 0$ and error bound estimates $\widehat{E}_i(t) = \infty$ for $t \leq 0$.
   (b) For $t = t_0 + 1, \ldots, \min\{t_0 + \Delta_{\text{ep}}, T\}$
      - If $t - t_0 \leq p \cdot k$, pull arm $i = \lfloor (t - t_0 - 1)/p \rfloor + 1$.     *// Pull each arm consecutively for p rounds.*
      - Else,
         - Update the identity of the superior arm and its estimated reward

         $$i_{\text{sup}}(t) = \arg\max_{i \in [k]} \widehat{r}_i(t) \quad \text{and} \quad \widehat{r}_{\text{sup}}(t) = \widehat{r}_{i_{\text{sup}}(t)}(t)$$

         - Trigger arms with potential: For $i \notin \mathcal{T} \cup \{i_{\text{sup}}(t)\}$, trigger arm $i$ if

         $$\widehat{r}_{\text{sup}}(t) - \widehat{r}_i(t) \leq c\sigma_i \sqrt{\widehat{E}_i(t)}.$$

         If triggered, add $i$ to $\mathcal{T}$.
         - Alternate between exploration and exploitation:
            i. If $t$ is odd and $\mathcal{T} \neq \emptyset$, pull a triggered arm with the highest upper confidence bound estimate: $I_t = \arg\max_{i \in \mathcal{T}} \widehat{r}_i(t) + c\sigma_i\sqrt{\widehat{E}_i(t)}$.
            ii. Otherwise, pull the superior arm $I_t = i_{\text{sup}}(t)$.
      - Receive a reward $R_{I_t}(t)$. Update $\widehat{r}_{I_t}(t) = R_{I_t}(t)$ and $\widehat{E}_{I_t}(t) = 0$.
      - Maintain Estimates: For each arm $i \in [k]$, set estimates for expected rewards and error bounds

      $$\widehat{r}_i(t) = \alpha_{i,0} + \sum_{j=1}^{p} \alpha_{i,j}\widehat{r}_i(t-j) \quad \text{and} \quad \widehat{E}_i(t) = \sum_{j=1}^{p} \alpha_{i,j}\widehat{E}_i(t-j) + \alpha_{i,j}.$$

   (c) Set $s = s + 1$.

---

**Algorithm 3** A modified UCB algorithm for non-stationary AR bandits (mod-UCB)

---

**Input:** AR Parameters $(\alpha_1, \ldots, \alpha_k)$, stochastic rates of change $(\sigma_1, \ldots, \sigma_k)$, parameter $\delta$.
1. Initialization: pull each arm $i \in [k]$ at round $i$. Set estimates $\widehat{r}_i(k+1) = \alpha^{k-i} \cdot \mathcal{B}(\alpha R_i(i))$ and $\tau_i = i$.
2. While $t \leq T$:
   (a) For each arm $i \in [k]$, compute its upper confidence bound (UCB):

   $$\text{UCB}_i(t) = \widehat{r}_i(t) + \sqrt{2\log(2/\delta)} \cdot \sigma_i \sqrt{(\alpha_i^2 - \alpha_i^{2(t-\tau_i(t))})/(1-\alpha_i^2)}.$$

   (b) Pull the arm with the highest UCB: $I_t = \arg\max_{i \in [k]} \text{UCB}_i(t)$. Receive a reward $R_{I_t}(t)$, and set $\tau_{I_t} = t$.
   (c) Maintain Estimates: Set $\widehat{r}_{I_t}(t+1) = \mathcal{B}(\alpha R_{I_t}(t))$ and set $\widehat{r}_i(t+1) = \alpha\widehat{r}_i(t)$ for $i \neq I_t$.

---

The details of modified UCB is presented in Algorithm 3. Similar to the original UCB algorithm, we consider an upper confidence bound of the expected reward of each arm $i$ at round $t$, denoted as $\text{UCB}_i(t)$, which consists of two terms. The first term $\widehat{r}_i(t)$ is the estimated reward of arm $i$ based on past observations; the second term is the size of the one-sided confidence interval for the expected reward $r_i(t)$. The confidence interval is constructed based on Hoeffding's Inequality (see Lemma J.3), and is designed such that the true expected reward $r_i(t)$ will fall into the confidence interval with probability at least $1 - \delta$, where $\delta$ is an input parameter. To derive the expression of the upper confidence bound in the AR setting, consider a non-stationary MAB problem, in which the expected reward of arm $i$ follows an AR-1 process with parameter $\alpha_i \in (0, 1)$, stochastic rate of change $\sigma_i \in (0, 1)$, and truncating boundary $[-\mathcal{R}, \mathcal{R}]$. (See Section 2 for details about the set-up.) The estimated reward of arm $i$ at round $t$, $\widehat{r}_i(t)$, can be maintained in the same way as done in Algorithm 1.

The modified UCB algorithm always selects the arm with the highest upper confidence bound. To construct the confidence bound, we first recall from Section 6 that we define a new noise term $\tilde{\epsilon}_i(t)$ (see (5)) that incorporates the influence of the truncating boundary, and the new noise term satisfies the recursive relationship: $r_i(t+1) = \alpha r_i(t) + \alpha \tilde{\epsilon}_i(t)$. Suppose that at round $t$, we last observed the reward of arm $i$ at round $\tau_i(t)$. Using the recursive relationship, we have:

$$r_i(t) = \alpha_i r_i(t-1) + \alpha_i \tilde{\epsilon}_i(t-1) = \alpha_i^{t-\tau_i(t)-1} r_i(\tau_i(t)+1) + \alpha_i^{t-\tau_i(t)-1}\tilde{\epsilon}_i(\tau_i(t)+1) + \cdots + \alpha_i \tilde{\epsilon}_i(t-1)$$

$$= \alpha_i^{t-\tau_i(t)-1} r_i(\tau_i(t)+1) + \sum_{t'=\tau_i(t)+1}^{t-1} \alpha^{t-t'}\tilde{\epsilon}_i(t') = \widehat{r}_i(t) + \sum_{t'=\tau_i(t)+1}^{t-1} \alpha^{t-t'}\tilde{\epsilon}_i(t')\,,$$

(8)

where the last equality follows from the way we maintain estimates. By Lemma H.1, we have that for any $s > 0$,

$$\mathbb{P}\Big[\Big|\sum_{t'=\tau_i(t)+1}^{t-1} \alpha^{t-t'}\tilde{\epsilon}_i(t')\Big| \le s\Big] \ge \mathbb{P}\Big[\Big|\sum_{t'=\tau_i(t)+1}^{t-1} \alpha^{t-t'}\epsilon_i(t')\Big| \le s\Big]\,,$$

(9)

where $\epsilon_i(t') \sim N(0, \sigma_i)$ is the independent Gaussian noise applied to arm $i$ at round $t'$. By Hoeffding's Inequality (Lemma J.3), for fixed $\delta > 0$, we also have the following concentration result:

$$\mathbb{P}\Big[\Big|\sum_{t'=\tau_i(t)+1}^{t-1} \alpha^{t-t'}\epsilon_i(t')\Big| \le \sqrt{2\log(2/\delta)} \cdot \sigma_i \sqrt{(\alpha_i^2 - \alpha_i^{2(t-\tau_i(t))})/(1-\alpha_i^2)}\Big] \ge 1-\delta\,. \quad (10)$$

Equations (8), (9) and (10) together establish a confidence bound centered around $\widehat{r}_i(t)$, such that the true expected reward $r_i(t)$ falls into the confidence bound with probability at least $1 - \delta$. In Algorithm 3, we thus define the upper confidence bound of arm $i$ at round $t$ as:

$$\texttt{UCB}_i(t) = \widehat{r}_i(t) + \sqrt{2\log(2/\delta)} \cdot \sigma_i \sqrt{(\alpha_i^2 - \alpha_i^{2(t-\tau_i(t))})/(1-\alpha_i^2)}\,,$$

and we have $\mathbb{P}[r_i(t) > \texttt{UCB}_i(t)] \le 1 - \delta$ when implementing the modified UCB algorithm for numerical experiments in Appendix A.

Finally, we remark that our algorithm `AR2` and the modified UCB algorithm share some similarities. Both algorithms incorporate our knowledge of the AR temporal structure into their design, which makes them more competitive against the rest of the benchmarks. The triggering criteria of `AR2` essentially constructs a confidence interval around the estimated reward $\widehat{r}_i(t)$ of arm $i$ at round $t$, and triggers arm $i$ if the upper confidence bound of arm $i$ exceeds the estimated reward of the superior arm. On the other hand, `AR2` also differs from the modified UCB in two main aspects. (i) By alternating between exploration and exploitation rounds, `AR2` avoids potential over-exploration. This is especially common when $\alpha_i$'s are close to one, and any arm that has not been explored for a few rounds would have a very high confidence bound. In that case, modified UCB is likely to shift in between arms very often, which leads to too little exploitation. (ii) As discussed in Section 4, unlike the modified UCB algorithm, `AR2` also deals with the tradeoff of "remembering" and "forgetting" via the restarting mechanism, which is especially important in a fast changing environment such as the AR setting.

## D Steady State Distribution of Rewards

In this section, we characterize the steady state distribution of the expected reward $r_i(t)$. Lemma D.1 shows that the steady state distribution depends on two probability masses at the boundaries and a probability density function between the boundaries.

**Lemma D.1** (Probability density of steady state distribution)**.** *Consider the steady state distribution of $r_i(t)$, which follows the AR process with truncating boundaries defined in Section 2. Suppose that at the steady state distribution, we have two probability masses at the boundaries, i.e., $\mathbb{P}[r_i(t) = \mathcal{R}] = \mathbb{P}[r_i(t) = -\mathcal{R}] = p$, where $p \in (0, 1/2)$ depends on $\alpha$ and $\sigma$. Then, the steady state distribution between the boundaries can be characterized by the following probability density function (PDF), denoted by $f_p : (-\mathcal{R}, \mathcal{R}) \to \mathbb{R}^+$.*

$$f_p(x) = (1-2p)C(\alpha, \sigma) \exp\Big(\frac{(\alpha-1)x^2}{\alpha^2 \sigma^2}\Big) \quad x \in (-\mathcal{R}, \mathcal{R}), \quad (11)$$

*where $C(\alpha, \sigma) = \dfrac{\sqrt{1-\alpha}}{\sqrt{\pi}\alpha\sigma\,\mathrm{erf}\Big(\frac{\mathcal{R}\sqrt{1-\alpha}}{\alpha\sigma}\Big)}$ is a normalization factor and $\mathrm{erf}(z) = \frac{2}{\sqrt{\pi}}\int_0^z e^{-s^2} ds$ is the*

*error function.*

Given Lemma D.1, we are now ready to define the steady state distribution of $r_i(t)$:

**Definition D.2** (Steady state distribution of rewards). *Let $r$ be a random variable with range $[-\mathcal{R}, \mathcal{R}]$ whose probability distribution characterizes the steady state distribution of $r_i(t)$. It has probability masses on the boundaries, i.e., $\mathbb{P}[r = \mathcal{R}] = \mathbb{P}[r = -\mathcal{R}] = p$, and probability density function $f_p$ on $(-\mathcal{R}, \mathcal{R})$ as defined in* (11), *that satisfies the following conditions:*

1. $\mathbb{P}\left[\alpha(r + \epsilon) \geq \mathcal{R}\right] = \mathbb{P}\left[\alpha(r + \epsilon) \leq -\mathcal{R}\right] = p.$
2. $\mathbb{P}\left[\alpha(r + \epsilon) < y\right] = p + \int_{-\mathcal{R}}^{y} f_p(x)dx \quad$ *for all $y \in (-\mathcal{R}, \mathcal{R})$.*

Note that the probability masses on the boundaries result from the truncating boundary condition, and by symmetry, the masses on upper and lower boundaries are the same. If one adopts a reflecting boundary condition, we would have $p = 0$; if one adopts an absorbing boundary condition, we would have $p = 1/2$ (see [44] for definitions of the aforementioned boundary conditions).

We remark that for any fixed $\sigma \in (0, 1)$, when $\alpha \to 1$, the probability density function $f_p(x)$ in the steady state distribution of $r_i(t)$ can be well approximated by a constant function. In Figure 3, we plot the PDF of the steady state distributions for $\alpha \in \{0.3, 0.6, 0.9\}$ (with $\sigma = 0.8$ and $\mathcal{R} = 1$), and numerically compute the value of $p$ (see legends in Figure 3)[8]. It can be seen that for larger $\alpha$, the distribution within the boundaries becomes almost "uniform". In the other extreme case, when $\alpha \to 0$, it can be seen both from Figure 3 and from the definition of $f_p(x)$ in Lemma D.1 that the distribution can be approximated by a normal distribution.

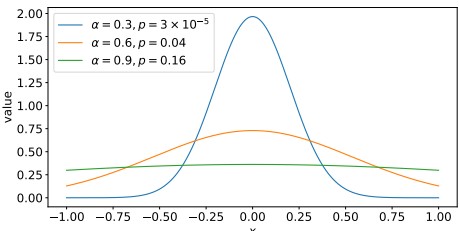

Figure 3: PDF of the steady state distribution of AR process with truncating boundaries for $\alpha \in \{0.3, 0.6, 0.9\}, \sigma = 0.8, \mathcal{R} = 1$. The probability masses $p$ associated with the steady state distribution are included in the legends.

**Proof of Lemma D.1:** We first note that the continuous-time analogue of an AR-1 process is a Ornstein-Uhlenbeck (OU) process [22], defined as the following stochastic differential equation (SDE):

$$dr_t = \kappa\left(\theta - r_t\right)dt + \zeta dW_t, \tag{12}$$

where $\kappa, \theta, \zeta$ are constant parameters[9] and $W_t$ denotes the Wiener process[10]. If we perform Euler-Maryuama discretization (see [34]) on the above OU process at $\Delta t, 2\Delta t, \dots$, we get

$$r_{t+1} = \kappa\theta\Delta t - (\kappa\Delta t - 1)r_t + \zeta\epsilon_t'\sqrt{\Delta t}, \tag{13}$$

where $\epsilon_t'$ is a standard normal noise. Now, recall from Section 2 that the evolution of expected rewards evolve as an independent AR-1 process:

$$r_i(t + 1) = \alpha R_i(t) = \alpha\left(r_i(t) + \epsilon_i(t)\right), \tag{14}$$

where $\epsilon_i(t) \sim N(0, \sigma)$. Let $r_t \triangleq r_i(t)$. By coefficients in (13) and (14), the two equations are equivalent if we let $\kappa = 1 - \alpha, \theta = 0, \zeta = \alpha\sigma, \Delta t = 1$. By plugging the values of coefficients into (12), the continuous-time analogue of (14) can be described by the following SDE:

$$dr_t = (\alpha - 1)r_tdt + \alpha\sigma dW_t.$$

Let $f_p(x)$ be the PDF for the steady state distribution of $r_i(t)$ on $(\mathcal{R}, \mathcal{R})$, given that the probability mass on either side of the boundary is $p$. We can now solve for $f_p(x)$ by solving the following

---

[8]For numerically simulating steady state distributions of SDEs, see [34].

[9]We will see that when $r_t$ follows an AR-1 process with AR parameter $\alpha$ and stochastic rate of change $\sigma$, these constants take the following values: $\kappa = 1 - \alpha, \theta = 0, \eta = \alpha\sigma$.

[10]A Wiener process $W_t$ is a continuous-time stochastic process that satisfies the following properties: (i) $W_0 = 0$; (ii) $W_t$ is continuous in $t$; (iii) $\{W_t\}_{t \geq 0}$ has stationary, independent increments; (iv) $W_{t+s} - W_s \sim N(0, t)$. (See [20] for more details.)

ordinary differential equation (see [23, 44] for a discussion of forward and backward equations for SDEs):

$$(1 - \alpha)(f_p(x) + x f_p'(x)) + \frac{1}{2}\alpha^2 \sigma^2 f_p''(x) = 0 \,, \tag{15}$$

which gives

$$f_p(x) = \tilde{C}(\alpha, \sigma) \exp(\frac{(\alpha - 1)x^2}{\alpha^2 \sigma^2}),$$

for $\tilde{C}$ that depends on $\alpha, \sigma$. Since $f_p(x)$ is a probability density, we can solve for $\tilde{C}(\alpha, \sigma)$ by requiring $\int_{-\mathcal{R}}^{\mathcal{R}} f_p(x) \, dx = 1 - 2p$, given that the total probability within the boundaries is $1 - 2p$. This then gives

$$\tilde{C}(\alpha, \sigma) = (1 - 2p)\sqrt{1 - \alpha}/(\sqrt{\pi}\alpha\sigma \, \mathrm{erf}\left(\frac{\mathcal{R}\sqrt{1 - \alpha}}{\alpha\sigma}\right)) = (1 - 2p)C(\alpha, \sigma) \,,$$

where $C(\alpha, \sigma)$ is defined in the statement of Lemma D.1. ∎

# E   Additional Illustrations

## E.1   Illustration of $g(k, \alpha, \sigma)$ in Regret Lower Bound

Recall that in Section 3, we characterize our regret lower bound to be $\Omega(g(k, \alpha, \sigma)\alpha\sigma)$, where $g(k, \alpha, \sigma)$ is the probability that two best arms are within $\alpha\sigma$ distance of each other at any given round under the steady state distribution, with its closed form presented in Equation (16) in Appendix F.

To better understand how $g(k, \alpha, \sigma)$ evolves in terms of our parameters, in Figure 4 we plot the evolution of $g(k, \alpha, \sigma)$ in terms of $\sigma$ in two different regimes: (i) a small-$\alpha$ regime when $\alpha = 0.4$, and (ii) a large-$\alpha$ regime when $\alpha = 0.9$. Note that here, we consider the two regimes separately because in Section 5, we will show that the large-$\alpha$ regime is inherently much more challenging than the small-$\alpha$ regime.

In Figure 4, we see that in the small-$\alpha$ regime ($\alpha = 0.4$), $g(k, \alpha, \sigma)$ is roughly constant as $\sigma$ increases, while in the large-$\alpha$ regime ($\alpha = 0.9$), $g(k, \alpha, \sigma)$ is roughly linear in terms of $\sigma$. To understand why, recall from our discussion in Appendix D that when $\alpha \to 0$, the steady state distribution approaches a normal distribution with standard deviation proportional to $\alpha\sigma$, while when $\alpha \to 1$, the steady state distribution resembles a uniform distribution. Hence, when $\alpha$ is small, we expect $g(k, \alpha, \sigma)$ to be roughly constant in terms of $\sigma$, while when $\alpha$ is close to one, we expect $g(k, \alpha, \sigma)$ to be roughly linearly in terms of $\sigma$. Here, our observations in Figure 4 concur with our discussion in Appendix D.

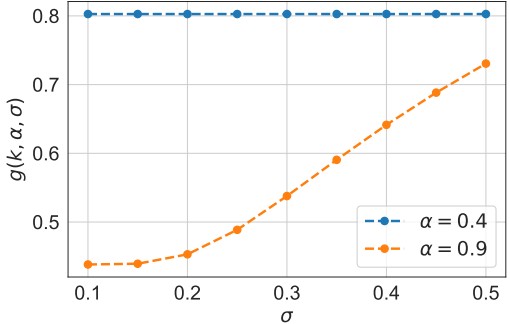

Figure 4: The evolution of function $g(k, \alpha, \sigma)$, as defined in Equation (16) in Appendix F, with respect to $\sigma$. Here, we set $\alpha \in \{0.4, 0.9\}$, $k = 5$ and $\mathcal{R} = 1$.

## E.2   Illustration of Upper/Lower Bounds in the Small-$\alpha$ Regime

In Figure 5, we illustrate the evolution of our upper and lower bounds in the small-$\alpha$ regime (i.e., $\alpha \in (0, \bar{\alpha})$). Here, we take $\alpha = 0.4$, and similar plots can be obtained for different values of $\alpha < \bar{\alpha}$. Recall from Section 5 that this regime is considered less challenging due to our result in Theorem 5.1, which states that any algorithm can attain near-optimal performance when $\alpha \in (0, \bar{\alpha})$. Here, Figure 5 reveals that our upper/lower bounds indeed have the same trend of increase in terms of $\sigma$ in the small-$\alpha$ regime, which complements our result in Figure 1 (that shows our upper/lower bounds match in terms of $\alpha$ in the small-$\alpha$ regime).

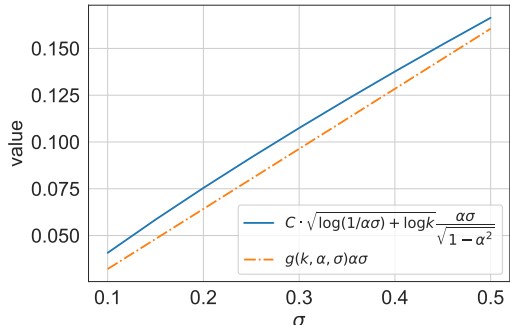

Figure 5: Comparison of the order of the regret upper bound in Theorem 5.1 and the order of the lower bound in Theorem 3.1, with respect to $\sigma$, obtained with $k = 5, \alpha = 0.4, \mathcal{R} = 1, C = 0.4$. This plot complements Figure 1 that shows dependency on $\alpha$.

### E.3 Illustrations of Upper/Lower Bounds and Per-Round Regrets of `AR2`

To further highlight the significance of Theorem 5.2 and how well it characterizes the performance of `AR2`, in Figure 6, we plot the evolution of the upper bound in Theorem 5.2, the lower bound $O(k\alpha^2\sigma^2)$ for $\alpha$ close to one, and the per-round regret attained by `AR2`, at different values of parameters $\alpha$ and $\sigma$ respectively. In Figure 6a, the per-round regret of `AR2` increases at roughly the same rate as the upper and lower bounds, and stays close to the lower bound, as $\alpha$ increases. In Figure 6b, the upper and lower bounds follow the same trend of increase with respect to $\sigma$. We note that as $\sigma$ increases, the upper bound loosens up as the per-round regret of `AR2` experiences smaller growth and approaches the worst-case lower bound. This especially highlights the strength of `AR2` in adapting to a rapidly changing environment.

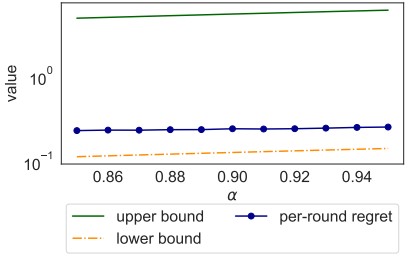
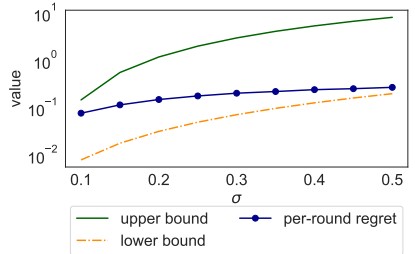

(a) Evolution of the upper/lower bounds and per-round regret of `AR2` with varying $\alpha$. We take $\alpha \in [0.85, 0.95]$ and $\sigma = 0.4$.

(b) Evolution of the upper/lower bounds and per-round regret of `AR2` with varying $\sigma$. We take $\sigma \in [0.1, 0.5]$ and $\alpha = 0.9$.

Figure 6: Comparison of the order of regret upper bound of `AR2` (Theorem 5.2), the order of the lower bound, and per-round regret attained by `AR2` (averaged across 100 experiments). The constants of upper/lower bounds are tuned for illustration purposes. Here, $k = 5$ and $\mathcal{R} = 1$.

## F  Missing Proof of Section 3

Before we show the proof for Theorem 3.1, we first present Lemma F.1 [17] which characterizes the joint distribution for order statistics.

**Lemma F.1** (The joint distribution of order statistics of a continuous distribution). *Let $X^{(1)} \leq X^{(2)} \leq \cdots \leq X^{(k)}$ be the order statistics of $k$ i.i.d. samples drawn from some continuous distribution with probability density function (PDF) $f(x)$ and cumulative density function (CDF) $F(x)$. The joint PDF of $X^{(j)}, X^{(j)}$ for any $1 \leq i < j \leq k$ is*

$$f_{X^{(i)}, X^{(j)}}(u, v) = \frac{k! F(u)^{i-1}(F(v) - F(u))^{j-i-1}(1 - F(v))^{k-j} f(u) f(v)}{(i-1)!(j-i-1)!(k-j)!}$$

In particular, if we take $i = k - 1$ and $j = k$, Lemma F.1 suggests that

$$f_{X^{(k-1)}, X^{(k)}}(u, v) = k(k-1)F(u)^{k-2} f(u) f(v),$$

which will be used in the following proof of Theorem 3.1.

**Proof of Theorem 3.1:** The proof proceeds in two steps. First, we compute the probability that the expected rewards of two best arms are close to each other at any given round $t$. Then, conditioning on this event, we show that at the next round $t+1$, any algorithm incurs $\Omega(\alpha\sigma)$ regret with constant probability.

**Step 1.** We first lower bound the probability that the expected rewards of the two best arms are within $\alpha\sigma$ of each other at round $t$. Recall from Appendix D that $r_i(t)$ follows the steady state distribution with probability mass $p$ on the boundary and PDF $f_p(x)$ defined in (11) within the boundaries. Let $F(x)$ be the CDF of the the steady state distribution of $r_i(t)$. We can compute the probability that the expected rewards of the two best arms are within $\alpha\sigma$ by breaking the event into the following cases:

1. *Both arms are at the upper boundary.* This will happen with probability $\sum_{j=2}^{k} \binom{k}{j} p^j (1 - p)^{k-j}$, where $j$ denote the number of arms are at the upper boundary, i.e., have expected reward $\mathcal{R}$.

2. *The best arm is at the upper boundary, and the second best arm is within the boundaries.* This will happen with the following probability: $k(k-1) \cdot p \cdot \int_{\mathcal{R}-\alpha\sigma}^{\mathcal{R}} F(x)^{k-2} f_p(x) dx$

3. *Both the best arm and the second best arm are between the boundaries.* This will happen with the following probability: $k(k-1) \int_0^{\alpha\sigma} \int_{-\mathcal{R}}^{\mathcal{R}} F(x)^{k-2} f_p(x) f_p(x+z) dx dz$. To obtain this probability, we used the joint PDF stated in Lemma F.1.

4. *The best arm is within the boundaries, and the second best arm is at the lower boundary.* This will happen with the following probability $k \cdot \int_{-\mathcal{R}}^{\alpha\sigma-\mathcal{R}} f_p(x) dx \cdot p^{k-1}$.

5. *Both arms are at the lower boundary.* This will happen with probability $p^k$.

We define $g(k, \alpha, \sigma)$ as the probability that the expected rewards of the two best arms are within $\alpha\sigma$ of each other at round $t$, which is the sum of the terms above:

$$g(k, \alpha, \sigma) = \sum_{j=2}^{k} \binom{k}{j} p^j (1 - p)^{k-j} + k(k-1) \cdot p \cdot \int_{\mathcal{R}-\alpha\sigma}^{\mathcal{R}} F(x)^{k-2} f_p(x) dx$$

$$+ k(k-1) \int_0^{\alpha\sigma} \int_{-\mathcal{R}}^{\mathcal{R}} F(x)^{k-2} f_p(x) f_p(x+z) dx dz + k \cdot \int_{-\mathcal{R}}^{\alpha\sigma-\mathcal{R}} f_p(x) dx \cdot p^{k-1} + p^k$$

$$(16)$$

We now make a brief remark about the order of $g(k, \alpha, \sigma)$ as $\alpha$ approaches one. First note that among the aforementioned cases, the first three cases would happen with probability at least $1/2$ given that the steady state distribution of expected reward is symmetric and the expected rewards of the two best arms are skewed towards the upper boundary. Conditioning on the third case, i.e., both the best and the second best arms are within the boundaries, as $\alpha$ approaches one, the conditional PDF can be approximated as a uniform function (see Remark G.1 in Appendix G for a related discussion), and hence the conditional probability that two best arms are within $\alpha\sigma$ of each other are of order $\Omega(k\alpha\sigma)$. Further, conditioning on the first and the second cases, i.e., the best arm is at the upper boundary, the conditional probability that two best arms are within $\alpha\sigma$ of each other are lower bounded by $\Omega(k\alpha\sigma)$. We thus have $g(k, \alpha, \sigma) \geq \Omega(k\alpha\sigma)$ as $\alpha$ approaches one.

**Step 2.** Let $\mathcal{A}$ be any algorithm that selects arm $\mathcal{A}(t)$ at round $t$. In this step, we show that conditioning on the event that the two best arms are within $\alpha\sigma$ of each other at round $t$, with constant probability, $\mathcal{A}$ will make a mistake at round $t+1$ and incur regret of order $O(\alpha\sigma)$.

Without loss of generality, let us suppose $r_1(t) \geq \cdots \geq r_k(t)$, and that algorithm $\mathcal{A}(t)$ selects arm 1 at round $t$ (if $\mathcal{A}(t)$ selects a different arm at round $t$, the following proof works similarly). We first assume that $r_1(t) \in [-\mathcal{R} + 3\sigma, \mathcal{R} - 3\sigma]$. Let $\Phi(x)$ be the CDF of standard normal distribution. Since $\epsilon_i(t) \overset{i.i.d.}{\sim} N(0, \sigma)$, when $\mathcal{A}$ pulls arm 1 at round $t$, with constant probability $2\Phi(\frac{1}{2}) - 1$, we have $|\epsilon_1(t)| \leq \frac{\sigma}{2}$. Let $\mathcal{A}(t+1)$ be the arm that $\mathcal{A}$ chooses to pull at round $t+1$. At round $t+1$, one of the following scenarios will take place:

(i) If $\mathcal{A}(t+1) = 1$, then if $\epsilon_2(t) \in [2\sigma, 3\sigma]$, it will incur regret

$$
\begin{aligned}
r_2(t+1) - r_1(t+1) &= \alpha\left(r_2(t) + \epsilon_2(t) - r_1(t) - \epsilon_1(t)\right) \\
&\geq \alpha(\epsilon_2(t) - |r_2(t) - r_1(t)| - |\epsilon_1(t)|) \\
&\geq \alpha(2\sigma - \alpha\sigma - \frac{\sigma}{2}) \\
&\geq \frac{\alpha\sigma}{2}.
\end{aligned}
$$

The event $\epsilon_2(t) \in [2\sigma, 3\sigma]$ happens with constant probability $\Phi(3) - \Phi(2)$.

(ii) If $\mathcal{A}(t+1) = j$ for any $j \neq 1$ at $t+1$, then if $\epsilon_j(t) \in [-2\sigma, -\sigma]$, it will incur regret

$$
\begin{aligned}
r_1(t+1) - r_j(t+1) &= \alpha\left(r_1(t) + \epsilon_1(t) - r_j(t) - \epsilon_j(t)\right) \\
&\geq \alpha(\epsilon_1(t) - \epsilon_j(t)) \\
&\geq \alpha(-\frac{\sigma}{2} + \sigma) \\
&= \frac{\alpha\sigma}{2}.
\end{aligned}
$$

The event $\epsilon_j(t) \in [-2\sigma, -\sigma]$ happens with constant probability $\Phi(2) - \Phi(1)$.

In all of the above cases, we show that $\mathcal{A}$ will incur regret at least $\frac{\alpha\sigma}{2}$ with constant probability, conditioning on the event that $r^{(1)} - r^{(2)} \leq \alpha\sigma$. If $r_1(t)$ is close to either side of the boundary, a similar analysis can establish the same result. To see that, if $r_1 \in [\mathcal{R} - 3\sigma, \mathcal{R}]$ and $\mathcal{A}$ pulls arm 1 at round $t$, with constant probability we have $\epsilon_1(t) \in [-2\sigma, -3\sigma]$. Then, we have one of the following: (i) $\mathcal{A}(t+1) = 1$. If $|\epsilon_2(t)| \leq \frac{\sigma}{2}$, arm 1 is sub-optimal and incurs regret by at least $\frac{\alpha\sigma}{2}$; (ii) $\mathcal{A}(t+1) = j$ for some $j \neq 1$. If $\epsilon_j(t) \in [-3\sigma, -4\sigma]$, arm $j$ is sub-optimal and incurs regret by at least $\alpha\sigma$. This, again, shows that $\mathcal{A}$ incurs regret at least $\Omega(\alpha\sigma)$ at round $t+1$ with constant probability. The proof follows similarly when $r_1 \in [-\mathcal{R}, -\mathcal{R} + 3\sigma]$.

Combining step 1 and 2, we conclude that the per-round regret of any algorithm $\mathcal{A}$ is at least $\Omega(g(k, \alpha, \sigma)\alpha\sigma)$, which concludes the proof. $\blacksquare$

## G  Missing Proof of Section 5

**Proof of Theorem 5.1:**  At any given round $t$, recall from Appendix D that $r_i(t)$ follows a steady state distribution as defined in Definition D.2. Let us define $r_i'(t) \overset{i.i.d.}{\sim} N(0, \frac{\alpha\sigma}{\sqrt{1-\alpha^2}})$. The variable $r_i'(t)$ has the following PDF:

$$
f'(x) = \frac{\sqrt{1-\alpha^2}}{\sqrt{2\pi}\alpha\sigma} \exp\left(\frac{(\alpha^2-1)x^2}{2\alpha^2\sigma^2}\right)
$$

for all $x \in \mathbb{R}$. Note that for all $0 < s \leq \mathcal{R}$, we have that $\mathbb{P}\left[|r_i(t)| > s\right] \leq \mathbb{P}\left[|r_i'(t)| > s\right]$, which can be obtained by directly computing both probabilities. Intuitively, one can think of the normal distribution above as the steady state distribution of $r_i(t)$ that follows the same AR-1 process $r_i(t+1) = \alpha(r_i(t) + \epsilon_i(t))$, without the truncating boundaries at $[-\mathcal{R}, \mathcal{R}]$. As a result of the truncating boundaries, the steady state distribution $\mathcal{D}$ (characterized in Appendix D) is more "concentrated" than the normal distribution $N(0, \frac{\alpha\sigma}{\sqrt{1-\alpha^2}})$. Now, using the concentration bound for normal random variable (see Lemma J.2), we have

$$
\mathbb{P}\left[|r_i(t)| > c\frac{\alpha\sigma}{\sqrt{1-\alpha^2}}\right] \leq \mathbb{P}\left[|r_i'(t)| > c\frac{\alpha\sigma}{\sqrt{1-\alpha^2}}\right] \leq 2\exp(-\frac{c^2}{2})
$$

for every arm $i \in [k]$. Now let $\delta = 2\exp(-\frac{c^2}{2})$. By the union bound, with probability at least $1 - k\delta$, we would have $|r_i(t)| \leq c\frac{\alpha\sigma}{\sqrt{1-\alpha^2}}$ for all $i \in [k]$. And hence $\Delta r_i(t) \leq 2c\frac{\alpha\sigma}{\sqrt{1-\alpha^2}}$ for all $i \in [k]$. Let us denote the above event as $\mathcal{E}$. If we set $c = \sqrt{2\log(1/\alpha\sigma) + 2\log(2k)}$, we would have $\delta = \frac{\alpha\sigma}{k}$, and hence $\mathbb{P}[\mathcal{E}] \geq 1 - k\delta = 1 - \alpha\sigma$.

Let $\mathcal{A}$ be any non-anticipating algorithm that selects arm $\mathcal{A}(t)$ at round $t$. The expected regret $\mathcal{A}$ incurs at round $t$ is

$$\mathbb{E}[\Delta r_{\mathcal{A}(t)}(t)] = \mathbb{E}[\Delta r_{\mathcal{A}(t)}(t)|\mathcal{E}]\mathbb{P}[\mathcal{E}] + \mathbb{E}[\Delta r_{\mathcal{A}(t)}(t)|\mathcal{E}^c]\mathbb{P}[\mathcal{E}^c] \leq 2c\frac{\alpha\sigma}{\sqrt{1-\alpha^2}} + 2\mathcal{R} \cdot \alpha\sigma$$

$$= O\left(c\frac{\alpha\sigma}{\sqrt{1-\alpha^2}}\right) = O\left(\sqrt{\log(1/\alpha\sigma) + \log k}\frac{\alpha\sigma}{\sqrt{1-\alpha^2}}\right)$$

Given that the per-round regret is also bounded by $2\mathcal{R}$, this naturally gives

$$\frac{1}{T}\mathbb{E}\left[\sum_{t=1}^{T}\Delta r_{\mathcal{A}(t)}(t)\right] = O\left(\min(\sqrt{\log(1/\alpha\sigma) + \log k}\frac{\alpha\sigma}{\sqrt{1-\alpha^2}}, 2\mathcal{R})\right). \quad \blacksquare$$

In Section 5, we choose $\bar{\alpha} \triangleq 0.5$ as the threshold value such that when $\alpha \in [\bar{\alpha}, 1)$, the problem is more challenging and thereby of our interest. Given the threshold $\bar{\alpha}$, we further expand on the remark made in Appendix D, which will be useful in the proof of Lemma 6.4 in Appendix H.

**Remark G.1** (Uniform Approximation). For any fixed $\sigma \in (0, 1)$, when $\alpha \to 1$, the probability density function that characterizes steady state distribution of $r_i(t)$ within the boundaries can be well approximated by a constant function. This can be seen from

$$\max_{x \in (-\mathcal{R}, \mathcal{R})} f_p(x) - \min_{x \in (-\mathcal{R}, \mathcal{R})} f_p(x) = (1 - 2p)\left[1 - \exp\left(\frac{(\alpha - 1)\mathcal{R}^2}{\alpha^2\sigma^2}\right)\right]C(\alpha, \sigma) \xrightarrow[\alpha \to 1]{} 0.$$

In particular, by Lemma D.1, for any $\alpha \in [\bar{\alpha}, 1)$ and $\sigma \geq c'\sqrt{1-\alpha}$, we can bound the PDF $f_p(x)$ that characterizes the steady state distribution of $r_i(t)$ for all $x \in [-\mathcal{R}, \mathcal{R}]$ using constants:

$$\rho_- \leq f_p(x) \leq \rho_+, \tag{17}$$

where $\rho_- = (1 - 2p)C(\bar{\alpha}, c'\sqrt{1-\bar{\alpha}})\exp\left(-\mathcal{R}^2/\bar{\alpha}^2 c'^2\right), \rho_+ = (1 - 2p)C(\bar{\alpha}, c'\sqrt{1-\bar{\alpha}})$ are both constants.

When $\alpha \in [\bar{\alpha}, 1)$ and $\sigma = o(\sqrt{1-\alpha})$, the steady state distribution of $r_i(t)$ can be well approximated by a normal distribution, and similar to what we show in Theorem 5.1, in this case even the naive algorithm performs well. We thus focus on the more challenging regime when $\alpha \in [\bar{\alpha}, 1)$ and $\sigma = \Omega(\sqrt{1-\alpha})$ in the following analysis.

# H  Missing Proofs of Section 6

## H.1  Step 2: Good event and its implications.

In Section 6, we define a new noise term in (5) that shows the influence of truncating boundary:

$$\tilde{\epsilon}_i(t) = \begin{cases} \epsilon_i(t) & \text{if } \alpha(r_i(t) + \epsilon_i(t)) \in [-\mathcal{R}, \mathcal{R}] \\ \frac{1}{\alpha}[\mathcal{B}(\alpha(r_i(t) + \epsilon_i(t))) - \alpha r_i(t)] & \text{otherwise} \end{cases}$$

The new noise $\tilde{\epsilon}_i(t)$ satisfies the recursive relationship: $r_i(t+1) = \alpha r_i(t) + \alpha\tilde{\epsilon}_i(t)$. Recall that here, the boundary function is defined as $\mathcal{B}(y) = \min\{\max\{y, -\mathcal{R}\}, \mathcal{R}\}$.

Before proceeding with the proof of Lemma 6.3, we first show an auxiliary lemma that establishes the relationship between $\tilde{\epsilon}_i(t)$ and $\epsilon_i(t)$. In particular, Lemma H.1 shows that if the weighted sum of $\epsilon_i(t)$ is small with high probability, the weighted sum of $\tilde{\epsilon}_i(t)$ is also small with high probability.

**Lemma H.1.** *For any $s > 0$, $i \in [k]$, and rounds $t_0 < t_1$, we have*

$$\mathbb{P}\left[\left|\sum_{t=t_0}^{t_1-1}\alpha^{t_1-t}\tilde{\epsilon}_i(t)\right| \leq s\right] \geq \mathbb{P}\left[\left|\sum_{t=t_0}^{t_1-1}\alpha^{t_1-t}\epsilon_i(t)\right| \leq s\right]. \tag{18}$$

**Proof of Lemma H.1:** For simplicity of notation, let us drop the notation for dependence on $i$. We first show that the following inequality holds for any $r(t_0)$:

$$\mathbb{P}\Big[\Big|\sum_{t=t_0}^{t_1-1}\alpha^{t_1-t}\tilde{\epsilon}(t)\Big|\le s\,|\,r(t_0)\Big]\ge \mathbb{P}\Big[\Big|\sum_{t=t_0}^{t_1-1}\alpha^{t_1-t}\epsilon(t)\Big|\le s\,|\,r(t_0)\Big]. \tag{19}$$

To show (19), we first consider a fictional scenario in which $r(.)$ only gets truncated at the first time it hits the boundary, and no longer gets truncated afterwards. Let $t'$ denote the first round at which $r(.)$ hits the boundary. With slight abuse of notation, we define corresponding fictional noises $\epsilon_1(t_0),\dots,\epsilon_1(t_1-1)$ such that $\epsilon_1(t)=\epsilon(t)$ for all $t\ne t'$ and $\epsilon_1(t')=\tilde{\epsilon}(t')$.

Since the expected reward $r(.)$ hits the boundary for the first time at $t'$, we have $|\alpha r(t')+\alpha\epsilon(t')|>\mathcal{R}$ for some $t_0\le t'<t_1$. Recall that in the fictional scenario, $\epsilon_1(t)=\epsilon(t)$ for all $t<t'$. Let $L=\alpha^{t'-t_0+1}r(t_0)$, $E_1=\sum_{t=t_0}^{t'-1}\alpha^{t'-t+1}\epsilon(t)$. Without loss of generality, let us assume that the expected reward hits the upper boundary $\mathcal{R}$, i.e.

$$\mathcal{R}<\alpha r(t')+\alpha\epsilon(t')=L+\sum_{t=t_0}^{t'}\alpha^{t'-t+1}\epsilon(t)=L+E_1+\alpha\epsilon(t').$$

Let $d=L+E_1+\alpha\epsilon(t')-\mathcal{R}>0$ be the distance by which the expected reward exceeds the boundary at $t'$. By (5) and the definition of the truncating boundary in Section 2, we have

$$\alpha r(t')+\alpha\epsilon_1(t')=\mathcal{B}(\alpha r(t')+\alpha\epsilon(t'))=\mathcal{R}.$$

From the discussion above, we have $E_1+\alpha\epsilon(t')=\mathcal{R}+d-L$ and $E_1+\alpha\epsilon_1(t')=\mathcal{R}-L$. Since in the fictional scenario, the expected reward only gets reflected at the first time of hitting the boundary, we also have $\epsilon_1(t)=\epsilon(t)$ for all $t>t'$. Let $E_2=\sum_{t=t'+1}^{t_1-1}\alpha^{t_1-t}\epsilon(t)$ denote the weighted sum of the noises after time of reflection $t'$. Now conditioning on the initial expected reward $r(t_0)$, we have

$$\mathbb{P}\Big[\Big|\sum_{t=t_0}^{t_1-1}\alpha^{t_1-t}\epsilon_1(t)\Big|\le s\,|\,r(t_0)\Big]=\quad \mathbb{P}\Big[\Big|\alpha^{t_1-t'-1}E_1+\alpha^{t_1-t'}\epsilon_1(t)+E_2\Big|\le s\,|\,r(t_0)\Big]$$

$$=\quad \mathbb{P}\Big[-s-\alpha^{t_1-t'-1}(\mathcal{R}-L)\le E_2\le s-\alpha^{t_1-t'-1}(\mathcal{R}-L)\,|\,r(t_0)\Big]$$

$$\ge\quad \mathbb{P}\Big[-s-\alpha^{t_1-t'-1}(\mathcal{R}+d-L)\le E_2\le s-\alpha^{t_1-t'-1}(\mathcal{R}+d-L)\,|\,r(t_0)\Big]$$

$$=\quad \mathbb{P}\Big[\Big|\alpha^{t_1-t'-1}E_1+\alpha^{t_1-t'}\epsilon(t)+E_2\Big|\le s\,|\,r(t_0)\Big]=\quad \mathbb{P}\Big[\Big|\sum_{t=t_0}^{t_1-1}\alpha^{t_1-t}\epsilon(t)\Big|\le s\,|\,r(t_0)\Big]. \tag{20}$$

Note that the inequality in (20) holds because $E_2$ is normally distributed (it is a weighted sum of normal noises). If we let $q_1=\alpha^{t_1-t'-1}(\mathcal{R}-L)$ and $q_2=\alpha^{t_1-t'-1}(\mathcal{R}+d-L)$, we must have $|q_1|<|q_2|$. This, along with the normal distribution of $E_2$, gives

$$\mathbb{P}[-s-q_1\le E_2\le s-q_1]\ge \mathbb{P}[-s-q_2\le E_2\le s-q_2].$$

Following the same procedure as above, we can define fictional scenarios in which $r(.)$ only gets truncated when it hits the boundary for the first $m$ times, for $1\le m\le t_1-t_0$. We can then define corresponding fictional noises $\epsilon_m(t_0),\dots,\epsilon_m(t_1-1)$ and use the same proof ideas to show

$$\mathbb{P}\Big[\Big|\sum_{t=t_0}^{t_1-1}\alpha^{t_1-t}\epsilon_m(t)\Big|\le s\,|\,r(t_0)\Big]\ge \mathbb{P}\Big[\Big|\sum_{t=t_0}^{t_1-1}\alpha^{t_1-t}\epsilon_{m-1}(t)\Big|\le s\,|\,r(t_0)\Big]$$

for all $1\le m\le t_1-t_0$. Note that if $m=t_1-t_0$, the fictional rewards $\epsilon_m=\tilde{\epsilon}$ for all $t\in[t_0,t_1-1]$. Hence,

$$\mathbb{P}\Big[\Big|\sum_{t=t_0}^{t_1-1}\alpha^{t_1-t}\tilde{\epsilon}(t)\Big|\le s\,|\,r(t_0)\Big]\ge \mathbb{P}\Big[\Big|\sum_{t=t_0}^{t_1-1}\alpha^{t_1-t}\epsilon(t)\Big|\le s\,|\,r(t_0)\Big].$$

Finally, since (19) holds for any initial expected reward $r(t_0)$, taking expectation with respect to $r(t_0)$ gives (18), which concludes the proof. ∎

**Proof of Lemma 6.3:** Let us first consider the sequence of noises $\epsilon_i(t)$ (i.e., the noises before applying the truncating boundary). For any fixed $i \in [k]$, and for $t' \in [t - \Delta_{\text{ep}}, t + \Delta_{\text{ep}}]$, we have $\epsilon_i(t') \overset{i.i.d.}{\sim} N(0, \sigma)$. Then, by Hoeffding's inequality (see Lemma J.3, where for $1 \le j \le t_1 - t_0$, we set $X_j = \alpha^j \epsilon_i(t_1 - j)$, $\sigma_j = \alpha^j \sigma$ and $s = \frac{c_0 \sigma}{t_1 - t_0} \sqrt{\frac{\alpha^2 - \alpha^{2(t_1 - t_0 + 1)}}{1 - \alpha^2}}$ ), for $t - \Delta_{\text{ep}} \le t_0 < t_1 \le t + \Delta_{\text{ep}}$, we have

$$\mathbb{P}\left[ \left| \sum_{t=t_0}^{t_1 - 1} \alpha^{t_1 - t} \epsilon_i(t) \right| > c_0 \sigma \sqrt{(\alpha^2 - \alpha^{2(t_1 - t_0 + 1)})/(1 - \alpha^2)} \right] \le 2\exp\left(-\frac{c_0^2}{2}\right) = \frac{(\alpha\sigma)^2}{2k\Delta_{\text{ep}}^2},$$

where the last equality follows from $c_0 = \sqrt{4\log(1/\alpha\sigma) + 4\log\Delta_{\text{ep}} + 2\log(4k)}$. By Lemma H.1, we can use the above inequality to also obtain a concentration inequality for the weighted sum of the new noises $\tilde{\epsilon}_i(t)$, which incorporate the influence of truncating boundary:

$$\mathbb{P}\left[ \left| \sum_{t=t_0}^{t_1 - 1} \alpha^{t_1 - t} \tilde{\epsilon}_i(t) \right| > c_0 \sigma \sqrt{(\alpha^2 - \alpha^{2(t_1 - t_0 + 1)})/(1 - \alpha^2)} \right]$$

$$\le \quad \mathbb{P}\left[ \left| \sum_{t=t_0}^{t_1 - 1} \alpha^{t_1 - t} \epsilon_i(t) \right| > c_0 \sigma \sqrt{(\alpha^2 - \alpha^{2(t_1 - t_0 + 1)})/(1 - \alpha^2)} \right] \le \frac{(\alpha\sigma)^2}{2k\Delta_{\text{ep}}^2}.$$

Since we have $k$ arms and the number of sub-intervals to consider in $[t - \Delta_{\text{ep}}, t + \Delta_{\text{ep}}]$ is $\frac{2\Delta_{\text{ep}}(2\Delta_{\text{ep}} - 1)}{2}$, we can apply union bounds to get

$$\mathbb{P}[\mathcal{G}^c(t)] \le \sum_{i=1}^{k} \sum_{t - \Delta_{\text{ep}} \le t_0 < t_1 \le t + \Delta_{\text{ep}}} \mathbb{P}\left[ \left| \sum_{t=t_0}^{t_1 - 1} \alpha^{t_1 - t} \tilde{\epsilon}_i(t) \right| > c_0 \sigma \sqrt{(\alpha^2 - \alpha^{2(t_1 - t_0 + 1)})/(1 - \alpha^2)} \right]$$

$$\le \sum_{i=1}^{k} \frac{2\Delta_{\text{ep}}(2\Delta_{\text{ep}} - 1)}{2} \cdot \frac{(\alpha\sigma)^2}{2k\Delta_{\text{ep}}^2} \quad \le \sum_{i=1}^{k} 2\Delta_{\text{ep}}^2 \cdot \frac{(\alpha\sigma)^2}{2k\Delta_{\text{ep}}^2} \quad = (\alpha\sigma)^2. \quad \blacksquare$$

Having showed that good event $\mathcal{G}(t)$ happens with high probability, we now establish a number of important implications of the good event:

(i) In Lemma H.2, we show that the expected reward $r_i(t)$ satisfies a series of concentration inequalities. These inequalities guarantee that our estimated reward $\widehat{r}_i(t)$ does not deviate too much from the true expected reward $r_i(t)$.

(ii) In Lemma H.3, we show that AR2 does not wait too long to play an arm with high expected reward.

(iii) In Lemma H.4, we show that the best expected reward $r^\star(t)$ is not too far away from our estimate reward for the superior arm $\widehat{r}_{\text{sup}}(t)$.

These implications will be critical for our proof of Lemma 6.4. Before we proceed, let us define the following:

$$\mathcal{F}(\Delta t) \triangleq c_0 \sigma \sqrt{\alpha^2 + \alpha^4 + \cdots + \alpha^{2\Delta t}} = c_0 \sigma \sqrt{(\alpha^2 - \alpha^{2(\Delta t + 1)})/(1 - \alpha^2)}, \tag{21}$$

where the second equality follows from the geometric sum formula. The expression (21) will be used throughout the rest of the analysis. In particular, since $c_1 = 24c_0$, the triggering criteria becomes: at round $t$, an arm $i \ne i_{\text{sup}}(t)$ gets triggered if the following is satisfied:

$$\widehat{r}_{\text{sup}}(t) - \widehat{r}_i(t) \le c_1 \sigma \sqrt{(\alpha^2 - \alpha^{2(t - \tau_i(t) + 1)})/(1 - \alpha^2)} = 24\mathcal{F}(t - \tau_i(t)).$$

**Lemma H.2.** *Given $\mathcal{G}(t)$, for any arm $i \in [k]$ and any rounds $t_0, t_1$ such that $t - \Delta_{ep} \le t_0 < t_1 \le t + \Delta_{ep}$, the expected rewards satisfy the following inequalities:*

$$|r_i(t_1) - \alpha^{t_1 - t_0} r_i(t_0)| \le \mathcal{F}(t_1 - t_0) \tag{22}$$

$$|r^\star(t_1) - \alpha^{t_1 - t_0} r^\star(t_0)| \le \mathcal{F}(t_1 - t_0) \tag{23}$$

$$|\Delta r_i(t_1) - \alpha^{t_1 - t_0} \Delta r_i(t_0)| \le 2\mathcal{F}(t_1 - t_0) \tag{24}$$

**Proof of Lemma H.2**    Given the AR-1 model and (5), we have the following recursive relationship

$$r_i(t_1) = \alpha\left[r_i(t_1 - 1) + \tilde{\epsilon}_i(t_1 - 1)\right]$$
$$= \alpha\left[\alpha(r_i(t_1 - 2) + \tilde{\epsilon}_i(t_1 - 2)) + \tilde{\epsilon}_i(t_1 - 1)\right]$$
$$= \alpha^{t_1 - t_0} r_i(t_0) + \sum_{t=t_0}^{t_1 - 1} \alpha^{t_1 - t}\tilde{\epsilon}_i(t) \tag{25}$$

by repeating the above steps. Then by Lemma 6.3, we have that

$$\left|r_i(t_1) - \alpha^{t_1 - t_0} r_i(t_0)\right| = \left|\sum_{t=t_0}^{t_1 - 1} \alpha^{t_1 - t}\tilde{\epsilon}_i(t)\right| \leq \mathcal{F}(t_1 - t_0),$$

which concludes the proof for (22).

As for (23), note that

$$r^\star(t_1) = \max_{i\in[k]} r_i(t_1) \leq \max_{i\in[k]}\left\{\alpha^{t_1 - t_0} r_i(t_0) + \mathcal{F}(t_1 - t_0)\right\} \leq \alpha^{t_1 - t_0} r^\star(t_0) + \mathcal{F}(t_1 - t_0)$$

and the other side of the inequality can be derived similarly.

The last inequality (24) easily follows from (22), (23), and the triangle inequality

$$|\Delta r_i(t_1) - \alpha^{t_1 - t_0}\Delta r_i(t_0)| \leq |(r^\star(t_1) - r_i(t_1)) - \alpha^{t_1 - t_0}(r^\star(t_0) - r_i(t_0))|$$
$$\leq |r^\star(t_1) - \alpha^{t_1 - t_0} r^\star(t_0)| + |r_i(t_1) - \alpha^{t_1 - t_0} r_i(t_0)| \leq 2\mathcal{F}(t_1 - t_0).$$

This concludes the proof of Lemma H.2. ∎

**Lemma H.3.** *Suppose we apply* AR2 *to the non-stationary MAB problem with* $\alpha \in [\bar{\alpha}, 1)$ *and* $k \leq \mathcal{K}(\alpha)$, *given* $\mathcal{G}(t)$, *the following holds for the best arm* $i$ *at round* $t$: $t - \tau_i(t) \leq 4k - 3$.

**Proof of Lemma H.3**    Suppose $\alpha \in [\bar{\alpha}, 1)$. Let arm $i$ be the best arm at round $t$, i.e. $i = \arg\max_j r_j(t)$. We let $\tau = \tau_i(t)$, and let $\tau^{\text{trig}} \geq \tau$ be the triggering time of arm $i$ after it gets pulled at round $\tau$. First note that by design of AR2, we must have $t - \tau^{\text{trig}} \leq 2(k - 1)$, since at any given round there are at most $k - 1$ triggered arms. We would prove the statement of the lemma by showing the following inequality:

$$\tau^{\text{trig}} - \tau \leq t - \tau^{\text{trig}} + 1 \tag{26}$$

If (26) holds, we then have $t - \tau = (\tau^{\text{trig}} - \tau) + (t - \tau^{\text{trig}}) \leq 2(t - \tau^{\text{trig}}) + 1 \leq 4k - 3$.

Suppose by contradiction that (26) does not hold. By the triggering criteria, we have

$$\widehat{r}_{\text{sup}}(\tau^{\text{trig}} - 1) - \widehat{r}_i(\tau^{\text{trig}} - 1) > 24 \cdot \mathcal{F}((\tau^{\text{trig}} - 1) - \tau) \tag{27}$$

Let $j = i_{\text{sup}}(\tau^{\text{trig}} - 1)$ as defined in (1). Hence we have $\widehat{r}_{\text{sup}}(\tau^{\text{trig}} - 1) = r_j(\tau^{\text{trig}} - 1)$ or $\widehat{r}_{\text{sup}}(\tau^{\text{trig}} - 1) = \alpha r_j(\tau^{\text{trig}} - 2)$. Let us assume $\widehat{r}_{\text{sup}}(\tau^{\text{trig}} - 1) = r_j(\tau^{\text{trig}} - 1)$ and the proof follows similarly for the other case. Since we also have $\widehat{r}_i(\tau^{\text{trig}} - 1) = \alpha^{\tau^{\text{trig}} - \tau - 2}\mathcal{B}(\alpha X_i(\tau)) = \alpha^{\tau^{\text{trig}} - \tau - 2} r_i(\tau + 1)$, (27) can be rewritten as

$$r_j(\tau^{\text{trig}} - 1) - \alpha^{\tau^{\text{trig}} - \tau - 2} r_i(\tau + 1) > 24 \cdot \mathcal{F}((\tau^{\text{trig}} - 1) - \tau) \tag{28}$$

By (22) of Lemma H.2, we also have the following two inequalities:

$$r_i(t) \leq \alpha^{t - (\tau + 1)} r_i(\tau + 1) + \mathcal{F}(t - (\tau + 1)) \tag{29}$$
$$r_j(t) \geq \alpha^{t - (\tau^{\text{trig}} - 1)} r_j(\tau^{\text{trig}} - 1) - \mathcal{F}(t - (\tau^{\text{trig}} - 1)) \tag{30}$$

Then by (29) and (30), we have

$$r_j(t) - r_i(t) \geq \left(\alpha^{t - (\tau^{\text{trig}} - 1)} r_j(\tau^{\text{trig}} - 1) - \mathcal{F}(t - (\tau^{\text{trig}} - 1))\right) - \left(\alpha^{t - (\tau + 1)} r_i(\tau + 1) + \mathcal{F}(t - (\tau + 1))\right)$$
$$\geq \alpha^{t - \tau^{\text{trig}} + 1}\left(r_j(\tau^{\text{trig}} - 1) - \alpha^{\tau^{\text{trig}} - \tau - 2} r_i(\tau + 1)\right) - \mathcal{F}(t - (\tau^{\text{trig}} - 1)) - \mathcal{F}(t - (\tau + 1))$$
$$\geq 24 \cdot \alpha^{t - \tau^{\text{trig}} + 1}\mathcal{F}((\tau^{\text{trig}} - 1) - \tau) - \mathcal{F}(t - (\tau^{\text{trig}} - 1)) - \mathcal{F}(t - (\tau + 1)),$$
$$\tag{31}$$

where the last inequality follows from (28). Since $k \leq \mathcal{K}(\alpha)$, we have $\alpha^{t-\tau^{\text{trig}}+1} \geq \alpha^{2k-1} \geq \frac{1}{8}$, which yields

$$r_j(t) - r_i(t) \geq 24 \cdot \frac{1}{8} \cdot \mathcal{F}(\tau^{\text{trig}} - \tau - 1) - \mathcal{F}(t - \tau^{\text{trig}} + 1) - \mathcal{F}(t - \tau - 1).$$

Since $\tau^{\text{trig}} - \tau \geq t - \tau^{\text{trig}} + 2$, we have $\mathcal{F}(\tau^{\text{trig}} - 1 - \tau) \geq \mathcal{F}(t - \tau^{\text{trig}} + 1)$. This further yields

$$r_j(t) - r_i(t) \geq (24 \cdot \frac{1}{8} - 1) \cdot \mathcal{F}(\tau^{\text{trig}} - \tau - 1) - \mathcal{F}(t - \tau - 1) = 2\mathcal{F}(\tau^{\text{trig}} - \tau - 1) - \mathcal{F}(t - \tau - 1).$$
$$(32)$$

Note that if $\tau^{\text{trig}} - \tau > t - \tau^{\text{trig}} + 1$, then since $t - \tau = (t - \tau^{\text{trig}}) + (\tau^{\text{trig}} - \tau)$, we must have $2(\tau^{\text{trig}} - \tau) > t - \tau + 1$, and thus

$$\begin{aligned}
2\mathcal{F}(\tau^{\text{trig}} - \tau - 1) &= 2c_0\sigma\sqrt{\alpha^2 + \cdots + \alpha^{2(\tau^{\text{trig}} - \tau - 1)}} \\
&> 2c_0\sigma\sqrt{\alpha^2 + \cdots + \alpha^{t - \tau - 1}} \\
&> c_0\sigma(\sqrt{\alpha^2 + \cdots + \alpha^{t-\tau-1}} + \sqrt{\alpha^{t-\tau+1} + \cdots + \alpha^{2(t-\tau)-2}}) \\
&\geq c_0\sigma\sqrt{\alpha^2 + \cdots + \alpha^{2(t-\tau)-2}} \quad \text{since } \sqrt{a} + \sqrt{b} \geq \sqrt{a+b} \\
&= \mathcal{F}(t - \tau - 1).
\end{aligned}$$

Plugging this into (32) gives $r_j(t) - r_i(t) > 0$, which contradicts that arm $i$ is the best arm at $t$. Hence, we must have $\tau^{\text{trig}} - \tau \leq t - \tau^{\text{trig}} + 1$. By our previous argument, this then gives $t - \tau \leq 4k - 3$. ■

**Lemma H.4.** *Suppose we apply* AR2 *to the non-stationary MAB problem with $\alpha \in [\bar{\alpha}, 1)$ and $k \leq \mathcal{K}(\alpha)$, for any arm $i \in [k]$ and round $t$, we have*

$$r^\star(t) - \widehat{r}_{sup}(t) \leq O(c_0\sigma\alpha k), \tag{33}$$

*where $\widehat{r}_{sup}(t) = \widehat{r}_{i_{sup}(t)}(t)$ and $i_{sup}(t)$ is defined in (1).*

**Proof of Lemma H.4:** Without loss of generality, suppose that $t$ is even, and as a result $I_t = i_{\text{sup}}(t)$. The proof then proceeds in two steps. In step one, we establish a recursive inequality for the estimated reward of the superior arm. In step two, we show that the estimated reward of the superior arm is in fact close to the expected reward of the best arm.

**Step 1: Establish a recursive inequality for $\widehat{r}_{\textbf{sup}}(t)$.** We first assume that $i_{\text{sup}}(t) = I_{t-2}$ and hence $\widehat{r}_{\text{sup}}(t) = \alpha\mathcal{B}(\alpha R_{i_{\text{sup}}(t)}(t-2)) = \alpha r_{i_{\text{sup}}(t)}(t-1)$. We also have

$$r_{i_{\text{sup}}(t)}(t) = \alpha r_{i_{\text{sup}}(t)}(t-1) + \alpha\epsilon_{i_{\text{sup}}(t)}(t-1) = \widehat{r}_{\text{sup}}(t) + \alpha\epsilon_{i_{\text{sup}}(t)}(t-1),$$

which then gives

$$r_{i_{\text{sup}}(t)}(t) \geq \widehat{r}_{\text{sup}}(t) - \alpha c_0\sigma \tag{34}$$

by definition of the good event $\mathcal{G}(t)$ (set $t_0 = t$ and $t_1 = t + 1$). Since $I_t = i_{\text{sup}}(t)$, we additionally have

$$\widehat{r}_{\text{sup}}(t+1) \geq \mathcal{B}(\alpha R_{i_{\text{sup}}(t)}(t)) = \alpha(r_{i_{\text{sup}}(t)}(t) + \epsilon_{i_{\text{sup}}(t)}(t)) \geq \alpha r_{i_{\text{sup}}(t)}(t) - \alpha c_0\sigma \tag{35}$$

Combining (34) and (35) gives us the following inequality, which is the desired result.

$$\widehat{r}_{\text{sup}}(t+1) \geq \alpha(\widehat{r}_{\text{sup}}(t) - \alpha c_0\sigma) - \alpha c_0\sigma \geq \alpha\widehat{r}_{\text{sup}}(t) - 2c_0\alpha\sigma. \tag{36}$$

On the other hand, if $i_{\text{sup}}(t) = I_{t-1}$, we have $r_{i_{\text{sup}}(t)}(t) = \mathcal{B}(\alpha R_{i_{\text{sup}}(t)}(t-1)) = \widehat{r}_{\text{sup}}(t)$, and combining this with (35) also gives

$$\widehat{r}_{\text{sup}}(t+1) \geq \alpha r_{i_{\text{sup}}(t)}(t) - \alpha c_0\sigma = \alpha\widehat{r}_{\text{sup}}(t) - \alpha c_0\sigma \geq \alpha\widehat{r}_{\text{sup}}(t) - 2c_0\alpha\sigma.$$

Overall, we show that $\widehat{r}_{\text{sup}}(t+1) \geq \alpha\widehat{r}_{\text{sup}}(t) - 2c_0\alpha\sigma$, which completes the proof of the first step.

**Step 2: Show that $\widehat{r}_{\textbf{sup}}(t)$ and $r^\star(t)$ are close.** For simplicity of notation, let $j$ be the best arm at round $t$, so $r^\star(t) = r_j(t)$. Let $\tau = \tau_j(t)$. Recall that by Lemma H.3, we have that $t - \tau \leq 4k - 3 = O(k)$. First note that

$$\begin{aligned}
\widehat{r}_{\text{sup}}(t) &\geq \alpha\widehat{r}_{\text{sup}}(t-1) - 2c_0\alpha\sigma && \text{by (36)} \\
&\geq \alpha^{t-(\tau+1)}\widehat{r}_{\text{sup}}(\tau+1) - 2c_0\sigma(\alpha + \cdots + \alpha^{t-(\tau+1)}) && \text{by applying (36) recursively} \\
&\geq \alpha^{t-(\tau+1)}\widehat{r}_{\text{sup}}(\tau+1) - 2c_0\sigma(\alpha + \cdots + \alpha^{4k-4}) && \text{since } t - \tau \leq 4k - 3 \\
&\geq \alpha^{t-\tau-1}\mathcal{B}(\alpha R_j(\tau)) - 2c_0\sigma\alpha(4k-4) && \text{since } \widehat{r}_{\text{sup}}(\tau+1) \geq \widehat{r}_j(\tau+1) = \mathcal{B}(\alpha R_j(\tau)) \\
&= \alpha^{t-\tau-1}r_j(\tau+1) - O(c_0\sigma\alpha k)
\end{aligned}$$
$$(37)$$

We also note that

$$
\begin{aligned}
r^\star(t) = r_j(t) &\le \alpha^{t-\tau-1} r_j(\tau+1) + \mathcal{F}(t-\tau-1) && \text{by (22) of Lemma H.2} \\
&= \alpha^{t-\tau-1} r_j(\tau+1) + c_0 \sigma \sqrt{\alpha^2 + \cdots + \alpha^{2(t-\tau-1)}} \\
&\le \alpha^{t-\tau-1} r_j(\tau+1) + c_0 \sigma \alpha \sqrt{t-\tau} \\
&= \alpha^{t-\tau-1} r_j(\tau+1) + O(c_0 \sigma \alpha \sqrt{k}) && \text{since } t - \tau \le 4k-3.
\end{aligned}
\tag{38}
$$

Combining (37) and (38) then gives $r^\star(t) - \widehat{r}_{\sup}(t) \le O(c_0 \sigma \alpha k)$. The proof works in a similar fashion for the case when $t$ is odd. ∎

## H.2 Step 3: Distributed regret analysis.

Recall from Definition 6.1 that the distributed regret of arm $i$ at round $t$ is defined as

$$
D_i(t) = \left( \frac{\Delta r_i(\tau_i(t))}{1 + \alpha^2 + \cdots + \alpha^{2(\Delta \tau_i(t)-1)}} \right) \alpha^{2(t-\tau_i(t))}.
\tag{39}
$$

Before proceeding with the proof of Lemma 6.4, we first establish the following two claims. In Claim H.5 and Claim H.6, we respectively bound the nominator and denominator of $D_i(t)$, conditioning on the good event $\mathcal{G}(t)$ and the value of $\Delta r_i(t)$.

**Claim H.5.** *Suppose we apply* AR2 *to the non-stationary MAB problem with $\alpha \in [\bar{\alpha}, 1)$ and $k \le \mathcal{K}(\alpha)$. Given $\mathcal{G}(t)$, for any arm $i \in [k]$, let $\eta \triangleq \Delta r_i(t) > 0$ for time $\tau_i(t) \le t < \tau_i^{\mathrm{next}}(t)$, we have that the nominator of* (39) *satisfies: $\alpha^{2(t-\tau_i(t))} \Delta r_i(\tau_i(t)) \le O(\eta + c_0 \sigma \alpha \sqrt{k})$.*

**Proof of Claim H.5:** For simplicity of notation, let $\tau = \tau_i(t)$ and $\tau^{\mathrm{next}} = \tau_i^{\mathrm{next}}(t)$. Let $\tau^{\mathrm{trig}}$ be the first round at which arm $i$ gets triggered after it gets pulled at round $\tau$. We divide our proof into two cases.

**Case 1: $\tau^{\mathrm{trig}} - \tau \le t - \tau^{\mathrm{trig}}$.** If $\tau^{\mathrm{trig}} - \tau \le t - \tau^{\mathrm{trig}} \le O(k)$, we must have $t - \tau = O(k)$. Then by (22) of Lemma H.2, we have

$$
\begin{aligned}
&|\Delta r_i(t) - \alpha^{t-\tau} \Delta r_i(\tau)| \le 2\mathcal{F}(t-\tau) \le 2c_0 \sigma \alpha \sqrt{t-\tau} \le O(c_0 \sigma \alpha \sqrt{k}) \\
\Rightarrow\; & \alpha^{t-\tau} \Delta r_i(\tau) \le \eta + O(c_0 \sigma \alpha \sqrt{k}) \\
\Rightarrow\; & \alpha^{2(t-\tau)} \Delta r_i(\tau) \le \alpha^{t-\tau} O(\eta + c_0 \sigma \alpha \sqrt{k}) \le O(\eta + c_0 \sigma \alpha \sqrt{k}).
\end{aligned}
$$

**Case 2: $\tau^{\mathrm{trig}} - \tau > t - \tau^{\mathrm{trig}}$.** Consider time $t$ such that $\tau \le t < \tau^{\mathrm{next}}$. Then, we have

$$
\begin{aligned}
\widehat{r}_{\sup}(t) = \max \left\{ \mathcal{B}(\alpha R_{I_{t-1}}(t-1)), \alpha \mathcal{B}(\alpha R_{I_{t-2}}(t-2)) \right\} &= \max \left\{ r_{I_{t-1}}(t), \alpha r_{I_{t-2}}(t-1) \right\} \\
\le \max \left\{ r^\star(t), \alpha r^\star(t-1) \right\} \quad &\le \max \left\{ r^\star(t), r^\star(t) + 2c_0 \sigma \sqrt{\alpha^2} \right\} \quad \text{by (23) of Lemma H.2} \\
&= r^\star(t) + 2c_0 \sigma \alpha.
\end{aligned}
\tag{40}
$$

Let $t' = \tau^{\mathrm{trig}} - 1$ denote the round right before the round arm $i$ gets triggered. Then since at round $t'$ the arm has not been triggered, we have

$$
\begin{aligned}
24\mathcal{F}(t'-\tau) &\le \widehat{r}_{\sup}(t') - \widehat{r}_i(t') \\
&\le r^\star(t') + 2c_0 \sigma \alpha - \alpha^{t'-\tau-1} r_i(\tau+1) \quad \text{by (40)} \\
&\le \Delta r_i(t') + 2c_0 \sigma \alpha + \mathcal{F}(t' - (\tau+1)) \quad \text{by (22) of Lemma H.2} \\
&\le \Delta r_i(t') + 2c_0 \sigma \alpha + \mathcal{F}(t' - \tau)
\end{aligned}
\tag{41}
$$

This then gives

$$
\mathcal{F}(t'-\tau) \le \frac{1}{23} \left( \Delta r_i(t') + O(c_0 \sigma \alpha) \right)
\tag{42}
$$

By (22) of Lemma H.2 and Equation (42), we also have

$$
\alpha^{t-t'} \Delta r_i(t') \le \Delta r_i(t) + 2\mathcal{F}(t-t') \le \Delta r_i(t) + \frac{2}{23} \Delta r_i(t') + O(c_0 \alpha \sigma)
\tag{43}
$$

We also have
$$\alpha^{2k-1}\Delta r_i(t') \leq \alpha^{t-t'}\Delta r_i(t'), \tag{44}$$

since $t - t' = t - \tau^{\text{trig}} + 1 \leq 2(k-1) + 1 = 2k - 1$. Now since $k \leq \mathcal{K}(\alpha) = \frac{1}{2}(\log(\frac{1}{8})/\log\alpha + 1)$, we have $\alpha^{2k-1} \geq \frac{1}{8}$, and hence

$$\left(\frac{1}{8} - \frac{1}{23}\right)\Delta r_i(t') \leq \Delta r_i(t) + O(c_0\alpha\sigma) \tag{45}$$

Now by (22) of Lemma H.2, we also have

$$\alpha^{t'-\tau}\Delta r_i(\tau) \leq \Delta r_i(t') + 2\mathcal{F}(t' - \tau) \leq \Delta r_i(t') + \frac{2}{23}\Delta r_i(t') + O(c_0\alpha\sigma), \tag{46}$$

where the last inequality follows from (42) and $t - t' \leq t' - \tau$. By multiplying the last equation by $\alpha^{t-t'}$, and by applying Equation (45), we then have

$$\alpha^{t-\tau}\Delta r_i(\tau) \leq \alpha^{t-t'}\left(\Delta r_i(t') + \frac{2}{23}\Delta r_i(t') + O(c_0\alpha\sigma)\right) \leq (1 + \frac{2}{23})\alpha^{t-t'}\Delta r_i(t') + O(c_0\alpha^{t-t'+1}\sigma)$$

$$\leq (1 + \frac{2}{23})\frac{\Delta r_i(t) + O(c_0\alpha\sigma)}{\frac{1}{8} - \frac{1}{23}} + O(c_0\alpha^{t-t'+1}\sigma) = O(\Delta r_i(t) + c_0\alpha\sigma) = O(\eta + c_0\alpha\sigma) \tag{47}$$

Since $t - \tau \geq 0$, we then have $\alpha^{2(t-\tau)}\Delta r_i(\tau) \leq O(\eta + c_0\alpha\sigma)$.

In summary of both cases, we have $\alpha^{2(t-\tau)}\Delta r_i(\tau) \leq O(\eta + c_0\sigma\alpha\sqrt{k})$. ∎

**Claim H.6.** *Suppose we apply* AR2 *to the non-stationary MAB problem with $\alpha \in [\bar{\alpha}, 1)$ and $k \leq \mathcal{K}(\alpha)$. Given $\mathcal{G}(t)$, for any arm $i \in [k]$, let $\eta \triangleq \Delta r_i(t) > 0$ for time $\tau \leq t < \tau^{\text{next}}$, the denominator of (3) satisfies*

$$1 + \alpha^2 + \cdots + \alpha^{2(\Delta\tau - 1)} = \frac{1 - \alpha^{2\Delta\tau}}{1 - \alpha^2} \geq \max\{\Omega(\frac{\eta^2}{c_0^2\sigma^2\alpha^2k^2}), 1\}.$$

**Proof of Claim H.6** The first equality follows from the geometric sum formula. Since $1 + \alpha^2 + \cdots + \alpha^{2(\Delta\tau-1)} \geq 1$, it suffices to show that it is also lower bounded by $\Omega(\frac{\eta^2}{c_0^2\sigma^2\alpha^2k^2})$. Let $\tau \leq t < \tau^{\text{next}}$ and recall that $\eta = \Delta r_i(t)$. We divide our proof into two cases.

**Case 1:** $\alpha^{t-\tau}\Delta r_i(\tau) \leq \eta/2$. Then by (22) of Lemma H.2, we have

$$|\eta - \alpha^{t-\tau}\Delta r_i(\tau)| = |\Delta r_i(t) - \alpha^{t-\tau}\Delta r_i(\tau)| \leq 2\mathcal{F}(t - \tau)$$

$$\Rightarrow 2\mathcal{F}(t - \tau) \geq \eta/2 \quad \Rightarrow \quad 2c_0\sigma\alpha\sqrt{(1 - \alpha^{2(t-\tau)})/(1 - \alpha^2)} \geq \eta/2$$

$$\Rightarrow (1 - \alpha^{2(t-\tau)})/(1 - \alpha^2) \geq \Omega(\frac{\eta^2}{c_0^2\sigma^2\alpha^2})$$

Since $\Delta\tau \geq t - \tau$, we have $\dfrac{1 - \alpha^{2\Delta\tau}}{1 - \alpha^2} \geq \dfrac{1 - \alpha^{2(t-\tau)}}{1 - \alpha^2} \geq \Omega(\frac{\eta^2}{c_0^2\sigma^2\alpha^2})$.

**Case 2:** $\alpha^{t-\tau}\Delta r_i(\tau) > \eta/2$. We would show that $\dfrac{1 - \alpha^{2\Delta\tau}}{1 - \alpha^2} \geq \Omega(\frac{\eta^2}{c_0^2\sigma^2\alpha^2k^2})$. We have

$$\begin{aligned}
\widehat{r}_{\text{sup}}(t) - \widehat{r}_i(t) &= \widehat{r}_{\text{sup}}(t) - \alpha^{t-\tau}r_i(\tau) \\
&\geq (r^\star(t) - O(c_0\sigma\alpha k)) - \alpha^{t-\tau}(r^\star(\tau) - \Delta r_i(\tau)) && \text{by Lemma H.4} \\
&= \alpha^{t-\tau}\Delta r_i(\tau) + \left(r^\star(t) - \alpha^{t'-\tau}r^\star(\tau)\right) - O(c_0\sigma\alpha k) \\
&\geq \frac{\eta}{2} - (2\mathcal{F}(t - \tau) + O(c_0\sigma\alpha k)) && \text{by (23) of Lemma H.2} \\
&\geq \frac{\eta}{2} - O\left(k\mathcal{F}(t - \tau)\right).
\end{aligned} \tag{48}$$

Now consider $\tau^{\text{trig}}$, the round at which arm $i$ gets triggered. By the triggering criteria, we have

$$\widehat{r}_{\sup}(\tau^{\text{trig}}) - \widehat{r}_i(\tau^{\text{trig}}) \leq 24\mathcal{F}(\tau^{\text{trig}} - \tau) \Rightarrow \frac{\eta}{2} - O\left(k\mathcal{F}(\tau^{\text{trig}} - \tau)\right) \leq 24\mathcal{F}(\tau^{\text{trig}} - \tau) \quad \text{by (48)}$$

$$\Rightarrow O\left(k\mathcal{F}(\tau^{\text{trig}} - \tau)\right) \geq \frac{\eta}{2}$$

$$\Rightarrow \frac{\alpha^2 - \alpha^{2(\tau^{\text{trig}} - \tau + 1)}}{1 - \alpha^2} \geq O(\frac{\eta^2}{c_0^2 \sigma^2 k^2}) \qquad \text{by (21)}$$

$$\Rightarrow \frac{1 - \alpha^{2(\tau^{\text{trig}} - \tau)}}{1 - \alpha^2} \geq O(\frac{\eta^2}{c_0^2 \alpha^2 \sigma^2 k^2})$$

$$(49)$$

Since we must have $\Delta\tau > \tau^{\text{trig}} - \tau$, we have

$$\frac{1 - \alpha^{2\Delta\tau}}{1 - \alpha^2} \geq \frac{1 - \alpha^{2(\tau^{\text{trig}} - \tau)}}{1 - \alpha^2} \geq \Omega(\frac{\eta^2}{c_0^2 \alpha^2 \sigma^2 k^2}). \qquad \blacksquare$$

Having showed Claim H.5 and Claim H.6, we now formally prove Lemma 6.4.

**Proof of Lemma 6.4:** Fix arm $i \in [k]$ and round $t$. For simplicity of notation, let $\tau = \tau_i(t)$, $\tau^{\text{next}} = \tau_i^{\text{next}}(t)$ and $\Delta\tau = \Delta\tau_i(t)$. Let us write $\mathbb{E}[D_i(t)\mathbb{1}_{\mathcal{G}(t)}] = \mathbb{E}[D_i(t)\mathbb{1}_{\mathcal{G}(t)}\mathbb{1}_{\{\Delta r_i(t)=0\}}] + \mathbb{E}[D_i(t)\mathbb{1}_{\mathcal{G}(t)}\mathbb{1}_{\{\Delta r_i(t)>0\}}]$, and in the following analysis, we will bound the two terms on the right-hand side respectively.

**First term (Arm $i$ is optimal at round $t$).** We first rewrite the first term as

$$\mathbb{E}[D_i(t)\mathbb{1}_{\mathcal{G}(t)}\mathbb{1}_{\{\Delta r_i(t)=0\}}] = \mathbb{E}[D_i(t)|\mathcal{G}(t) \cap \{\Delta r_i(t) = 0\}] \cdot \mathbb{P}[\mathcal{G}(t) \cap \{\Delta r_i(t) = 0\}].$$

Note that if $\mathcal{G}(t)$ takes place and $\Delta r_i(t) = 0$, we have

$$\alpha^{t-\tau}\Delta r_i(\tau) = \left|\alpha^{t-\tau}\Delta r_i(\tau) - \Delta r_i(t)\right| \leq 2\mathcal{F}(t - \tau). \qquad (50)$$

by (24) in Lemma H.2. By Definition 6.1, we have

$$D_i(t) = \left(\frac{\Delta r_i(\tau)}{1 + \alpha^2 + \cdots + \alpha^{2(\Delta\tau - 1)}}\right)\alpha^{2(t-\tau)} = \left(\frac{\alpha^{t-\tau}\Delta r_i(\tau)}{\frac{1}{\alpha^2}(\alpha^2 + \cdots + \alpha^{2\Delta\tau})}\right)\alpha^{t-\tau}$$

$$\leq \left(\frac{2c_0\sigma\sqrt{\alpha^2 + \cdots + \alpha^{2(t-\tau)}}}{\alpha^2 + \cdots + \alpha^{2\Delta\tau}}\right)\alpha^{t-\tau+2} \qquad \text{by (21) and (50)}$$

$$\leq \left(\frac{2c_0\sigma}{\sqrt{\alpha^2 + \cdots + \alpha^{2(t-\tau)}}}\right)\alpha^{t-\tau+2} \qquad \text{since } 1 \leq t - \tau \leq \Delta\tau$$

$$(51)$$

That is,

$$\mathbb{E}[D_i(t)|\mathcal{G}(t) \cap \{\Delta r_i(t) = 0\}] \leq \left(\frac{2c_0\sigma}{\sqrt{\alpha^2 + \cdots + \alpha^{2(t-\tau)}}}\right)\alpha^{t-\tau+2}. \qquad (52)$$

Now, let $j$ be the optimal arm at time $\tau$. If $\mathcal{G}(t)$ takes place and $\Delta r_i(t) = 0$, we have

$$\Delta r_j(t) = \left|\alpha^{t-\tau}\Delta r_j(\tau) - \Delta r_j(t)\right| \leq 2\mathcal{F}(t - \tau).$$

by (24) in Lemma H.2. Hence we have

$$\mathbb{P}[\mathcal{G}(t) \cap \{\Delta r_i(t) = 0\}] \leq \mathbb{P}[\Delta r_j(t) \leq 2\mathcal{F}(t - \tau), \text{where } j = \arg\max_{j'} r_{j'}(\tau)]$$

$$\leq \mathbb{P}\left[\exists j \neq i \text{ such that } \Delta r_j(t) \leq 2\mathcal{F}(t - \tau)\right].$$

$$(53)$$

Recall from the discussion in Appendix D and Remark G.1 that the steady state distribution of $r_i(t)$ can be reasonably approximated by a uniform distribution, and (17) states that the PDF of the steady state distribution $f(x)$ satisfies $\rho_- \leq f(x) \leq \rho_+$ for all $x \in [-\mathcal{R}, \mathcal{R}]$, where $\rho_-, \rho_+$ are constants. Now let $r'_1, \ldots, r'_k$ be $k$ i.i.d. uniform$(-1/2\rho_+, 1/2\rho_+)$ random variables. We have

$$\mathbb{P}\left[\exists j \neq i \text{ such that } \Delta r_j(t) \leq 2\mathcal{F}(t - \tau)\right]$$

$$= \mathbb{P}\left[\text{the two best arms of } \{r_1(t), \ldots, r_k(t)\} \text{ are within } 2\mathcal{F}(t - \tau)\right]$$

$$\leq \mathbb{P}\left[\text{the two best arms of } \{r'_1, \ldots, r'_k\} \text{ are within } 2\mathcal{F}(t - \tau)\right] = O(k \cdot \mathcal{F}(t - \tau))^{11} \qquad (54)$$

Combining (53) and (54) thus gives

$$\mathbb{P}[\mathcal{G}(t) \cap \{\Delta r_i(t) = 0\}] \leq O(k \cdot \mathcal{F}(t - \tau)) \tag{55}$$

Overall, by (52) and (55), for all $t \geq \tau$ we have

$$\mathbb{E}[D_i(t)\mathbb{1}_{\mathcal{G}(t)}\mathbb{1}_{\{\Delta r_i(t)=0\}}] = \mathbb{E}[D_i(t)|\mathcal{G}(t) \cap \{\Delta r_i(t) = 0\}] \cdot \mathbb{P}[\mathcal{G}(t) \cap \{\Delta r_i(t) = 0\}]$$
$$\leq \left( \frac{2c_0\sigma}{\sqrt{\alpha^2 + \cdots + \alpha^{2(t-\tau)}}} \right) \alpha^{t-\tau+2} \cdot O\left(k \cdot \mathcal{F}(t - \tau)\right) = O(c_0^2\sigma^2\alpha^2 k). \tag{56}$$

**Second term (Arm $i$ is sub-optimal at round $t$).** Given $\mathcal{G}(t)$ and the value of $\eta \triangleq \Delta r_i(t) > 0$, we have bounded the nominator and denominator of $D_i(t)$ in Claim H.5 and Claim H.6 respectively. For any time $\tau \leq t < \tau^{\text{next}}$, in Claim H.5 we have showed $\alpha^{2(t-\tau)}\Delta r_i(\tau) \leq O(\eta + c_0\sigma\alpha\sqrt{k})$ and in Claim H.6 we have showed $1 + \alpha^2 + \cdots + \alpha^{2(\Delta\tau-1)} \geq \max\left\{\Omega(\frac{\eta^2}{c_0^2\sigma^2\alpha^2 k^2}), 1\right\}$. These results then give

$$\mathbb{E}\left[D_i(t)\mathbb{1}_{\mathcal{G}(t)}\mathbb{1}_{\{\Delta r_i(t)>0\}}|\eta\right] \lesssim \min\left\{\frac{\eta + c_0\alpha\sigma\sqrt{k}}{\frac{\eta^2}{c_0^2\alpha^2\sigma^2 k^2}}, \eta + c_0\alpha\sigma\sqrt{k}\right\}$$

Let $\tilde{C} = c_0\alpha\sigma\sqrt{k}$, this then yields

$$\mathbb{E}\left[D_i(t)\mathbb{1}_{\mathcal{G}(t)}\mathbb{1}_{\{\Delta r_i(t)>0\}}|\eta\right] \lesssim \min\left\{k\left(\frac{\eta + \tilde{C}}{\frac{\eta^2}{\tilde{C}^2}}\right), \eta + \tilde{C}\right\} = \min\left\{k\left(\frac{\tilde{C}^2}{\eta} + \frac{\tilde{C}^3}{\eta^2}\right), \eta + \tilde{C}\right\}.$$

This then further leads to

$$\mathbb{E}\left[D_i(t)\mathbb{1}_{\mathcal{G}(t)}\mathbb{1}_{\{\Delta r_i(t)>0\}}|\eta\right] \lesssim \begin{cases} k\frac{\tilde{C}^2}{\eta} & \text{if } \eta > \tilde{C} \\ \tilde{C} & \text{if } 0 < \eta \leq \tilde{C} \end{cases}$$

We can now bound the second term by taking expectation of the above expression with respect to $\eta$. Recall from (17) that the PDF of the steady state distribution of $r_i(t)$ satisfies $\rho_- \leq f(x) \leq \rho_+$ for all $x \in [-\mathcal{R}, \mathcal{R}]$, where $\rho_-$ and $\rho_+$ are constants. This then gives

$$\mathbb{E}[D_i(t)\mathbb{1}_{\mathcal{G}(t)}\mathbb{1}_{\{\Delta r_i(t)>\tilde{C}\}}] \lesssim \int_{\tilde{C}}^1 k\frac{\tilde{C}^2}{\eta}d\eta = O(k\tilde{C}^2\log(\tilde{C})) = O(c_0^2\alpha^2\sigma^2 k^2\log(c_0\alpha\sigma\sqrt{k})) \tag{57}$$

and

$$\mathbb{E}[D_i(t)\mathbb{1}_{\mathcal{G}(t)}\mathbb{1}_{\{0<\Delta r_i(t)\leq\tilde{C}\}}] \lesssim \tilde{C} \cdot \mathbb{P}[0 < \Delta r_i(t) \leq \tilde{C}] = O(\tilde{C}^2) = O(c_0^2\alpha^2\sigma^2 k), \tag{58}$$

which provides an upper bound for the second term.

Finally, summing (56), (57), and (58) gives $\mathbb{E}[D_i(t)\mathbb{1}_{\mathcal{G}(t)}] \leq O(c_0^2\alpha^2\sigma^2 k^2\log(c_0\alpha\sigma\sqrt{k}))$, which concludes the proof of Lemma 6.4. ∎

# I  Missing Proof of Section 8

**Proof of Proposition 8.1:** Before proceeding, we first make some notational changes that would simplify our proof. Since we pull arm $i$ consecutively for $T_{\text{est}}$ rounds for some fixed arm $i \in [k]$, in the rest of this proof we omit the dependency of $R_i(t)$ on $i$ and denote $R_t \triangleq R_i(t)$.

Let us first define the loss function $\mathcal{L}(\alpha)$ stated in the optimization problem

$$\widehat{\alpha} = \arg\min_{\alpha \in (0,1)} \mathcal{L}(\alpha), \tag{59}$$

---

[11]This follows from the fact that if $X_{(1)} \leq X_{(2)} \leq \cdots \leq X_{(k)}$ are the order statistics of $k$ i.i.d. uniform$(0, 1)$ random variables, then $X_{(k)} - X_{(k-1)}$ follows the Beta$(1, k)$ distribution with CDF $F(x) = 1 - (1 - x)^k$ for $x \in [0, 1]$ (see, e.g., [17]).

which is the negative of the log-likelihood function:

$$\mathcal{L}(\alpha) = -\frac{1}{T_{\text{est}}} \sum_{t=1}^{T_{\text{est}}} \mathbb{1}\{R_t = \mathcal{R}\} \log\left[1 - \Phi(\frac{\mathcal{R} - \alpha R_{t-1}}{\sigma})\right]$$

$$+ \mathbb{1}\{R_t = -\mathcal{R}\} \log\left[\Phi(\frac{-\mathcal{R} - \alpha R_{t-1}}{\sigma})\right] \tag{60}$$

$$+ \mathbb{1}\{-\mathcal{R} < R_t < \mathcal{R}\} \log\left[\Phi(\frac{\mathcal{R} - \alpha R_{t-1}}{\sigma}) - \Phi(\frac{-\mathcal{R} - \alpha R_{t-1}}{\sigma})\right],$$

where $\Phi(.)$ is the CDF of standard normal distribution Note that the three terms in the loss function respectively represent the log likelihood of the observed reward at the upper boundary, at the lower boundary and in between the boundaries.

Recall that $\widehat{\alpha}$ is the solution to the optimization problem in (59). We can apply the second-order Taylor's Theorem to obtain the following:

$$\mathcal{L}(\alpha) - \mathcal{L}(\widehat{\alpha}) = -\mathcal{L}'(\alpha)(\widehat{\alpha} - \alpha) - \frac{1}{2}\mathcal{L}''(\tilde{\alpha})(\alpha - \widehat{\alpha})^2 \tag{61}$$

for some $\tilde{\alpha}$ between true AR parameter $\alpha$ and estimated AR parameter $\widehat{\alpha}$. In the rest of this proof, we will provide a bound for $|\widehat{\alpha} - \alpha|$ via bounding the first and second derivatives of the loss function $\mathcal{L}(.)$.

We thus have

$$\mathcal{L}'(\alpha) = -\frac{1}{T_{\text{est}}} \sum_{t=1}^{T_{\text{est}}} \mathbb{1}\{R_t = \mathcal{R}\} \cdot \log'\left[1 - \Phi(\frac{\mathcal{R} - \alpha R_{t-1}}{\sigma})\right] \cdot \left(\frac{-R_{t-1}}{\sigma}\right)$$

$$+ \mathbb{1}\{R_t = -\mathcal{R}\} \cdot \log'\left[\Phi(\frac{-\mathcal{R} - \alpha R_{t-1}}{\sigma})\right] \cdot \left(\frac{-R_{t-1}}{\sigma}\right)$$

$$+ \mathbb{1}\{-\mathcal{R} < R_t < \mathcal{R}\} \cdot \log'\left[\Phi(\frac{\mathcal{R} - \alpha R_{t-1}}{\sigma}) - \Phi(\frac{-\mathcal{R} - \alpha R_{t-1}}{\sigma})\right] \cdot \left(\frac{-R_{t-1}}{\sigma}\right).$$

where $\log' h(y)$ denote the derivative of function $\log h(y)$ with respect to $y$. Let

$$u_t(\alpha) = \mathbb{1}\{R_t = \mathcal{R}\} \cdot \log'\left[1 - \Phi(\frac{\mathcal{R} - \alpha R_{t-1}}{\sigma})\right] + \mathbb{1}\{R_t = -\mathcal{R}\} \cdot \log'\left[\Phi(\frac{-\mathcal{R} - \alpha R_{t-1}}{\sigma})\right]$$

$$+ \mathbb{1}\{-\mathcal{R} < R_t < \mathcal{R}\} \cdot \log'\left[\Phi(\frac{\mathcal{R} - \alpha R_{t-1}}{\sigma}) - \Phi(\frac{-\mathcal{R} - \alpha R_{t-1}}{\sigma})\right].$$

Note that $\mathcal{L}'(\alpha) = \frac{1}{T_{\text{est}}} \sum_{t=1}^{T_{\text{est}}} u_t(\alpha) \cdot \frac{R_{t-1}}{\sigma}$. To bound the first derivative of the loss function, our next goal is to bound $\sum_{t=1}^{T_{\text{est}}} u_t(\alpha)$. Let $S_j = \sum_{t=1}^{j} u_t(\alpha)$. Observe that

$$|S_j - S_{j-1}| \leq \sup_{\substack{y_1 \in (0, 2\mathcal{R}/\sigma), \\ y_2 \in -2\mathcal{R}/\sigma, 0)}} \left\{ \max\left\{ \log'[1 - \Phi(y_1)], \log' \Phi(y_2), \log'(\Phi(y_1) - \Phi(y_2)) \right\} \right\}.$$

Let us define

$$L_1 = \sup_{\substack{y_1 \in (0, 2\mathcal{R}/\sigma), \\ y_2 \in -2\mathcal{R}/\sigma, 0)}} \left\{ \max\left\{ \log'[1 - \Phi(y_1)], \log' \Phi(y_2), \log'(\Phi(y_1) - \Phi(y_2)) \right\} \right\}. \tag{62}$$

Since $\Phi(y)$ is the CDF of the standard normal distribution, $y_1 \in (0, 2\mathcal{R}/\sigma)$ and $y_2 \in -2\mathcal{R}/\sigma, 0)$, we have that $L_1$ is a constant well defined by the above expression. Also observe that

$$\mathbb{E}[S_j - S_{j-1}|S_{j-1}, \ldots, S_1] = \mathbb{E}[u_t(\alpha)|S_{j-1}, \ldots, S_1]$$

$$= \mathbb{P}\{R_t = \mathcal{R}\} \cdot \frac{-\phi(\frac{\mathcal{R} - \alpha R_{t-1}}{\sigma})}{\left[1 - \Phi(\frac{\mathcal{R} - \alpha R_{t-1}}{\sigma})\right]} + \mathbb{P}\{R_t = -\mathcal{R}\} \cdot \frac{\phi(\frac{-\mathcal{R} - \alpha R_{t-1}}{\sigma})}{\left[\Phi(\frac{-\mathcal{R} - \alpha R_{t-1}}{\sigma})\right]}$$

$$+ \mathbb{P}\{-\mathcal{R} < R_t < \mathcal{R}\} \cdot \frac{\phi(\frac{\mathcal{R} - \alpha R_{t-1}}{\sigma}) - \phi(\frac{-\mathcal{R} - \alpha R_{t-1}}{\sigma})}{\left[\Phi(\frac{\mathcal{R} - \alpha R_{t-1}}{\sigma}) - \Phi(\frac{-\mathcal{R} - \alpha R_{t-1}}{\sigma})\right]}$$

$$= -\phi(\frac{\mathcal{R} - \alpha R_{t-1}}{\sigma}) + \phi(\frac{-\mathcal{R} - \alpha R_{t-1}}{\sigma}) + \phi(\frac{\mathcal{R} - \alpha R_{t-1}}{\sigma}) - \phi(\frac{-\mathcal{R} - \alpha R_{t-1}}{\sigma})$$

$$= 0.$$

where $\Phi(y)$ and $\phi(y)$ are the CDF and PDF of the standard normal distribution. The third equality above follows from $\epsilon_t$ being independent from $S_1, \ldots, S_{j-1}$. Hence, $S_j$ is a martingale with bounded difference.

By Azuma-Hoeffding Inequality (see Lemma J.5), we thus have that for $\gamma > 0$, the following concentration result holds:

$$\mathbb{P}[|S_{T_{\text{est}}}| > L_1 \sqrt{2\beta T_{\text{est}} \log T_{\text{est}}}] \leq \frac{2}{T_{\text{est}}^{\gamma}} . \tag{63}$$

Let us denote the high probability event, i.e., $|S_{T_{\text{est}}}| \leq L_1 \sqrt{2\gamma T_{\text{est}} \log T_{\text{est}}}$, as $\mathcal{E}_1$. Note that conditioning on $\mathcal{E}_1$, we would have

$$\mathcal{L}'(\alpha) = \frac{1}{T_{\text{est}}} \sum_{t=1}^{T_{\text{est}}} u_t(\alpha) \cdot \frac{R_{t-1}}{\sigma} \leq \left(\frac{L_1 \mathcal{R}}{\sigma}\right) \cdot \sqrt{\frac{2\gamma \log T_{\text{est}}}{T_{\text{est}}}} . \tag{64}$$

We also bound the second derivative using a constant $L_2 \geq 0$ as follow:

$$\begin{aligned}
\mathcal{L}''(\alpha) = \ & \frac{1}{T_{\text{est}}} \sum_{t=1}^{T_{\text{est}}} \mathbb{1}\{R_t = \mathcal{R}\} \cdot \log'' \left[1 - \Phi(\frac{\mathcal{R} - \alpha R_{t-1}}{\sigma})\right] \cdot \left(\frac{R_{t-1}^2}{\sigma^2}\right) \\
& + \mathbb{1}\{R_t = -\mathcal{R}\} \cdot \log'' \left[\Phi(\frac{-\mathcal{R} - \alpha R_{t-1}}{\sigma})\right] \cdot \left(\frac{R_{t-1}^2}{\sigma^2}\right) \\
& + \mathbb{1}\{-\mathcal{R} < R_t < \mathcal{R}\} \cdot \log'' \left[\Phi(\frac{\mathcal{R} - \alpha R_{t-1}}{\sigma}) - \Phi(\frac{-\mathcal{R} - \alpha R_{t-1}}{\sigma})\right] \cdot \left(\frac{R_{t-1}^2}{\sigma^2}\right) \\
\geq \ & \frac{1}{T_{\text{est}}} \sum_{t=1}^{T_{\text{est}}} \left(\frac{R_{t-1}^2}{\sigma^2}\right) \cdot \inf_{\substack{y_1 \in (0, 2\mathcal{R}/\sigma), \\ y_2 \in -2\mathcal{R}/\sigma, 0)}} \left\{ \min \left\{ -\log''[1 - \Phi(y_1)], -\log'' \Phi(y_2), -\log''(\Phi(y_1) - \Phi(y_2)) \right\} \right\} \\
\triangleq \ & \frac{1}{T_{\text{est}}} \sum_{t=1}^{T_{\text{est}}} \left(\frac{R_{t-1}^2}{\sigma^2}\right) \cdot L_2
\end{aligned} \tag{65}$$

Here, we define the constant $L_2$ as

$$L_2 = \inf_{\substack{y_1 \in (0, 2\mathcal{R}/\sigma), \\ y_2 \in -2\mathcal{R}/\sigma, 0)}} \left\{ \min \left\{ -\log''[1 - \Phi(y_1)], -\log'' \Phi(y_2), -\log''(\Phi(y_1) - \Phi(y_2)) \right\} \right\} . \tag{66}$$

Similar to our arguments above, since $\Phi(y)$ is the CDF of the standard normal distribution, $y_1 \in (0, 2\mathcal{R}/\sigma)$ and $y_2 \in -2\mathcal{R}/\sigma, 0)$, we have that $L_2 \geq 0$ is a constant well defined by the last equality. Let $V = \mathbb{E}[R_t^2] > 0$ denote the variance of reward under the steady state distribution. Since $R_t^2$ is bounded, we can apply Hoeffding's Inequality (see Lemma J.4) that states

$$\mathbb{P}[|\sum_{t=1}^{T_{\text{est}}} R_{t-1}^2 - T_{\text{est}}V| \geq \frac{T_{\text{est}}V}{2}] \leq 2\exp\left(-\frac{T_{\text{est}}V^2}{2\mathcal{R}^2}\right) .$$

Let $\mathcal{E}_2$ denote the high probability event that $|\sum_{t=1}^{T_{\text{est}}} R_{t-1}^2 - T_{\text{est}}V| \leq \frac{T_{\text{est}}V}{2}$. Then, under this event, we have that $\sum_{t=1}^{T_{\text{est}}} R_{t-1}^2 \geq \frac{T_{\text{est}}V}{2}$. Hence, conditioning on the event $\mathcal{E}_2$, we have

$$\mathcal{L}''(\alpha) \geq \frac{V L_2}{2\sigma^2} . \tag{67}$$

After bounding the first and second derivatives of $\mathcal{L}(\alpha)$ respectively, we are now ready to show the proximity between our estimated parameter $\widehat{\alpha}$ and the true parameter $\alpha$, conditioning on the high-probability events $\mathcal{E}_1 \cap \mathcal{E}_2$. By optimality of $\widehat{\alpha}$, we have that $\mathcal{L}(\widehat{\alpha}) \leq \mathcal{L}(\alpha)$. This, together with (61), gives

$$\frac{1}{2}\mathcal{L}''(\tilde{\alpha})(\alpha - \widehat{\alpha})^2 \leq -\mathcal{L}'(\alpha)(\widehat{\alpha} - \alpha) \leq |\mathcal{L}'(\alpha)| \cdot |\widehat{\alpha} - \alpha| . \tag{68}$$

Using (67), we can bound the left-hand side as

$$\frac{1}{2}\mathcal{L}''(\tilde{\alpha})(\alpha - \widehat{\alpha})^2 \geq \frac{1}{2} \cdot \left(\frac{V L_2}{2\sigma^2}\right) \cdot (\alpha - \widehat{\alpha})^2 . \tag{69}$$

Conditioning on $\mathcal{E}_1$, we also know from (64) have that

$$|\mathcal{L}'(\alpha)| \cdot |\widehat{\alpha} - \alpha| \leq \left(\frac{L_1 \mathfrak{R}}{\sigma}\right) \cdot \sqrt{\frac{2\gamma \log T_{\text{est}}}{T_{\text{est}}}} \cdot |\widehat{\alpha} - \alpha| \tag{70}$$

Finally, combining (68), (69) and (70), we have that with probability at least $1 - 2/T_{\text{est}}^{\gamma} - 2\exp(-T_{\text{est}} V^2/2\mathfrak{R}^2)$, we have

$$\frac{1}{2} \cdot \left(\frac{V L_2}{2\sigma^2}\right) \cdot (\alpha - \widehat{\alpha})^2 \leq \left(\frac{L_1 \mathfrak{R}}{\sigma}\right) \cdot \sqrt{\frac{2\gamma \log T_{\text{est}}}{T_{\text{est}}}} \cdot |\widehat{\alpha} - \alpha|$$

which implies

$$|\widehat{\alpha} - \alpha| \leq \frac{4\sigma \mathfrak{R} L_1}{V L_2} \sqrt{\frac{2\gamma \log T_{\text{est}}}{T_{\text{est}}}} . \qquad \blacksquare$$

## J   Concentration Inequalities

In this section, we state some useful concentration inequalities for subgaussian variables (see [29, 8, 45] for references).

**Definition J.1** (Subgaussian variables). *We say that a zero-mean random variable $X$ is $\sigma$-subgaussian (or, $X \sim subG(\sigma)$), if for all $\lambda \in \mathbb{R}$, $\mathbb{E}[e^{\lambda X}] \leq e^{\frac{\lambda^2 \sigma^2}{2}}$.*

In particular, note that if $X \sim N(0, \sigma)$, then $X \sim \text{subG}(\sigma)$. Subgaussian variables satisfy the following concentration inequalities:

**Lemma J.2** (Concentration of subgaussian variables). *A zero-mean random variable $X$ is $\sigma$-subgaussian if and only if for any $s > 0$, $\mathbb{P}\left[|X| \geq s\right] \leq 2\exp(-\frac{s^2}{2\sigma^2})$.*

**Lemma J.3** (Hoeffding's Inequality). *Let $X_1, \ldots, X_n$ be independent zero-mean random variables such that $X_j \sim subG(\sigma_j)$, then for any $s > 0$, $\mathbb{P}\left[\left|\frac{1}{n}\sum_{j=1}^{n} X_j\right| \geq s\right] \leq 2\exp(-\frac{n^2 s^2}{2\sum_{j=1}^{n}\sigma_j^2})$.*

In particular, bounded variables are known to be subgaussian, and we thus have a special case of the Hoeffding's Inequality that applies to bounded variables:

**Lemma J.4** (Hoeffding's Inequality for Bounded Variables). *Let $X_1, \ldots, X_n$ be independent random variables such that $a_i \leq X_i \leq b_i$. Consider the sum of these random variables $S_n = \sum_{i=1}^{n} X_i$. We have that for $s > 0$, $\mathbb{P}\left[|S_n - \mathbb{E}[S_n]| \geq s\right] \leq 2\exp\left(-\frac{2s^2}{\sum_{i=1}^{n}(b_i - a_i)^2}\right)$.*

Another related concentration result is the Azuma-Hoeffding's Inequality stated below, which applies to martingales with bounded differences.

**Lemma J.5** (Azuma-Hoeffding's Inequality). *Suppose that $\{S_j : j = 0, 1, 2, \ldots\}$ is a martingale and has bounded difference $|S_k - S_{k-1}| \leq b_k$ almost surely. Then, the following concentration bound holds for any $N \in \mathbb{Z}^+$ and $s > 0$: $\mathrm{P}\left(|S_N - S_0| \geq s\right) \leq 2\exp\left(\frac{-s^2}{2\sum_{k=1}^{N} b_k^2}\right)$.*

