# OpenReview forum: "Non-Stationary Bandits with Auto-Regressive Temporal Dependency"
_NeurIPS.cc/2023/Conference — NeurIPS 2023 poster_

### Official Review · Reviewer_s53A · 2023-06-26

**Soundness:** 2 fair
**Presentation:** 3 good
**Contribution:** 2 fair
**Rating:** 6
**Confidence:** 2

**Summary:**

The authors propose a bandit algorithm in the restless setting, when the rewards have an auto-regressive structure.

**Strengths:**

1. Contributes to the non-stationary case, in contrast to the vast majority of results in the stationary setting.


**Weaknesses:**

1. Knowledge of (single parameter) alpha, and the problem of estimating it simultaneously. It is unclear of the theoretical claims (in addition to the numerical results) hold when these are relaxed (ref. Sec 2)

**Questions:**

1. Some discussion of Exp3's limitation is already discussed in the lower bound section. Would be great to see a discussion on on adversarial bandits and how much tighter your results are compared to some of the algorithms there.
2. Unclear why the dynamic benchmark was chosen instead of the static one (other than it does seem suitable and is harder?). If instead we use the static benchmark, would there be any regret UB/LB gains?

**Limitations:**

None identified.

---

> ### Author Rebuttal · Authors · 2023-08-09
>
> Thank you for your insightful feedback! We’ve addressed your questions below and will integrate the discussions into our revised paper.
>
> **Regarding learning the AR parameter,**
> * In Sec. 8, we introduced a MLE-based approach that learns the AR parameter. We provided theoretical guarantees for our estimation (Prop. 8.1), and also numerically showed that the performance of our algorithm remains robust and competitive even when there exists noises in estimated AR parameters (Fig. 2).
>
>     Specifically, our numerical results reveal that AR2 is robust to the lack of knowledge of $\alpha$. For instance, in experiments we conducted for Fig. 2, performing MLE with only 50-150 data points and using the estimated AR parameter in AR2 would only increase normalized per-round regret by at most 4% compared to using the true AR parameter.
>
>     Given the promising numerical results, we believe that our MLE-based approach, which also handles heterogeneous $\alpha_i$’s (as noted in Sec.8), can be a useful method for practitioners to estimate the AR parameters in reality.
>
> * Many decision-makers in practice also have some prior knowledge of the AR parameters given past data (see discussion in Sec. 2 and our case study in Sec. 7). This motivates our initial assumption of known AR parameters. Even if such prior knowledge isn’t fully accurate, using them as initial estimates can significantly ease our learning process.
>
> * The main focus of our paper is to study how to use knowledge of the temporal structure to enhance online decision-making. With this in mind, we first focused on the AR-based bandit problem assuming knowledge of $\alpha$, and addressed learning of the AR parameters as a subsequent problem. Even with ample information on the underlying AR process, our non-stationary MAB problem remains challenging.
>
>     We appreciate your comment on the simultaneous learning of AR parameters. Given the focus of this paper, we will leave its theoretical analysis as an exciting future direction, and will add it to our future works section.
>
> **Regarding comparison with Exp3,**
> * We first highlight that Exp3 is designed for a completely adversarial setting where rewards can change arbitrarily. It is shown to be optimal against a weak static benchmark, which only considers a single best arm in hindsight, rather than a dynamic benchmark that chooses the best arm at every round. Further, it does not consider any temporal structure associated with the reward distributions. Hence in our setting where we intend to capture the best moving target, Exp3 doesn’t seem to be the most appropriate choice.
>
> * In our numerical studies (Appendix A), we’ve already compared our algorithm with a variation of Exp3—the RExp3 algorithm from [Besbes et al. 2014]. RExp3 uses Exp3 as a subroutine, restarting at the start of each epoch. Similar to AR2, their restarting mechanism balances the "remembering" and "forgetting" tradeoff, improving Exp3's performance in non-stationary bandits with dynamic benchmarks.
>
>     Yet, there are key differences between RExp3 and our approach: (i) RExp3 assumes sublinear total variation in expected rewards, while our AR setup involves $O(T)$ total variation, and (ii) RExp3 does not use any knowledge of the temporal structure. Theoretical results for RExp3 merely suggest that its total regret would be $O(T)$ in our setup, but do not offer any per-round regret guarantees in terms of AR parameters $\alpha$ and $\sigma$. Our simulations (Table 2, Appendix A) also show that RExp3 does not perform well in the AR setup, as it's tailored for environments with limited changes (see Appendix A for more discussion).
>
> * For completeness, we have additionally evaluated the performance of Exp3 and compared it with AR2 and RExp3. Please refer to _Table 3 in the PDF of the global response_. From the table, we see that Exp3's performance is very close to (sometimes slightly worse than) RExp3, which is expected considering their similarities. Nonetheless, AR2 performs much better than EXP3/Rexp3 in all types of environments, as AR2 leverages knowledge of the AR temporal structure to swiftly adapt to changes.
>
> * Please also refer to Table 2 in Appendix A to see the comparison of AR2 with other algorithms that adopt the static benchmark (e.g., UCB, $\epsilon$-greedy, etc.). There, AR2 also displays superior performance.
>
> **Regarding our choice of the dynamic benchmark,**
>
> * Following our discussion on Exp3, if we were to adopt a static benchmark in an adversarial setting, the Exp3 algorithm could already yield a $O(\sqrt{T})$ static regret. Yet, as illustrated by our numerical results (Table 2 in Appendix A and _Table 3 in the PDF of global response_), algorithms tailored for static benchmarks, such as UCB and Exp3, do not adapt well to the rapid changes inherent to our setting.
>
> * While we recognize that the dynamic benchmark presents a tougher challenge than the static benchmark and does not permit sublinear regret, it allows us to aim high and track the best moving target. This is especially important in a fast-changing environment that we studied, as no single arm can consistently dominate. Algorithms designed for static benchmarks tend to fall behind, as seen in our numerical studies. Essentially, the dynamic benchmark highlights the value of leveraging temporal structure in online decision-making, which is our main goal, and also where our algorithm excels.
>
> * The real-world applications discussed in Sec. 1 also underscore the importance of adopting a dynamic benchmark. In practice, the decision-maker usually wishes to be agile and swiftly react to changes in the environment. For example, in online product recommendations, as product demand shifts, it's vital to quickly adjust to show in-demand items. Relying on a single historically "best" product can’t be optimal in such dynamic scenarios. The same rationale applies to predicting ad CTR, where ad platforms must pivot quickly in response to fluctuating CTRs.

---

> > ### Comment · Reviewer_s53A · 2023-08-10
> > **Thank you**
> >
> > Thank you for such a detailed response!

---

### Official Review · Reviewer_6t1b · 2023-06-27

**Soundness:** 3 good
**Presentation:** 3 good
**Contribution:** 2 fair
**Rating:** 5
**Confidence:** 3

**Summary:**

This work uses AR(1) model to model the non-stationary multi-armed bandit (MAB) problem. This paper considers a new performance metric, dynamic steady-state regret, and establishes lower bound on regret. Furthermore, This paper proposes the AR2 algorithm and provides a relatively tight regret upper bound.

**Strengths:**

Previous works on non-stationary multi-armed bandit (MAB) problems have often assumed bounded non-stationarity, as only bounded non-stationarity allows for achieving sublinear dynamic regret. This work, however considers the setting of unbounded non-stationarity. This provides a new perspective on handling non-stationarity and addresses the evaluation of algorithm performance in the context of linear regret. Additionally, this paper introduces the concept of steady state error to quantify the algorithm's performance under these conditions.

**Weaknesses:**

1. In this paper, the AR(1) model is defined with α∈(0,1), which actually qualifies it as a stationary model. α would need to be at least equal to 1 to be considered non-stationary. For example, in scenarios involving bounded non-stationarity, the expected reward's variation bound is typically represented by \sum_{t=1}^T||r(t+1)-r(t)||_\infty, which corresponds to the case of AR(1) with α=1. Therefore, using this stationary AR(1) model ( α∈(0,1) ) to characterize non-stationary multi-armed bandit (MAB) problems seems to be unreasonable.

2. It seems unreasonable to assume that the expected reward has upper and lower bounds of [-R, R] in this setting. If we assume that the expected reward is bounded within [-R, R], even with random arm selection, the per-round dynamic regret should be at most 2R. This implies that any algorithm (e.g., algorithms without restarts) would eventually reach a steady state, which is unrealistic in a non-stationary environment. However, Theorem 5.1 in the paper does not seem to make this point clear: as α approaches 1, the per-round dynamic regret upper bound for random arm selection tends to infinity (which should be at most 2R when the expected reward is bounded by [-R,R]).

**Questions:**

I hope the authors can provide an explanation for the issues I mentioned regarding the stationary AR(1) model, the bounded expected rewards, and Theorem 5.1.

If the response is reasonable, I will consider increasing the score.

**Limitations:**

As mentioned in the weakness.

---

> ### Author Rebuttal · Authors · 2023-08-09
>
> Thank you for your thoughtful feedback! We’d like to address each of your questions below. We will also integrate the following discussions and clarifications into our revised paper.
>
> **(1) Stationarity in time series vs. bandits.**
>
> We appreciate your comments regarding the AR-1 model's stationarity and would like to clarify the terminology used in our paper.
>
> * In time series analysis, we agree that the definition of a stationary time series is a stochastic process with a constant unconditional joint probability distribution, which characterizes the AR-1 model with $\alpha < 1$ as a stationary time series model.
>
> * However, within the bandit literature, the terms "stationary" and "non-stationary" are used differently. "Stationary bandits" refer to setups where the underlying distribution of each arm remains fixed, while "non-stationary bandits" encompass any scenario where the reward distribution of each arm changes over time. In our context, since the expected reward $r_i(t)$ varies over time following the AR process and since the best arm (i.e., the arm with the highest expected reward) changes over time, our problem aligns with the non-stationary bandit category. In fact, our work delves into a notably more volatile environment than most studies within the non-stationary bandit literature (e.g., [10, 52]).
>
> * We suspect the term “non-stationarity” might be causing some confusion due to its varied interpretations across different domains. We are open to renaming our setting to “dynamic bandits” if this helps alleviate potential misunderstandings and align terminologies with existing literature. We thank the reviewer for pointing this out and will make the necessary adjustments for better clarity in the paper.
>
> * We'd also like to remark on the total variation term:
>      - In our context, the total variation you mentioned can be computed as follows. Recall that our AR-process is defined as $r_i(t+1) = \alpha r_i(t) + \alpha \epsilon_i(t)$.
> Hence, the total variation is given by $\sum_{t=1}^T \Vert r(t+1) - r(t) \Vert_\infty = \sum_{t=1}^T \Vert (\alpha-1) r(t) + \alpha \epsilon(t) \Vert_\infty,$ where $r(t) = (r_1(t), \dots, r_k(t))$ and $\epsilon(t) = (\epsilon_1(t), \dots, \epsilon_k(t))$.
>
>     - Note that the total variation here scales linearly with the number of rounds $T$. To see this, consider a simple example where $\alpha = 1/2$, boundary $R = 1$, and for illustrative purposes, assume $\epsilon_i(t)$ independently takes value from $\pm 1$ with equal probability $1/2$. In this simplified scenario, for any $r(t) \in [-1,1]^k$, the per-round variation at any given round (i.e., $||r(t+1) - r(t)||_\infty$) would be at least $1/2$ with probability at least $1/2$. Therefore, the total variation is $O(T)$.
>     - This suggests that our AR-based environment is more volatile than settings characterized by sublinear total variation, such as the one studied in [Besbes et al. 2014]. Our established lower bound also aligns with their lower bound result, which states that if the total variation is $O(T)$, our best achievable total regret would be $O(T)$ (see Line 184-187 of Section 3). Hence, studying per-round regret and presenting upper bounds based on AR parameters $\alpha$ and $\sigma$ would provide better insights into an algorithm’s performance in our rapidly changing environment.
>
> **(2) Per-round regret is bounded.**
>
> * Given the bound $R$ on our reward, we agree with the reviewer that a more accurate statement of Theorem 5.1 would be: **any algorithm** (including the naive algorithm) would incur per-round regret of at most $\mathbf{O(\min(\sqrt{\log (1/\alpha \sigma)+\log k} \cdot \frac{\alpha \sigma}{\sqrt{1-\alpha^2}}, 2R))}$. We would update this in the revised paper to add clarity.
>
> * The main purpose of presenting Theorem 5.1 is to show that under the setting where $\alpha$ is small (i.e., $\alpha < \bar{\alpha}$), the upper bound attainable by any algorithm (even the naive algorithm) stays close to the lower bound (refer to Figure 1). As Figure 1 also shows, when $\alpha \rightarrow 1$, the upper/lower bounds deviate from each other, highlighting the need to design an algorithm for the case of large $\alpha$ as we did in the paper (see Theorem 5.2).
>
> * We’d like to further add that the lower bound presented in Section 3 scales with the AR parameters $\alpha$ and $\sigma$. If the constant $R$ is large enough, our lower bound would be quite far away from the bound of $2R$. Hence, assuming $R$ is large enough, our goal is to devise an algorithm whose per-round regret upper bound can be characterized in terms of the AR parameters ($\alpha, \sigma$), and ensure that its upper bound stays close to our lower bound.
>
> * In our setting, in the absence of such boundedness, the variance of expected rewards under the stationary distribution would go to infinity as $\alpha$ goes to 1, and it then becomes impossible to design any useful algorithm for such a volatile environment. Further, note that the boundedness of rewards is a common assumption in the MAB literature (e.g., Lai & Robbins 1985, Auer et al. 2002, Besbes et al. 2014), and hold in most real-world applications such as demand forecasting and ad CTR prediction.

---

> > ### Comment · Reviewer_6t1b · 2023-08-19
> > **Response to Rebuttal**
> >
> > Thank you for the explanation. I still have one question. If the expected reward is bounded by [-R, R], intuitively, this R should directly impact the regret, or, in your context, the steady-state error, such that, this bound should scale with R, as implied by the series of works mentioned in your rebuttal (Lai & Robbins 1985, Auer et al. 2002, Besbes et al. 2014). However, the O(min(..., 2R)) proposed by the authors in the rebuttal still doesn't seem to capture the influence of R on the problem.

---

> > > ### Author Response · Authors · 2023-08-19
> > > **Rebuttal by Authors**
> > >
> > > Thank you for your question! Let us provide further clarifications below.
> > >
> > > - The upper bound in Thm 5.1 can be slightly updated to reflect full dependency on $R$, as $O(\min(\sqrt{\log(1/\alpha \sigma)+\log k} \cdot \frac{\alpha\sigma}{\sqrt{1-\alpha^2}} + \mathbf{R \alpha \sigma} , 2R))$. Please refer to Appendix G for the proof of this bound.
> > >
> > > - We initially left out the term $R \alpha \sigma$ in our Big-O notation. This was because, with $R$ being a constant, this term has the same dependency on $\alpha$ and $\sigma$ as the main term, $\sqrt{\log (1/\alpha \sigma)+\log k} \cdot \frac{\alpha \sigma}{\sqrt{1-\alpha^2}}$. The main term, however, has an added dependency on $\sqrt{\log (1/\alpha \sigma)+\log k}$, which makes it the dominating term in the Big-O notation. We can update the upper bound in Thm 5.1 to improve clarity.

---

> > > > ### Comment · Reviewer_6t1b · 2023-08-20
> > > > **Response**
> > > >
> > > > Thank you for the response. I believe it's important to explicitly highlight the influence of R in the regret bound. Additionally, I find the author's statement that the first term dominates to be rather ambiguous. I suggest that the authors should clarify in the revised version under what ranges of $alpha$, $sigma$, $k$, and $R$, this bound holds nontrivial, without devolving into a simple $2R$. Just like we consider a linear regret as a trivial result in the context of regret analysis, I believe that in the context of steady-state error, it's crucial to clarify the distance between the bound and $2R$.

---

> > > > > ### Author Response · Authors · 2023-08-21
> > > > > **Rebuttal by Authors**
> > > > >
> > > > > Thank you for your response. We will update our regret bound to show dependency in $R$.
> > > > >
> > > > > Regarding when the first term in Thm 5.1 becomes dominating, since Thm 5.1 primarily characterizes the small-$\alpha$ regime, we have that $\alpha \in (0, 0.5]$ and $\sigma \in (0, 1)$ (see Sec. 2 & 5). In this case, the first term dominates the $2R$ term as long as $R \ge 1$ and $k \le 400$.
> > > > >
> > > > > We will clarify the above in the revised version of the paper as you suggested.

---

### Official Review · Reviewer_jsRP · 2023-07-06

**Soundness:** 3 good
**Presentation:** 3 good
**Contribution:** 2 fair
**Rating:** 3
**Confidence:** 3

**Summary:**

This paper studies the problem of non-stationary bandit learning in bandits where rewards have auto-regressive temporal dependency. More specifically, this paper considers bandits where the evolution of the expected mean reward of each arm undergoes an independent AR-1 process truncated to a bounded interval. All arms share the same AR parameter $\alpha$. The authors propose an algorithm Alternating and Restarting algorithm for dynamic AR bandits (AR2), which takes the AR parameter $\alpha$, a stochastic rate of change $\sigma$, epoch size $\Delta_{\mathrm{ep}}$ and a parameter $c_0$ as input. The algorithm “addresses the exploration-exploitation tradeoff by alternating between exploiting the superior arm and exploring the triggered arm within each epoch,” and it “handles the tradeoff between remembering and forgetting” via restarting. To illustrate the efficacy of the algorithm, an upper bound on the regret of AR2 is established, and is compared with a lower bound on the regret. Numerical experiments and a real-world case study on tourism have also been conducted.

**Strengths:**

The paper is well-organized and the exposition is clear and easy to follow. The use of AR-1 model in characterizing non-stationary dynamics is also well-motivated.

**Weaknesses:**

The main weakness of the paper lies in the insufficient justification of the efficacy of the algorithm AR2. Specifically:
1. The paper suggests that a key mechanism of the algorithm is “an alternation mechanism adept at leveraging temporal dependencies to dynamically balance exploration with exploitation,” without discussing the intuition behind this counter-intuitive argument.
- One question I have is that it seems that via alternation, the algorithm explores and exploits at the same intensity. Shouldn’t the intensities change according to the AR-1 parameter $\alpha$ and the stochastic rate of change $\sigma$?
2. The regret upper bound established by Theorem 5.2 doesn’t sufficiently justify that AR2 performs well, due to the somewhat restrictive assumptions.
- The bound does not consider the range $\alpha \in [0, \overline{\alpha})$. Does AR2 perform well in this regime? In addition, I understand that some other naive algorithm performs well in this regime, but Theorem 5.1 cannot fully justify that because the bound on average regret explodes as the number of arms $k \rightarrow +\infty$ (the bound becomes vacuous when $k$ is large enough because rewards are bounded). Also, Theorem 5.1 does not apply to AR2.
- The assumption of $k \leq \mathcal{K}(\alpha)$ suggests that Theorem 5.2 can only be applied when the number of arms $k$ is small. For example, when $\alpha = \overline{\alpha} = 0.5$, the number of arms $\mathcal{K}(\alpha) = 2$: the theorem only applies to bandits with two arms.
- Moreover, the bound on average regret grows in $k^3$, which can be very large when the number of arms $k$ is moderately large.
3. I also have concerns on the numerical experiments.
- One concern is that although the paper suggests that a real-world case study has been completed, experiments were not conducted on the real data, but rather, synthetic data generated by an AR-4 model fitted to the data.

**Questions:**

My concerns and questions are raised in the Weakness section. In summary:
1. It would be great if the authors could explain the intuition behind why the alternation mechanism can effectively balance exploration with exploitation, although it seems that the extent to which the algorithm explores is the same as the extent to which the algorithm exploits.
2. Can the authors provide justifications of how well the algorithm AR2 performs when each of the assumptions $\alpha \in [\overline{\alpha}, 1)$, and $k \leq \mathcal{K}(\alpha)$ in Theorem 5.2, is violated, and when $k$ is large?
3. Can the authors explain why experiments are run against synthetic data instead of real data, when real data is provided in the tourism case study?

---

> ### Author Rebuttal · Authors · 2023-08-09
>
> Thank you for your valuable feedback! We’d like to address each of your questions below.
>
> **(1) Regarding the alternation mechanism,**
> * We'd like to first clarify that we do not always explore and exploit at the same intensity. At the exploration step, we only pull a triggered arm during odd rounds if the triggered set $\mathcal{T}$ is non-empty (see Alg. 1). Our triggering condition (Eq. 2), defined by AR parameters ($\alpha, \sigma$), naturally determines the rate of exploration.
> * Unlike stochastic bandits, the dynamic environment mandates **continuous exploration** alongside continuous exploitation because the best arm keeps changing. Neglecting any arm for too long reduces our confidence in its potential. This makes balancing exploration and exploitation especially challenging (as discussed in Sec. 1), since the rate of exploration needs to be adjusted based on the amount of changes in our environment. Our alternation mechanism is tailored for this task, with the help of the superior arms and the triggered set.
>
>     To see that, consider the following two scenarios:
>     - In slower-changing environments (e.g., with small $\sigma$), the triggered set may not always include arms needing exploration, allowing focused exploitation of the superior arm. For instance, if there are only two arms whose rewards change slowly, one might dominate for extended periods before the other arm meets the triggering condition (Eq. 2). In this case, our alternation mechanism makes exploitation its primary strategy, only exploring occasionally.
>     - In fast-changing environments where the best arm shifts rapidly, the triggered set is likely to always contain some arms worth exploring. Here, the rate of the exploration can be as high as the rate of exploitation to ensure that we keep track of all arms while not over-exploring.
>
> We will add the above discussion to Section 4 to add more clarity and justification.
>
> **(2a)  Regarding the regime of small $\alpha$ and Thm. 5.1,**
> * We’d like to clarify that a more accurate statement of Thm. 5.1 should be that **any algorithm** (including the naive algorithm and our algorithm AR2) would incur per-round regret at most $\mathbf{O(\min(\sqrt{\log (1/\alpha\sigma)+\log k}\cdot\frac{\alpha\sigma}{\sqrt{1-\alpha^2}},2R))}$. The proof of Thm. 5.1 does not use any property unique to the naive algorithm. Our initial statement is meant to emphasize that in the small $\alpha$ regime, even the regret upper bound of a rudimentary algorithm matches the lower bound (see Fig. 1).
> * We remark that it is reasonable for the bound to scale with $k$ here. Even in vanilla stochastic bandits, the per-round regret scales with $\sqrt{k/T}$, inherently increasing in $k$.
> * Given that Thm. 5.1 suggests theoretically sound performance for any algorithm in the small-$\alpha$ regime, we therefore focus on analyzing the performance of AR2 in the large-$\alpha$ regime in Thm. 5.2.
> * Moreover, our numerical studies in Appendix A show the superior performance of AR2 in both small-$\alpha$ and large-$\alpha$ regimes compared to various benchmarks.
>
> We will revise the statement of Thm. 5.1 and add clarity in our revised paper.
>
> **(2b/2c) Regarding the number of arms $k$,**
>
> Thank you for your question on (i) the bound $k \leq \mathcal{K}(\alpha)$ and (ii) the dependency on $k$ in Thm. 5.2. We will address both points below.
> * The bound $k \leq \mathcal{K}(\alpha)$ is mainly required for the rigor of theoretical analysis. Both this bound $\mathcal{K}(\alpha)$ and the $k^3$ dependency in our upper bound come from the loose upper bound we used for the number of triggered arms (in Lemma H.3, we used the fact that at any given round, there are at most $k-1$ triggered arms). See Line 252-259 of Sec. 5 for more details.
> * Theoretically, if at any given round, the number of triggered arms is $O(1)$, the bound $k \leq \mathcal{K}(\alpha)$ would no longer be needed, and our upper bound can be tightened to $O(c_0^2\alpha^2 \sigma^2k\log(c_0\alpha\sigma))$, which exactly matches the lower bound up to logarithmic factors.
> * Our numerical studies in Appendix A also reveal that (i) AR2 maintains its competitive performance even when $k$ exceeds $\mathcal{K}(\alpha)$; and (ii) our per-round regret grows modestly with $k$, despite the theoretical $k^3$ dependency. These suggest that the assumption/dependency on $k$ for our upper bound are artifacts of our analysis, rather than an intrinsic property of our algorithm.
>
>     For instance, when $\alpha = 0.9$, we’d have $\mathcal{K}(\alpha) = 10.4$, but AR2 remains competitive even when $k=20$. Also, as the number of arms doubles from 10 to 20, the regret of AR2 increases gracefully by 20%, significantly milder than the $k^3$ dependency. See Appendix A for more details.
>
> **(3) Regarding the real-world case study,**
> * Using raw data from [36] in our case study wasn’t feasible for two reasons.
>     - Their data only consists of quarterly arrivals during 1975-1989, amounting to merely 60 data points in total. For our algorithm's learning process to be meaningful, a substantially larger dataset is crucial.
>     - Our bandit setup seeks to establish competing arms (demand for vacation packages) correlated with the same exogenous variables (the tourism demand). Yet, the provided dataset comprises only a single time series without accompanying data on the demand for vacation packages.
> * To address these, we used the AR parameters from [36] to model demand for different vacation packages (arms). The authors of [36] fitted the logarithms of quarterly tourist arrivals to an AR-4 model, validating each lag with computed t-statistics. This model allowed us to craft time series for competing arms over a multitude of rounds.
> * Finally, we’d also like to highlight that the main purpose of our case study is to demonstrate the adaptability of AR2 to more complicated time series with real-world characteristics (AR-p processes with trend). This has indeed been shown in our case study.

---

> > ### Comment · Reviewer_jsRP · 2023-08-19
> > **Primary Concerns Remain Unresolved After Reading the Author Response**
> >
> > Thank you for your detailed response!
> > Although some of my questions are addressed,  my primary concerns remain unresolved.
> >
> > 2 (b/c)  From your response, $k \leq \mathcal{K}(\alpha)$ is indeed required as an assumption, and that you agree that $k^3$ is the dependency given by the theoretical results, where $k$ is the number of arms. As I suggested in my review, the assumption is strong, and the dependency of $k^3$ on $k$ is worse than other regret bounds established in the literature.
> >
> > 2 (a)  Regarding small $\alpha$ regime.
> > * Since the paper lets $\epsilon_i(t) \sim \mathcal{N}(0, \sigma)$, in the AR-p setting, the long-run average reward of all arms are zero. This setting is strange because the AR-p setting doesn't encompass stationary environments as special cases. For example, the setting doesn't encompass stationary environments where different arms have different means as special cases.
> > * Moreover, it is the fact that the long-run average reward of all arms are zero that makes Theorem 5.1 applicable to all algorithms. Indeed, Theorem 5.1 does not even apply to all stationary bandits and all algorithms: the bound converges to $0$ as $\alpha \rightarrow 0$, but an algorithm that always aim to pull the worst arm cannot incur such small regret.
> > * When Theorem 5.1 cannot be applied to AR2, it is concerning how the algorithm performs in this regime.
> >
> >
> > 3  It is claimed in both the abstract and the introduction that this paper conducts a real-world case study on tourism demand prediction, yet your response suggested that you could not use the raw data in this dataset ....
> > * Why not use the raw data in other datasets presented in other bandit learning papers? For example, the one used in Chapelle and Li 2011? Or, Zhou et. al. 2020 (https://arxiv.org/abs/1911.04462)?
> > * Since your algorithm is specifically designed for AR-p bandits, it is unfair to claim that you conduct a real-world case study, while comparing your algorithm with other algorithms designed for general non-stationary bandits on AR-p data (which are generated from AR-p simulators fitted from real data) instead of real data.
> >
> > 1  Could you explain why the algorithm AR2 alternates between 1 exploration period and 1 exploitation period, instead of e.g. have 1 exploration period but 2 exploitation periods? What's the intuition behind this specific alternation pattern?
> >
> > Thus, I will keep my score as 3.

---

> > > ### Author Response · Authors · 2023-08-19
> > > **Rebuttal by Authors**
> > >
> > > **(2b/c) Regarding the number of arms $k$,**
> > > - Our numerical studies (Appendix A) show that the assumption in Thm 5.2 is **not** necessary and can be relaxed with a tighter analysis. These studies also show that the regret increases gracefully with $k$. The assumption and regret dependency on $k$ are **only** an artifact of our analysis (in particular Lemma H.3) and are not inherent to our algorithm.
> > > - We’d like to ask the referee for their understanding. Our research is in fact the first to investigate AR-based non-stationary bandits under a strong dynamic benchmark, aiming to characterize per-round regret w.r.t. AR parameters. We've devised an algorithm adaptable for different AR-based processes, which shows superior performance against all non-stationary benchmarks, including a follow-up work [39] that AC recently mentioned.
> > >
> > > **(2a) Regarding the small-$\alpha$ regime,**
> > > - **Regarding Thm 5.1,**
> > >     - We believe that there is a **misunderstanding**. The referee thinks that the reason Thm 5.1 holds is because the long-run average reward goes to zero. This is not true. The reason that Thm 5.1 holds is because for small $\alpha$, the steady-state distributions of expected rewards for all arms cluster around zero. These two statements are not equivalent.
> > >     - To see that, consider a case unrelated to our setting, where each arm's reward switches between $\pm1$. Here, the long-run average reward of the arm is 0. But, as the referee suggested, an algorithm that keeps pulling the worst arm would incur a (dynamic) loss of 2 at every single round.
> > >     - In the small-$\alpha$ regime, the expected rewards for all arms cluster around zero under the steady-state. This means that at any given round, the gap between the best and worst arms is extremely small, so even an algorithm that always pulls the worst arm won't incur significant (dynamic) loss.
> > > - **Regarding arms having the same steady-state mean of expected rewards,**
> > >     - Our setting is more challenging than those with distinct steady-state means of expected rewards. When arms have different steady-state means, there usually exists a dominating arm, and hence an algorithm designed for static benchmark is expected to work well. In our case with equal steady-state means of expected rewards, to achieve a good regret against the dynamic benchmark, we need to closely monitor the expected rewards of each arm over time to select the high-reward arm in every round.
> > >     - We further note that Thm 5.1 can be extended to a setting where the arms have the same, nonzero steady-state mean of expected rewards (e.g., consider $r(t+1)=\alpha r(t)+(1-\alpha)c+\epsilon(t)$, which yields a steady-state distribution clustered around $c$ for small $\alpha$). As stated earlier, this is a challenging setting as there does not exist a dominating arm.
> > >
> > > **(3) Regarding the case study,**
> > > - Upon a quick look at the papers you suggested, [Zhou et al. 2020] used data from UCI Machine Learning Repository, which are mainly classification datasets, so it is not clear if there exists AR-based temporal structure. The display advertising data in [Chapelle & Li 2011] can be potentially useful; however, the data is not publicly available, as claimed by the authors of “Estimating rates of rare events with multiple hierarchies through scalable log-linear models”, who used the same data.
> > > - We've explained the needs for certain data simulation due to limitations in the available data. We’d like to further add that using AR-p simulations fitted from real-world data ensures a controlled experimental environment, enabling us to rigorously assess the benefits of leveraging knowledge of the temporal structure.
> > > - **Regarding benchmark algorithms in our case study/numerical studies**, until recently, there did not exist any algorithm designed specifically even for AR-1 process. We've already taken one step further and modified the UCB algorithm based on the AR process (see mod-UCB, Appendix C). See also the predictive sampling (PS) algorithm, which we tailored for our AR-1 setup, that we simulated in our response to the AC.
> > >
> > >     This necessitates comparisons with algorithms for general non-stationary MAB, under an AR-based setup. Our choice of the dynamic benchmarks includes RExp3 and sliding window UCB/TS (see Appendix A), similar to related follow-up work [39].
> > >
> > > **(1) Regarding alteration mechanism,**
> > > - Our current alternation mechanism is a design choice, with both theoretical and empirical evidence confirming its efficacy.
> > > - Adjusting the exploration/exploitation pattern (e.g., 2 rounds of exploitation and 1 round of exploration) would result in the same theoretical upper bounds and we'll comment on this.
> > > - The true value of our alternation mechanism lies in its adaptability to various time-varying environments through the triggering condition. In slowly-changing environments, exploitation naturally increases, while in fast-changing environments, the mechanism increases the amount of exploration.

---

> > > > ### Comment · Reviewer_jsRP · 2023-08-21
> > > > **Thank you!**
> > > >
> > > > Thank you for the prompt and detailed response! After reading the responses, what I perceive is that:
> > > >
> > > > (2 b/c) The regret bounds established in this paper depends on the assumption that $k \leq \mathcal{K}(\alpha)$, and that $k^3$ is the dependency given by the bound, where $k$ is the number of arms. However, the numerical results suggest that the algorithm performs well despite the assumption and the loose regret bound.
> > > >
> > > > (2 a) "The reason that Theorem 5.1 holds is because for small $\alpha$, the steady-state distributions of expected rewards for all arms cluster around zero." (author response) Theorem 5.1 can be extended to the case where different arms have "the same steady-state mean of expected rewards" (author response), but cannot be applied to all algorithms when different arms have different steady-state mean of expected rewards.
> > > >
> > > > (3) For various reasons, it is hard to use raw data from any real dataset, including the one used in the paper and some other datasets used in the bandit learning literature. As a result, data simulated from an AR-4 model fitted to real-world tourism data is used in the real-world case study.

---

> > > > > ### Author Response · Authors · 2023-08-21
> > > > > **Thank you and a final remark**
> > > > >
> > > > > Thanks again for your insightful comments; we hope you find our responses satisfactory. Let us also make a final remark about point (2a).
> > > > >
> > > > > As the referee observed, when different arms have different steady-state means of their expected rewards, not all algorithms would perform well in the small-$\alpha$ regime. Nonetheless, as we noted in our prior response, under this setting there would exist a dominating arm, so we’d expect an algorithm designed for a static benchmark to already perform well. This is why our main focus is to characterize the more challenging setting, i.e., when all arms have the same steady-state mean of expected rewards in the large-$\alpha$ regime.
> > > > >
> > > > > There also exists a natural extension of our algorithm if each arm has a distinct steady-state mean of expected reward $c_i$. For example, one plausible time series model is to assume that the expected reward $r_i(t)$ evolves as $r_i(t+1) = \alpha r_i(t) + (1-\alpha) c_i + \epsilon_i(t)$ (see Sec 8.1 in [39] for a similar model). In this case, we can simply adjust the estimated reward update in AR2 to be $\hat{r}_i (t+1) = \alpha \hat{r}_i (t) + (1-\alpha)c_i$, while all other steps remain unchanged. This shows the adaptability of our algorithm to different time series.

---

### Official Review · Reviewer_UBzK · 2023-07-07

**Soundness:** 4 excellent
**Presentation:** 4 excellent
**Contribution:** 3 good
**Rating:** 7
**Confidence:** 4

**Summary:**

This paper studies a non-stationary bandits problem where the reward rate of each arm evolves according to the AR(1) model, i.e., shrinks by a known multiplicative factor and adds an indep noise. They proposed an algo that strikes a balance between (i) exploration v. exploitation and (ii) remembering v. forgetting. As their main results, they showed that this algo achieves nearly optimal regret. Moreover, they presented a real world case study.

**Strengths:**

- the model is novel, neat and fundamental.
- provided a nearly optimal regret
- presented a high quality real world case study
- the manuscript is well-written


**Weaknesses:**

- Since the main term in the regret is sigma, it is better to specify the function g in the lower bound (Thm 3.1), especially how it depends on sigma. After skimming the appendix, I am still not 100% sure whether the claimed UB and LB really match.

**Questions:**

- To ensure boundedness of the mean reward, you truncated the AR process. Why is this wlog?
- this problem is reminiscent of the Slivkins Upfal COLT'08 paper - that paper also considered stochastically evolving reward rates, truncated at the boundary, and their results mainly depends on \sigma. Can you comment on the connections between these two papers? E.g. does the S-U paper implies any preliminary results for your problem?
- Can this problem be approximately viewed as RL? Specifically, assuming alpha is a constant, then the history older than log # rounds has little impact on the current reward rate. Can we encode the log n - step history as the state? (If so, the transition matrix is also known) Can we immediately obtain a regret bound using existing RL results?
- the theoretical part considers AR(1) but the case study considers AR(4). Is this difference essential?

other comments:
line 145: "2exogenous" - is this a typo?

**Limitations:**

(typesetting issue) when I click on a reference, the pdf file does not jump to the reference page

---

> ### Author Rebuttal · Authors · 2023-08-09
>
> Thank you for your thoughtful feedback! We’d like to address your questions below. We will incorporate the following discussions and fix the typo in our revised paper.
>
> **(1) Dependency of function $g$, upper and lower bounds on $\sigma$.**
>
> * Our function $g(k, \alpha, \sigma)$ represents the probability that two best arms are within $\alpha\sigma$ of each other under the stationary distribution. To see how $g$ evolves in terms of $\sigma$, we can consider the scenarios of small $\alpha$ and large $\alpha$ separately.
>
>     As depicted in Figure 3 in Appendix D, when $\alpha$ is small, the stationary distribution resembles a normal distribution with standard deviation proportional to $\alpha\sigma$; while when $\alpha$ is close to one, the stationary distribution resembles a uniform distribution. Hence, when $\alpha$ is small, we expect $g$ to be roughly constant in terms of $\sigma$, while when $\alpha$ is close to one, $g$ should be roughly linear in terms of $\sigma$.
>
>     Our statement above is confirmed in _Figure 5 in the PDF of the global response_, where we numerically computed the values of $g$ for different $\sigma$ under the two scenarios ($\alpha = 0.4$ and $\alpha = 0.9$) respectively.
>
> * Having seen the evolution of $g$, we can see that our upper bound (UB) and lower bound (LB) indeed match in terms of $\sigma$.
>     - For the regime with small $\alpha$, we illustrate the evolution of our UB and LB in _Figure 6 in the PDF of the global response_. This plot reveals that our UB and LB have the same trend of increase in terms of $\sigma$ in the small-$\alpha$ regime, which complements our result in Figure 1 of Section 5 (that shows our UB and LB match in terms of $\alpha$ in the same regime).
>     - For the regime with large $\alpha$, we have already illustrated the evolution of UB, LB and the per-round regret of AR2 in Figure 4(b) of Appendix E, which again shows that the UB and LB follow the same trend of increase with respect to $\sigma$.
>
> **(2) Boundness of rewards and our truncation approach.**
>
> * Here, we adopt a truncating approach at the boundary $[-R, R]$ mainly because it is a simple and natural method to ensure boundedness of the expected rewards. However, it's important to note that this truncating boundary is merely one of the many we're equipped to handle. Our main results would remain valid across various boundary conditions such as reflecting or absorbing boundaries, with slight changes in the PDF/PMF of the stationary distribution (see Appendix D for our discussion).
>
> * In our setting, in the absence of such boundedness, the variance of expected rewards under the stationary distribution would go to infinity as $\alpha$ goes to 1, and it then becomes impossible to design any useful algorithm for such a volatile environment. Further, note that the boundedness of rewards is a common assumption in the MAB literature (e.g., Lai & Robbins 1985, Auer et al. 2002, Besbes et al. 2014), and hold in most real-world applications such as demand forecasting and ad CTR prediction.
>
> **(3) Comparison with [Slivkins et al. 2008].**
>
> * [Slivkins et al. 2008] mainly studies non-stationary bandits where the rewards evolve as Brownian processes. Their setup can be viewed as a special case of the AR model where $\alpha = 1$ (see our comment in Section 1.2). In fact, if we simply take $\alpha = 1$, our method and results are also valid for the Brownian process, and our upper/lower bounds would share the same dependency on $\sigma$ as their upper/lower bounds.
>
> * However, our general AR model studies a broader spectrum of environments that accommodates $\alpha \in (0,1)$. As such, the algorithm designed for Brownian bandits is not directly applicable for general AR bandits. One key challenge that arises from the AR model is that the correlation between past and future information now decays exponentially fast. Hence, the design of our algorithm is different from theirs in two important aspects:
>     - While the algorithm of [Slivkins et al. 2008] can seamlessly leverage past observations from any arm to estimate its current state, we have to factor in the exponential decay of past values we've observed. This impacts our definition of the estimated reward, triggering condition, and the rate of the alternation mechanism.
>     - Our approach also mandates periodic restarts to judiciously discard outdated information. This becomes crucial for the general AR setup as any older data can swiftly become misleading.
>
> **(4) Viewing the problem as an RL problem.**
> * Traditional RL methods (e.g., Q-learning) operate best with limited, finite state space. In our case, following your suggestion, the state would be a $\log(T)$-dimensional vector, where each coordinate can be any real number between $[-R, R]$. That is, the size of the state space is infinite, and hence such an algorithm cannot be implemented in polynomial time. Discretizing the state space could be a potential alternative, but the impact of such discretization on regret remains unclear given how the AR process evolves.
>
>     Nonetheless, we really appreciate your suggestion and believe that it is certainly a promising area for further exploration. We will mention it in our future works section in the revised paper.
>
> **(5) Our theory and the case study.**
> * We would like to remark that our theory and case study serve different purposes. We develop our theory around AR-1 processes mainly to shed light on the design of our two main mechanisms (alternation and restarting) and show near-optimally of our algorithm with theoretical rigor. On the other hand, our case study highlights the adaptability of AR2 to more complicated time series observed in the real-world (AR-$p$ processes with trend), underscoring its robust performance. Our hope is that the combined insights from our theoretical and empirical findings would inspire more works that explore bandits in a time-varied environment.

---

> > ### Comment · Reviewer_UBzK · 2023-08-19
> >
> > Thanks for your response.

---

### Author Rebuttal · Authors · 2023-08-09

We would like to express our sincere gratitude to all reviewers for your valuable and insightful feedback! We have carefully addressed each reviewer’s comments and questions below. For Figures 5, 6 and Table 3 crafted during the rebuttal period, please kindly refer to the attached PDF in this global response. We will incorporate all clarifications and discussions in our response into the revised version of the paper.

Our work is, to our knowledge, the first that studies non-stationary bandits with reward distributions governed by time series that encapsulates real-world characteristics, specifically the AR process. Our aim is to identify the major challenges inherent to such rapidly changing environments commonly observed in practice, and propose effective mechanisms for addressing them. We hope that the high-level ideas shown in this paper would provide insights for any future research that explores bandits with other types of temporal structures.

Once again, thank you for your time and thoughtful contributions to improving our work!

---

### Author Response · Authors · 2023-08-18

We would like to express our gratitude once more to all the reviewers for dedicating their time to evaluate our submission. As the period for discussion is nearing its conclusion, we are wondering if we have effectively addressed the concerns raised by the reviewers.
 If there remain any uncertainties, we are more than willing to provide additional explanations.

---

> ### Comment · Area_Chair_H4x7 · 2023-08-18
> **A few more questions from me**
>
> Dear authors,
>
> I am chiming in as after looking at the paper myself I have one more question. You mention the recent paper ([39] in your list): "Non-Stationary Bandit Learning via Predictive Sampling". In this paper, the authors propose a Thompson Sampling for AR(1) bandits (section 8.1). Why did you not implement this algorithm in your benchmark? Why could you not compare with algorithms minimizing the dynamic regret (windowed or discounted UCB with regret bounds, or change-detection based algorithms, or restless bandit algorithms)?
>
> I am also generally surprised by the lack of comments on your rather original and unusual choice of regret: it is asymptotic, as opposed to most regrets studied in this literature. I would have expected a deeper discussion. It also differs from the static regret studied in rested / restless bandits. In their recent follow-up ("A Definition of Non-Stationary Bandits") the authors discuss in depth the choices of regret definitions for non-stationary bandits but they do not talk about asymptotic metrics (*). My main concern is that when you consider the limit of a Cesaro average like you do, you're essentially looking at a static limit (the convergence of the series of dynamic regret). So why does your regret choice make more sense than another here?
>
> Your AC
>
> (*) I am not suggesting you to cite it as I believe it is posterior to the submission, I just refer to it in case it is relevant for your work.

---

> > ### Author Response · Authors · 2023-08-18
> > **Thank you for your questions and constructive feedback!**
> >
> > Dear AC,
> >
> > Thank you for your questions! We appreciate the opportunity to address your questions, and will add the following discussions into our revised paper.
> >
> > **Regarding comparison with [39],**
> > - Thank you for this constructive feedback! For completeness, we have additionally evaluated the performance of the predictive sampling (PS) algorithm in [39] under our AR-1 setup (which is slightly different from their AR-1 bandit example in their Section 8.1). The experiments are done under the same setup as our numerical studies in Appendix A. (It is unclear how to implement the PS algorithm for the AR-4 model in our case study. As acknowledged by the authors of [39], deriving the conditional probability distribution in their PS procedure and implementing it can be complex in general.)
> >
> >     In the Table below, we list the performance of AR2, PS, as well as sliding window Thompson sampling (TS) (see comparisons with more benchmarks in Table 2 of Appendix A). Our results concur with the results in [39] that PS dominates TS-based approaches in non-stationary environments. However, our comparison shows that AR2 remains competent in all kinds of AR-based setups.
> >     | $\mathbb{E}[\alpha]$|$k$|AR2|Predictive sampling (PS)|Sliding Window TS
> >     | :--- | :--- | :----: | :----: | :----: |
> >     | 0.4|2|0.38 (0.10)|0.51 (0.04)|1.00 (0.03)
> >     | 0.4|10|0.67 (0.01)|0.78 (0.03)|0.98 (0.01)
> >     | 0.4|20|0.72 (0.01)|0.84 (0.02)|0.99 (0.01)
> >     | 0.9|2|0.18 (0.06)|0.52 (0.33)|0.99 (0.15)
> >     | 0.9|10|0.40 (0.04)|0.47 (0.03)|0.86 (0.04)
> >     | 0.9|20|0.49 (0.02)|0.52 (0.02)|0.91 (0.02)
> >
> >     One main difference between PS and AR2 is that, for each arm, PS samples from a normal distribution with *fixed* variance, while we *keep adjusting* our rate of exploration by adjusting size of the confidence bound in our triggering condition. As we discussed, this is important especially in volatile environments governed by AR processes.
> >
> > - We also note that it is difficult to directly compare the regret bounds of [39] for the PS algorithm with our results, because their regret bound depends on a term $\Delta_t$, which measures the “incremental predictive information”. In their work, the authors only provide interpretable regret bounds for modulated Bernoulli bandits, and it is unclear how to quantify $\Delta_t$ for our setting.
> >
> > **Regarding comparison with other dynamic algorithms,**
> > - We've already compared AR2 against a number of non-stationary MAB algorithms designed for dynamic benchmarks, including RExp3, sliding window UCB, sliding window TS, and a modified UCB approach tailored for AR-1. Please refer to **Table 2, Appendix A** for the comparisons. There, we show that AR2 is competitive against all non-stationary benchmarks, especially since many of them assume sublinear amounts of changes in the environment. We will add the result of PS to this table.
> >
> > **Regarding the definition of our regret**, we'd like to clarify that
> > - Our dynamic regret is defined asymptotically mainly because we want to evaluate the **steady-state** performance of our algorithm (i.e., under the steady-state distribution). This kind of regret notion has been adopted in prior works including [Slivkins et al. 2008].
> > - If we assume that the initial distribution of expected rewards is the steady-state distribution, we can remove the asymptotic limit in our definition. In this case, our total regret would exactly match the dynamic regret considered by “A Definition of Non-Stationary Bandits”, Liu et al. 2023 (see their Definition 2). This is also the approach taken by Liu et al. 2023 in their Examples 3 and 4, where they assumed the initial distribution of rewards follows the steady-state distribution.
> >
> >     Since the total dynamic regret scales linearly in $T$, our main focus is to characterize the per-round dynamic regret in terms of AR parameters $\alpha$ and $\sigma$.
> > - All of our theoretical analyses for per-round regret upper/lower bounds are performed under the steady-state distribution. Hence, as stated earlier, if we assume that the initial distribution follows the steady-state distribution, we can simply define our per-round regret as $\mathbb{E}\left[\frac{1}{T}\sum_{t=1}^T\text{REG}_A(t)\right]$ (which matches the dynamic regret Def. in Liu et al. 2023) and all results remain valid.
> > - In our case study (Sec. 7) and numerical studies (Appendix A), our regret is also the per-round dynamic regret in hindsight, i.e., the difference between the maximum attainable rewards and our obtained rewards, which matches the dynamic regret Def. in Liu et al. 2023.
> > - Finally, regarding restless and rested bandits literature that adopt static regret notions (e.g., “Online Learning of Rested and Restless Bandits” and “Online Model Selection: a Rested Bandit Formulation”), we remark that in our setting, the mean expected rewards of all arms are zero under the steady-state distribution, meaning that no arm would dominate. Hence, adopting a static regret would not be helpful.

---

### Decision · Program_Chairs · 2023-09-21

**Decision:**

Accept (poster)

**Comment:**

The reviewers and myself have come to an agreement that the paper should be accept due to the quality of the contributions as well as the clarity and topic of the paper.

We recommend the authors to add the discussions on the choice of regret metric and the additional results to the final draft as we believed they add value to the work.